# A Benchmark for Self-Evolving Agents via Experience-Driven Lifelong Learning

## Abstract

As AI advances toward general intelligence, the focus is shifting from systems optimized for static tasks to creating open-ended agents that learn continuously and adapt autonomously from experiences. This vision emphasizes long-term memory, self-driven exploration, persistent experience retention, and the internalization of knowledge into intuitive behavior as key to enabling self-evolving agents through experience-driven lifelong learning (ELL). In this paper, we introduce StuLife, a novel benchmark designed to evaluate whether current models can exhibit these foundational capabilities of ELL. Particularly, StuLife simulates a student's holistic college journey, from enrollment to academic and personal development, across three core phases and ten detailed sub-scenarios. StuLife is designed around four key paradigm shifts: **From Simulation to Reality**, **From Passive to Proactive**, **From Context to Memory**, and **From Imitation to Learning**. In this dynamic environment, agents must acquire and distill practical skills and maintain persistent memory to make decisions based on evolving state variables (e.g., resource availability and time). Critically, these agents are also expected to demonstrate intrinsic motivation by setting their own goals and initiating actions without external prompting. To this end, StuLife provides a comprehensive evaluation platform featuring our novel metrics (e.g., StuGPA) to specifically assess these critical capabilities. Our evaluation reveals that even the best model, GPT-5, scores only 17.9/100, revealing a vast gap toward AGI, demonstrating fundamental deficiencies in retaining long-term memory and acting with self-motivated initiative. Beyond evaluating state-of-the-art LLMs on the StuLife, we also explore the role of context engineering in advancing AGI. Our results suggest that optimizing how we guide models may be as crucial as improving the models themselves, positioning context engineering as a key enabler of progress toward AGI.

## 1 Introduction

Modern machine learning systems have achieved remarkable success in solving well-defined and isolated tasks, such as image classification (He et al., 2016; Dosovitskiy et al.), game playing (Silver et al., 2016; 2018), protein structure prediction (Jumper et al., 2021), or language modeling (Brown et al., 2020; Touvron et al., 2023; DeepSeek-AI, 2025; Yang et al., 2025a). However, these systems typically operate under strong assumptions: they are trained on static datasets, optimized for a single objective, and deployed in environments assumed to remain unchanged. While effective in controlled settings, this paradigm falls short in capturing the essence of real-world intelligence, where environments are dynamic, goals evolve, and new challenges emerge continuously. Life, unlike most machine learning benchmarks, does not present itself as a series of independent tasks with clear labels and fixed endpoints. Instead, it demands constant adaptation, lifelong learning, and the ability to build upon past experiences to navigate an uncertain future.

While continual learning (CL) has advanced in mitigating catastrophic forgetting (Wang et al., 2024a), it primarily emphasizes knowledge preservation in static, task-structured settings, falling short of proactive learning in open-ended environments. This has motivated research on self-evolving agents capable of autonomous improvement (Liang et al., 2025; Chhikara et al., 2025), yet these often lack the capacity for sustained and experience-driven growth. True Artificial General Intelligence (AGI) requires more than incremental adaptation, leading us to posit Experience-Driven Lifelong Learning (ELL) as a potential pathway: the ability to continuously acquire new skills through self-motivated

Figure 1: The StuLife Benchmark for evaluating ELL agents. (a) A schematic of the interaction flow within the StuLife Benchmark. (b) An example of an ELL agent's journey, showcasing how the agent progressively learns from its experiences, leading to tangible knowledge growth over time.

exploration, grounded in long-term memory and shaped by real-world interaction. Despite this need, current benchmarks for lifelong or self-evolving agents remain limited by static designs, narrow domains, and insufficient interactivity (Prabhu et al., 2024; Zhang et al., 2022; Liu et al., 2023; Zheng et al., 2025a), with no comprehensive evaluation platforms targeting the core capabilities of ELL.

To address this gap, we introduce StuLife, a novel benchmark designed to evaluate the core capabilities required for ELL in intelligent agents (Figure 1). StuLife simulates the entire college experience of a student, from enrollment to personal growth, providing a comprehensive benchmark for assessing agents' continuous learning and autonomous decision-making capabilities in complex, dynamic environments. The dataset features three core phases and ten granular sub-scenarios, with the following key characteristics:

**(1) From Simulation to Reality**: The dataset covers pivotal stages of university life, including in-class tasks (e.g., Regulations Learning, Core Course Instruction), Daily Campus Tasks (e.g., Campus Exploration, Initial Course Selection, Preliminary Planning, Academic Activity, Library Resource Management, and Student Club Engagement), and Examination Tasks (e.g., Midterm and Final Exams). This structure faithfully mirrors the trajectory of a real student's academic journey.

**(2) From Imitation to Learning**: Rather than merely retrieving past experiences, agents must abstract generalizable skills from their interactions. They autonomously acquire practical competencies, such as course registration, campus navigation, scheduling, and email communication, through repeated engagement and reflection. This shift emphasizes skill consolidation and transfer, requiring agents to learn how to act, not just what to imitate.

**(3) From Context to Memory**: Tasks are tightly interconnected, both temporally and logically, with knowledge and skills from earlier tasks directly impacting later performance. This necessitates robust long-term memory mechanisms for retaining and retrieving critical experiences, transforming transient context into persistent, actionable knowledge.

**(4) From Passive to Proactive**: Agents are expected to move beyond reactive behavior by developing a sense of time, goal awareness, and intrinsic motivation. They must proactively manage agendas, set personal objectives, anticipate future needs, and adapt to changing conditions, demonstrating initiative and contextual intelligence akin to human learners navigating complex, open-ended environments.

We also introduce **StuGPA**, a unified metric for evaluating long-term agent development, and present the first comprehensive assessment of state-of-the-art LLMs in a realistic, longitudinal learning environment. Even the strongest model, GPT-5, achieves only 17.90 out of 100, revealing a vast gap between current AI and human-level autonomous learning. While context engineering, such as proactive prompting and memory augmentation, can improve performance, agents still fail critically in long-term memory retention and self-motivated behavior. These results highlight the limitations of stateless architectures and underscore that true AGI requires not just better prompts, but fundamentally more capable, memory-grounded, and goal-driven agents.

Table 1: The statistic information of `StuLife`. #Num means the number of samples. #Avg Len and #Max Len mean the average and max number of tokens. #LTM and #Self-Motivation are the number of samples that need long-term memory and self-motivation.

| Core Scenarios | Interconnected Scenarios | #Num | #Avg Len | #Max Len | #LTM | #Self-Motivation |
|---|---|---|---|---|---|---|
| In-Class | Regulations Learning | 70 | 9125 | 9969 | 23 | 70 |
| | Core Course Instruction | 416 | 9203 | 10368 | 129 | 416 |
| | Total | 486 | 9191 | 10368 | 152 | 486 |
| Daily Campus | Campus Exploration | 76 | 2921 | 3006 | 25 | 25 |
| | Initial Course Selection | 150 | 3136 | 3420 | 50 | 0 |
| | Preliminary Planning | 50 | 3069 | 3133 | 50 | 0 |
| | Academic Activity | 72 | 3193 | 3466 | 22 | 22 |
| | Library Study | 151 | 2080 | 3068 | 50 | 50 |
| | Club Activity | 140 | 2981 | 3124 | 45 | 45 |
| | Total | 638 | 2883 | 3466 | 242 | 142 |
| Examination | Midterm Exams | 80 | 3264 | 3520 | 80 | 0 |
| | Final Exams | 80 | 3507 | 3686 | 80 | 0 |
| | Total | 160 | 3386 | 3686 | 160 | 0 |
| Total | Total | 1284 | 5792 | 10368 | 554 | 628 |

## 2 STULIFE BENCHMARK

### 2.1 DATASET DESCRIPTION

We introduce `StuLife`, a comprehensive benchmark for Experience-driven Lifelong Learning (ELL) that simulates a student's academic journey through a dynamically evolving environment (Table 1). The dataset is structured around three core activity modules, including *In-Class Tasks*, *Daily Campus Tasks*, and *Examination Tasks*, designed to evaluate agents' abilities in continuous learning, long-term planning, memory retention, and adaptive decision-making. Spanning a simulated academic term, `StuLife` comprises 1,284 task instances across 10 interconnected scenarios, organized to reflect the natural distribution of student activities in real-world educational settings.

**In-Class Tasks** This module focuses on structured academic learning and foundational knowledge acquisition, encompassing a total of 486 tasks. It includes formal instruction scenarios where agents engage with curricular content, adhere to academic norms, and develop domain-specific understanding. **(1) Regulations Learning**: Agents study the Academic Integrity Guidelines and Student Handbook, answering comprehension questions to internalize academic integrity rules and campus regulations. This forms the basis for compliant and responsible behavior. **(2) Core Course Instruction**: Each course consists of weekly learning episodes (e.g., lectures, readings, and concept checks), totaling 416 in-class interactions with 8 courses. Agents must process textual materials, answer subject-specific questions, and, critically, attend sessions at the correct times and locations. This temporal-spatial requirement evaluates organizational discipline and routine adherence, simulating real-world accountability.

**Daily Campus Tasks** This module captures the diverse, self-directed activities that constitute student life beyond the classroom, comprising 638 tasks in total. These tasks emphasize planning, resource management, social integration, and goal-oriented behavior. **(1) Campus Exploration**: Agents use a digital map tool to locate key facilities (e.g., library, registrar) based on peer-suggested itineraries, developing spatial awareness and environmental familiarity. **(2) Initial Course Selection**: A complex, multi-step task requiring agents to analyze degree requirements, browse course offerings, manage a draft schedule, and strategically use limited "priority cards" to secure preferred classes. This evaluates early-stage decision-making under constraints and goal prioritization. **(3) Preliminary Planning**: Agents check course prerequisite relationships to conduct mandatory course planning and pre-selection for the upcoming semester. This task assesses foresight, knowledge consolidation, and long-term strategic planning. **(4) Academic Activity**: Agents filter advisors by research area or teaching style, initiate contact via email, and complete assigned preparatory tasks, simulating advisor-student collaboration. **(5) Library Resource Management**: Agents perform complex queries to

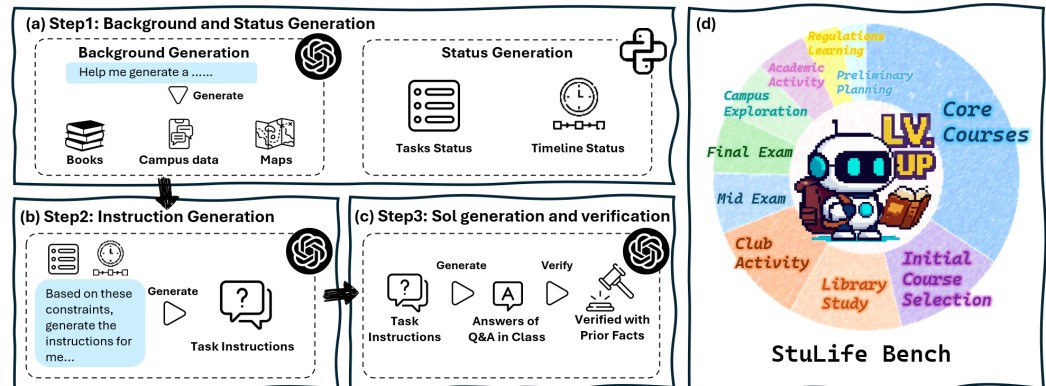

Figure 2: The data generation pipeline for the StuLife Benchmark. The pipeline consists of three sequential steps to create a complete learning instance. (a) Step 1: Background and Status Generation, where the initial context and the agent's current state are established. (b) Step 2: Instruction Generation, where a specific task is formulated based on the generated background. (c) Step 3: Solution Generation and Verification, where a correct solution to the instruction is produced and subsequently verified. (d) A overview of the StuLife Benchmark.

locate books or reserve study spaces under constraints (e.g., noise level, power availability), assessing information retrieval and optimization skills. **(6) Student Club Engagement**: During recruitment events, agents select clubs based on interest tags and complete organizational tasks (e.g., booking rooms, scheduling meetings), promoting responsibility and social integration.

**Examination Tasks**  This module evaluates knowledge retention through 160 examination tasks, including formative and summative assessments that shape subsequent academic trajectories. **(1) Midterm Exams**: Held in Week 10 as "in-class exams", midterms assess partial retention across all enrolled courses (10 questions per subject), requiring accurate recall of first-half content. Agents must physically attend the classroom during class time. Post-exam, targeted learning tasks are assigned for reinforcement. **(2) Final Exams**: Conducted online at semester's end, finals test comprehensive understanding (10 questions per subject) through synthesis and recall of the full curriculum. By removing attendance requirements, the focus shifts to long-term retention, serving as a key measure of memory stability and cumulative learning.

## 2.2 BENCHMARK CONSTRUCTION

To ensure the quality, realism, and logical coherence of StuLife, we implement a multi-stage construction process, including (a) background and status generation, (b) instruction generation, and (c) solution generation and verification, as illustrated in Figure 2.

**Step 1: Background and Status Generation.**  This initial stage establishes the foundational environment and task states. First, we generate a rich campus background using Deepseek-R1, encompassing geographical data (e.g., maps, building layouts, library seating), institutional documents (e.g., handbooks, policies, degree requirements), and essential databases (e.g., course catalogs, faculty profiles, student clubs, library holdings). Second, to ensure strong logical and temporal coherence across tasks, we use deterministic algorithms and local scripts to define task states. This enables precise control over task dependencies and sequencing, particularly for Daily Campus Tasks. For example, campus exploration routes (start, end, waypoints) are generated deterministically from the map to establish verifiable ground truth. Similarly, for In-Class Tasks, we design a structured progression of "key knowledge points" per course, ensuring pedagogical coherence and directly informing exam content.

**Step 2: Instruction Generation.**  In the second stage, we transform the structured, deterministic information from Step 1 into natural language instructions for the agent. Using the Deepseek-R1 model, we convert the pre-defined constraints and objectives into clear, actionable prompts. For Daily Campus Tasks, this involves translating the deterministically generated parameters (e.g., exploration

Table 2: Comparison with existing datasets. Seq, SkilL, SelfMotivat, Interact, and LfE mean Sequentiality, Skill Learning, Self-Motivation, Interactivity, and Learning from Experience.

| Datasets | Task Type | Seq | SkilL | LTM | SelfMotivat | Real | Interconnected | Interact | LfE |
|---|---|---|---|---|---|---|---|---|---|
| Lifelong-CIFAR10 | CL | ✓ | ✗ | ✗ | ✗ | ✗ | ✗ | ✗ | ✗ |
| Lifelong-ImageNet | CL | ✓ | ✗ | ✗ | ✗ | ✗ | ✗ | ✗ | ✗ |
| CGLB | CL | ✓ | ✗ | ✗ | ✗ | ✗ | ✗ | ✗ | ✗ |
| EgoThink | Embodied AI | ✗ | ✗ | ✗ | ✗ | ✓ | ✗ | ✗ | ✗ |
| EmbodiedBench | Embodied AI | ✗ | ✗ | ✗ | ✗ | ✓ | ✗ | ✓ | ✓ |
| AgentBench | Agent | ✗ | ✗ | ✗ | ✗ | ✗ | ✗ | ✓ | ✗ |
| LoCoMo | Agent | ✗ | ✗ | ✓ | ✗ | ✗ | ✗ | ✗ | ✗ |
| StoryBench | Agent | ✓ | ✗ | ✓ | ✗ | ✗ | ✗ | ✓ | ✗ |
| LifelongAgentBench | Self-Evolving | ✓ | ✓ | ✗ | ✗ | ✗ | ✗ | ✓ | ✓ |
| **StuLife** (Our) | ELL | ✓ | ✓ | ✓ | ✓ | ✓ | ✓ | ✓ | ✓ |

waypoints or course selection rules) into natural language. For In-Class and Examination Tasks, this means using the curated "key knowledge points" and "teaching focus" to generate relevant lecture content, in-class questions, and examination questions. This step effectively bridges the gap between the backend logic and the agent's interactive experience, resulting in a complete set of task data for "Daily Campus Tasks" and instructional prompts for In-Class Tasks.

**Step 3: Solution Generation and Verification.** To ensure correctness, especially for tasks requiring long-term knowledge retention, we employ a **generate-and-verify protocol** for In-Class and Examination Tasks. The LLM first generates a correct answer and plausible distractors using lecture and textbook content. We then discard distractors and use LLM to verify the correctness of the intended answer. To validate solvability from long-term memory, we simulate an agent with "optimal memory" by providing only the distilled "key knowledge points" and "teaching focus" from Step 1, and ask the model to answer based solely on this information (detailed in Appendix I). Under these idealized conditions, advanced models demonstrate near-perfect performance across In-Class and Examination tasks, exemplified by GPT-5 achieving a 98.18% total success rate, thereby empirically confirming the inherent solvability and validity of the generated questions. Questions that fail to be answered consistently are manually revised until they meet the criteria. Finally, all data undergoes manual sampling and inspection to ensure quality and consistency across the benchmark.

## 2.3 Comparison with Existing Benchmarks

In this section, we compare our `StuLife` benchmark with the existing benchmarks about continual learning, embodied AI, agent, and self-evolving (Table 2). `StuLife` stands out as a comprehensive benchmark for ELL by addressing key limitations of existing datasets. As shown in Table 2, while benchmarks like Lifelong-CIFAR10 and Lifelong-ImageNet (Prabhu et al., 2024) focus on continuous learning (CL) with sequential task presentation, they lack features such as skill learning, long-term memory, self-motivation, and real-world interactivity. Similarly, CGLB (Zhang et al., 2022) addresses catastrophic forgetting in graph data but does not incorporate realistic, interconnected tasks or experience-driven learning. Embodied AI benchmarks like EgoThink (Cheng et al., 2024) and EmbodiedBench (Yang et al., 2025b) introduce dynamic environments and interactive tasks but still fall short in supporting lifelong learning, skill abstraction, and longitudinal growth. AgentBench (Liu et al., 2023) represents a significant advancement by evaluating LLMs as agents across eight interactive environments, emphasizing multi-turn reasoning and decision-making. However, it is primarily designed for static evaluation of agent capabilities at a fixed point in time, rather than tracking continuous growth or self-evolution. LifelongAgentBench (Zheng et al., 2025a) is the first to target self-evolving agents, offering interdependent tasks across diverse domains. However, it primarily focuses on technical environments (e.g., databases, operating systems) and lacks the rich, evolving personal context and intrinsic motivation that characterize natural learning processes.

`StuLife` is designed as a principled, realistic, and extensible benchmark that advances the evaluation of self-evolving, ELL agents. Unlike conventional benchmarks focused on isolated task performance, `StuLife` integrates environmental realism, self-evolutionary dynamics, and scalable evaluation into a unified framework. Particularly, it simulates a student's college journey, featuring interconnected tasks that evolve over time based on dynamic factors such as advisor selection, course availability, and social interactions. This structure enables agents to learn from experience, retain knowledge across

tasks, transfer skills, and exhibit self-motivated behavior, all essential components of true lifelong learning. Unlike existing datasets, `StuLife` provides a holistic evaluation platform that captures the complexity of real-world intelligence, making it an ideal benchmark for advancing self-evolving AI systems. Below, we highlight its core advantages.

**Environmental Realism.** `StuLife` simulates a full academic term through a longitudinal structure that mirrors the continuous and evolving nature of real-world learning. Tasks unfold in a natural timeline, from enrollment and coursework to extracurriculars and long-term planning, enriched with vivid scenes, character interactions, and contextual dialogues. This sequential and interdependent design ensures that early decisions (e.g., course selection, mentor choice) have lasting consequences on later outcomes (e.g., preliminary planning, Academic Activity), fostering cumulative knowledge acquisition and skill transfer. By modeling both academic and social dimensions of student life, `StuLife` captures the multifaceted challenges of real-world environments, effectively bridging the "sim-to-real" gap. The environment is highly interactive, supporting dynamic agent-object interactions, while promoting open-ended exploration: agents are not constrained by fixed goals but are encouraged to engage in self-directed learning, hallmarks of autonomous intelligence.

**Support for Self-Evolving Intelligence.** As shown in Fig 1, `StuLife` is designed to evaluate self-evolving behavior in intelligent agents through three foundational principles: **1) From Imitation to Learning**: Tasks are explicitly structured around skill acquisition and reuse, focusing on generalizable competencies such as time management, information retrieval, navigation, and social coordination. Rather than merely replicating patterns, agents must abstract experiences into reusable skills and transfer them across evolving challenges, a hallmark of true lifelong learning. **2) From Context to Memory**: The benchmark emphasizes long-term memory and dynamic state evolution, requiring agents to retain and apply knowledge across weeks of simulated academic time. Key contextual variables, such as course eligibility, advisor trust, and scheduling constraints, evolve based on agent decisions, creating a feedback-rich environment that rewards consistency and foresight knowledge retention over time. **3) From Passive to Proactive**: Moving beyond reactive task execution, the narrative-driven design encourages intrinsic motivation and self-directed behavior. Agents must proactively manage deadlines, interpret academic feedback, reflect on past outcomes, and adjust strategies autonomously, mimicking the initiative and adaptive planning seen in real learners.

Together with dedicated metrics for memory utilization, skill acquisition, and cross-task generalization, `StuLife` provides a comprehensive framework for evaluating not just task performance, but the deeper cognitive growth and autonomous development necessary for self-evolving AI. In summary, `StuLife` transcends traditional benchmarks by embedding lifelong learning within a realistic, evolving, and narratively grounded environment. It uniquely supports the development and evaluation of agents that do not merely perform tasks, but learn from experience, grow over time, and act with autonomy, making it a foundational platform for the next generation of self-evolving AI.

## 3 EXPERIMENTAL ANALYSIS

### 3.1 DEFINING EVALUATION METRICS

To provide a multi-faceted assessment of agent intelligence on `StuLife`, we establish a suite of metrics that measure not only task success but also the nuances of learning, knowledge durability, and behavioral efficiency. **1) StuGPA (General Performance Assessment)**: A composite score out of 100 designed to mirror real-world student assessment by holistically measuring academic and personal development beyond simple task accuracy. It is a weighted sum of three components: Exam Performance (50 pts) based on average exam accuracy; Class Performance (30 pts) based on in-class task attendance rate, rewarding routine adherence; and Campus Daily Life (20 pts), which scores advisor/club tasks and a *Personal Responsibility* score that is penalized for infractions like resource squandering or broken commitments. **2) Long-Term Retention Rate (LTRR)**: Evaluates the agent's capacity to combat catastrophic forgetting. It is calculated as the success rate on tasks (e.g., exams) that require recalling and synthesizing information from the distant past. A high LTRR indicates a robust long-term memory system. **3) Proactive Initiative Score (PIS)**: Assesses an agent's self-motivation and prospective memory. It is the success rate on tasks that must be self-initiated based on a prior schedule (e.g., attending a class at the correct time), measuring the ability to autonomously act on temporal commitments. **4) Success Rate & Average Turns**: The primary metric for task completion is the Success Rate, defined as the proportion of tasks where the

Table 3: The performance of existing SOTA LLMs. Icons denote model properties: ⊛ Open-source, ⊛ Thinking model. The LTRR, PIS, and Success Rate, are presented in percentages.

| | StuGPA | LTRR | PIS | In-Class | | Daily Campus | | Exam | | Total | |
|---|---|---|---|---|---|---|---|---|---|---|---|
| | | | | Success | AvgTurn | Success | AvgTurn | Success | AvgTurn | Success | AvgTurn |
| Llama-3.1-8B⊛ | 5.81 | 3.30 | 0.90 | 0.90 | 61.34 | 0.00 | 35.91 | 10.63 | 28.46 | 2.13 | 44.62 |
| Qwen3-8B⊛ | 13.31 | 4.33 | 0.54 | 0.90 | 10.12 | 8.31 | **10.25** | 14.38 | 6.31 | 6.71 | 9.71 |
| Qwen3-30B-A3B⊛⊛ | 16.30 | 5.05 | 0.72 | 0.60 | 9.45 | 10.79 | 11.75 | 17.50 | 5.46 | 8.31 | 10.09 |
| Qwen3-32B⊛ | 7.36 | 3.97 | 0.54 | 0.60 | 7.80 | 2.25 | 13.79 | 13.13 | 4.88 | 3.51 | 10.41 |
| Qwen3-32B⊛⊛ | 12.67 | 5.42 | 1.26 | 1.80 | 8.31 | 7.64 | 10.74 | 17.50 | 4.94 | 7.24 | 9.10 |
| QwQ-32B⊛ | 13.21 | 5.78 | 3.42 | 4.79 | 7.72 | 6.97 | 13.25 | 16.88 | 4.52 | 7.88 | 10.06 |
| DeepSeek-V3⊛ | 11.22 | 6.14 | 2.88 | 3.59 | **5.84** | 6.74 | 11.87 | 16.25 | **4.26** | 7.24 | **8.64** |
| DeepSeek-R1⊛⊛ | 14.25 | 8.30 | 3.96 | 5.09 | 8.04 | 13.26 | 13.02 | 18.13 | 4.56 | 11.18 | 10.08 |
| DeepSeek-V3.1⊛ | 14.26 | 4.51 | 0.54 | 0.90 | 14.03 | 12.81 | 12.62 | 15.00 | 6.78 | 8.95 | 12.43 |
| DeepSeek-V3.1⊛⊛ | 17.04 | 6.14 | 3.78 | 6.29 | 9.83 | 12.58 | 13.03 | 17.50 | 5.54 | 11.18 | 10.88 |
| Qwen3-235B-A22B⊛ | 16.03 | 5.42 | 1.80 | 2.10 | 18.71 | 10.34 | 17.17 | 16.88 | 10.75 | 8.52 | 16.95 |
| Gemini-2.5-Pro⊛⊛ | 16.43 | 7.04 | 3.24 | 5.39 | 14.94 | 18.88 | 12.78 | 15.63 | 9.51 | 13.53 | 13.19 |
| Grok4⊛ | 17.38 | **10.65** | 4.50 | 4.79 | 6.31 | **21.80** | 11.25 | **18.75** | 5.69 | **15.23** | 8.68 |
| GPT-5⊛ | **17.90** | 6.50 | **4.68** | **7.78** | 12.70 | 14.16 | 14.31 | 16.88 | 6.24 | 12.35 | 12.69 |
| Human | 85.24 | 84.91 | 88.13 | 88.62 | - | 91.13 | - | 82.50 | - | 88.81 | - |

final objective is successfully achieved. We also measure interaction efficiency using Average Turns (AvgTurns), which calculates the average number of turns required for successful task completion. A lower AvgTurns value signifies higher efficiency and more sophisticated problem-solving.

## 3.2 EVALUATING THE EXISTING SOTA LLMS

**Experimental Setup.** We assess a diverse set of ten prominent Large Language Models (LLMs) as the cognitive engines for our intelligent agents. The models under study include: Llama-3.1-8B[1], Qwen3-7B (Yang et al., 2025a), Qwen3-32B (Yang et al., 2025a), QWQ-32B (Yang et al., 2024), Deepseek-V3 (Liu et al., 2024), Qwen3-235B (Yang et al., 2025a), GPT-5[2], Gemini 2.5 Pro[3], and Grok-4[4]. These models represent the current frontier in understanding, reasoning, and agentic capabilities. The StuLife benchmark simulates a continuous and stateful trajectory where tasks unfold sequentially and actions in one task can have lasting effects on later ones. In our main evaluation setup, each task is presented to the agent as an isolated instance. The agent starts with no access to prior trajectory, any retention of cross-task information must be explicitly managed by the agent itself, using available environmental tools like the calendar or draft system. Additionally, we recruit several undergraduate and graduate students to establish a human baseline for comparison.

**RQ I**: How do current LLMs perform on long-horizon agency tasks, and how does performance vary with model scale and type?

**Finding 1**: Scaling up parameters isn't enough—what AI really needs is to grow up, not just level up. The path to AGI may not lie in bigger models, but in building agents that learn how to learn, not just answer questions.

As shown in Table 3, the performance of all models under our default stateless setting is exceptionally poor. This stems from a fundamental architectural limitation: current large language models are inherently stateless and do not possess any native long-term memory modules. This core deficiency manifests in two critical failures observed throughout our experiments.

**First,** our results reveal that all current agents, regardless of whether they are open-source or proprietary, exhibit mediocre performance and are far from achieving human-like competency in long-horizon tasks. As shown in Table 3, even the most advanced model, GPT-5, achieves a StuGPA of only 17.90 out of 100, underscoring the significant gap between current SOTA models and the general, autonomous intelligence required for such challenges. When contrasted with the human StuGPA of

---

[1] https://ai.meta.com/research/publications/the-llama-3-herd-of-models/
[2] https://openai.com/index/introducing-gpt-5/
[3] https://deepmind.google/models/gemini/pro/
[4] https://x.ai/news/grok-4

85.24, this gap becomes even more glaring, illustrating that current LLMs have yet to bridge the vast divide separating statistical pattern matching from human-level functional competence. **Second**, there is a strong positive correlation with **model scale**. A significant performance chasm exists between small models like `Llama-3.1-8B-Instruct` (StuGPA of 5.81) and the best-performing large-scale models. The progression from 8B models to 32B, 235B, and proprietary models generally yields higher scores, indicating that larger parameter counts do enhance the complex reasoning required for these tasks. **Third**, and more revealingly, is the impact of **model type**. A direct comparison within the same size class shows that models designed for step-by-step reasoning significantly outperform their standard instruction-following counterparts. For example, `Qwen3-32B-Thinking` achieves a StuGPA of 12.67, substantially higher than the 7.36 scored by `Qwen3-32B-Instruct`. The "Thinking" model also demonstrates better proactivity, with a Proactive Initiative Score (PIS) of 1.26 compared to the Instruct model's 0.54. This suggests that for complex agency tasks that require planning and foresight, a model's internal architecture designed to support deliberative reasoning is a more critical determinant of success than its raw instruction-following ability alone.

**RQ II**: Can stateless LLMs manage long-term memory and proactive behavior?

**Finding 2**: No memory by design, no motivation by default: stateless LLMs cannot grow through experience.

The evidence for this finding, detailed in Table 3, is stark and can be broken down into two key areas of failure. **First,** proactive, self-motivated behavior is arguably the greatest challenge for current agents. This is demonstrated by a common failure mode: for a task scheduled at 8:00, even when an agent is notified that the current time is precisely 8:00, it often fails to initiate the corresponding action. This issue stems from a more fundamental problem where the agent does not even think to consult its schedule, regardless of whether the event was successfully added previously. This qualitative failure is directly quantified by the Proactive Initiative Scores (PIS), which are critically low across the board. Even the top-performing model, `GPT-5`, only achieves a PIS of 4.68%, while many powerful models like `Qwen3-8B` and `DeepSeek-V3.1-chat` score below 1%. While a reasoning-focused "Thinking" architecture offers a slight advantage (`Qwen3-32B-Thinking` PIS of 1.26% vs. its "Instruct" counterpart's 0.54%), it does not overcome this fundamental lack of self-motivation. This stands in sharp relief against the human PIS of 88.13%, revealing that while models may possess vast knowledge, they completely lack the intrinsic drive and autonomy that define human agency. **Second,** the widespread failure in long-term retention is a direct and expected consequence of the fact that **all current LLMs are architecturally stateless**. In this context, the LTRR metric does not measure an LLM's intrinsic memory. Instead, it evaluates the success rate of a fragile, multi-step, tool-dependent process: the agent must first recognize critical information, then correctly use a tool to externalize it (e.g., adding an event to the calendar), and finally, think to retrieve it for a future exam. Therefore, the slightly higher LTRR scores from top models like `Grok4` (10.65%) do not indicate superior intrinsic memory. Rather, they reflect a higher probability of having made correct tool-use decisions in prerequisite tasks. The fact that even the best models fail 90% of the time on this metric underscores the brittleness of this process and confirms that reliable knowledge accumulation is currently unattainable without a dedicated memory system. The profound gap with the human LTRR of 84.91% indicates that current architectures lack the cognitive continuity required to preserve critical information over time, a fundamental trait of human intelligence that stateless models fail to replicate.

**RQ III**: To what extent can existing self-evolving mechanisms address the challenges of lifelong learning?

**Finding 3**: Self-evolving mechanisms provide tangible benefits in both learnability and efficiency. However, these gains remain bounded, as existing strategies have yet to bridge the gap required for proactive initiative and sustained long-term memory retention.

To demonstrate the flexibility of our benchmark in evaluating experience-driven improvement, we investigate two distinct categories of self-evolving strategies across different model scales. For training-based evolution, we employ an on-policy Rejection Sampling Fine-Tuning (RFT) (Yuan et al., 2023) on the Qwen3-8B model, where the model generates eight rollouts per prompt during exploration and is subsequently fine-tuned using only the successful trajectories. For inference-time evolution, we evaluate Reflexion (Shinn et al., 2023) and Agent Workflow Memory (AWM) (Wang

Table 4: Performance of Self-Evolving mechanisms across different model scales. The 8B model utilizes training-based evolution (RFT), while the 235B model utilizes inference-time evolution (AWM, Reflexion).

| | StuGPA | LTRR | PIS | In-Class | | Daily Campus | | Exam | | Total | |
|---|---|---|---|---|---|---|---|---|---|---|---|
| | | | | Success | AvgTurn | Success | AvgTurn | Success | AvgTurn | Success | AvgTurn |
| **Qwen3-8B** | | | | | | | | | | | |
| Vanilla | 13.31 | 4.33 | 0.54 | 0.90 | **10.12** | 8.31 | 10.25 | **14.38** | 6.31 | 6.71 | **9.71** |
| + RFT | **15.43** | **4.51** | **0.90** | **1.50** | 12.08 | **11.91** | 9.43 | **14.38** | **5.05** | 8.63 | 9.89 |
| **Qwen3-235B-A22B** | | | | | | | | | | | |
| Vanilla | 16.03 | 5.42 | 1.80 | 2.10 | 18.71 | 10.34 | 17.17 | **16.88** | 10.75 | 8.52 | 16.95 |
| + AWM | **17.81** | **6.87** | 1.80 | 1.50 | **15.66** | **14.38** | 14.23 | 16.25 | **7.68** | **10.12** | **13.96** |
| + Reflexion | 16.18 | 5.60 | 1.44 | 0.90 | **15.66** | 12.13 | 15.05 | **16.88** | 9.08 | 8.95 | 14.54 |

et al., 2024b) on the larger Qwen3-235B-A22B model. These methods leverage context and historical feedback to dynamically optimize future actions without updating model weights.

**Analysis of Self-Evolving Approaches.** The results in Table 4 highlight the distinct advantages offered by different evolutionary paths. For the smaller model, RFT demonstrates strong data efficiency and learnability, significantly boosting Qwen3-8B's StuGPA from 13.31 to 15.43 and Total Success from 6.71% to 8.63%. This improvement confirms that the complex decision-making logic within our benchmark follows learnable patterns that can be effectively distilled into smaller models through high-quality experience. For the larger model, inference-time strategies also yield performance gains. AWM achieves a robust improvement, lifting StuGPA to 17.81 and Total Success to 10.12%, while Reflexion shows marginal gains. Crucially, AWM stands out by simultaneously optimizing efficiency, reducing the Total AvgTurn from 16.95 to 13.96. This suggests that retrieving successful workflows not only aids in task completion but also helps the agent avoid redundant steps. Nevertheless, despite these improvements across both paradigms, the gains remain modest relative to the complexity of the lifelong learning setting. This validates the difficulty of our benchmark: while agents can evolve to become more accurate or efficient, existing mechanisms have yet to bridge the gap required for proactive initiative and sustained cognitive coherence.

**RQ IV**: Can Context Engineering mitigate the LLMs' limitations in lifelong learning scenarios?

**Finding 4**: Context Engineering isn't just programming—it's propping up intelligence, a critical pathway toward more general and autonomous intelligence.

We also make a critical contribution by systematically investigating the impact of *context engineering*, on agent performance in complex and long-horizon tasks. While much of current AI development focuses on scaling models or improving weights through training, our results demonstrate that *optimizing the context prompt* can also improve performance effectively. Particularly, we design five distinct prompting strategies to investigate different dimensions of agent intelligence in a simulated academic environment: (1) Vanilla Prompt: A minimal baseline assigning only a high-level role, measuring the agent's intrinsic reasoning without procedural or cognitive support. (2) Proactive Prompt: Adds structured guidance on time-awareness, goal decomposition, and forward planning to emulate strategic, student-like behavior. (3) Skill-Augmented Prompt: Provides step-by-step "recipes" for specific skills (e.g., problem-solving), enabling structured and generalizable task decomposition. (4) Memory-Augmented Prompt: Integrates external memory mechanisms (e.g., Vanilla RAG Lewis et al. (2020), GraphRAG (Edge et al., 2024), MemoryBank (Zhong et al., 2024), MemGPT (Packer et al., 2023)) to simulate cumulative learning and assess long-term knowledge retention. (5) All-in-One Prompt: Unifies proactive planning, skill execution, and memory augmentation into a comprehensive cognitive framework, aiming to evaluate the upper bound of self-evolving intelligence.

**Analysis of Proactive and Skill Prompts.** Explicit cognitive prompts delivered targeted gains. The **Proactive Prompt**, which instills time management principles, sharply improved time-sensitive performance: In-Class success more than doubled (2.10% to 5.09%), and PIS rose significantly (1.80% to 3.06%). Yet, because it doesn't teach *how* to break down or execute complex tasks and still depends on fragile tool-based memory, the overall StuGPA gain was modest (16.03 to 16.90). In

Table 5: The performance of various Context Engineering methods based on Qwen3-235B-A22B😩.

| | | StuGPA | LTRR | PIS | In-Class | | Daily Campus | | Exam | | Total | |
|---|---|---|---|---|---|---|---|---|---|---|---|---|
| | | | | | Success | AvgTurn | Success | AvgTurn | Success | AvgTurn | Success | AvgTurn |
| **Vanilla** | | 16.03 | 5.42 | 1.80 | 2.10 | 18.71 | 10.34 | 17.17 | 16.88 | 10.75 | 8.52 | 16.95 |
| **Proactive** | | 16.90 | 5.96 | 3.06 | **5.09** | **16.70** | 10.34 | 16.38 | 16.88 | 7.73 | 9.58 | 15.42 |
| **Skill** | | 17.28 | 6.86 | 0.90 | 1.50 | 16.89 | 15.28 | 16.51 | 17.50 | 9.28 | 10.76 | 15.75 |
| **Memory** | + Vanilla RAG | 10.98 | 4.69 | 0.18 | 0.00 | 17.87 | 5.84 | 14.20 | 16.25 | 10.04 | 5.54 | 15.07 |
| | + Graph RAG | 15.34 | 4.87 | 0.72 | 0.90 | 20.68 | 10.11 | 14.03 | 16.25 | 10.61 | 7.88 | 16.13 |
| | + MemGPT | 19.99 | 6.86 | 1.44 | 2.40 | 17.28 | 13.03 | **13.59** | 23.75 | 9.02 | 11.08 | 14.42 |
| | + MemoryBank | 17.64 | 5.96 | 1.62 | 0.90 | 16.68 | 12.36 | 14.15 | 20.00 | 8.04 | 9.58 | **14.35** |
| **All-in-One** | | **21.07** | **9.39** | **3.76** | 2.69 | 16.82 | **17.75** | 15.65 | **25.63** | **6.30** | **13.74** | 14.93 |

contrast, the **Skill-Augmented Prompt**, which supplies a step-by-step problem-solving framework, greatly increased success on complex Daily Campus tasks (10.34% to 15.28%). But by focusing solely on execution, it neglects initiative, causing PIS to plummet (1.80% to 0.90%).

**Analysis of Memory-Augmented Approaches.** Adding external memory produced sharply different outcomes, highlighting that memory system design is critical. Naive RAG methods (e.g., **Vanilla RAG** and **Graph RAG**), which retrieve raw trajectories via vector similarity and inject them directly into context, actively harmed performance. They worsened the LLM's long-context degradation by introducing unfiltered noise, collapsing scores (e.g., Vanilla RAG StuGPA: 10.98) and even reducing LTRR below baseline (5.42% to 4.69%). In contrast, **MemGPT**, a sophisticated memory system that structures and manages stored information, achieved the highest overall StuGPA (19.99). Its reliable recall improved Examination success (23.75%). Yet, despite its strong reactive memory, MemGPT cannot compensate for the agent's lack of time awareness.

**The Power of the All-in-One Approach.** Our analysis shows each strategy tackles a unique bottleneck: Proactive prompts teach "when," Skill prompts teach "how," and Memory systems supply the "what." The results show that **All-in-One Prompt** outperforms all others, achieving the highest StuGPA (21.07, surpassing GPT-5), PIS (3.76%) and Daily Campus success (17.75%). The LTRR soars to 9.39%, proving that combining memory with proactivity enables the agent to both remember and act on long-term goals. This confirms our core thesis: true autonomous agency emerges not from isolated components, but from the integrated synergy of proactivity, skill, and structured memory.

## 4 CONCLUSIONS

In this paper, we present `StuLife` for Experience-driven Lifelong Learning (ELL), a comprehensive benchmark that simulates a student's college journey as a rich and longitudinal environment with interconnected academic, social, and planning tasks. `StuLife` goes beyond conventional benchmarks by incorporating evolving contexts, sparse rewards, and the need for self-initiated behavior, making it a powerful platform for evaluating lifelong learning, memory retention, and autonomous decision-making. Our experiments reveal critical limitations in current LLM-based agents, particularly their lack of self-motivation and poor long-term memory, highlighting the gap between today's systems and truly self-evolving intelligence. Importantly, we demonstrate that performance on complex, extended tasks can be significantly improved not only through model scale but through *context engineering*, particularly via well-designed system prompts. This suggests that the path to AGI may depend as much on how we structure an agent's cognitive framework as on the raw capabilities of the underlying model. Looking forward, `StuLife` provides a principled foundation for developing agents that accumulate knowledge, transfer skills, and evolve autonomously over time.

### REPRODUCIBILITY STATEMENT

We are committed to ensuring the reproducibility of our research. The complete source code for our experimental framework, along with the full `StuLife` benchmark dataset, is included in the Supplementary Materials. Detailed descriptions of our experiments and the complete text of all system prompts used for context engineering and data generation are provided in the Appendix. We

believe these materials provide all the necessary components to replicate our findings and facilitate future research in experience-driven lifelong learning.

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

## USE OF LARGE LANGUAGE MODELS

During the preparation of this manuscript, we utilized a Large Language Model (LLM), specifically [e.g., OpenAI's GPT-4 or Google's Gemini], as a general-purpose writing assistance tool.

The use of the LLM was strictly confined to improving the language, polishing sentences, and checking for grammatical errors to enhance the overall clarity and readability of the text. We emphasize that the LLM did not contribute to any of the core research aspects of this work. This includes, but is not limited to, the generation of research ideas, experimental design, code generation, data analysis, interpretation of results, or the formulation of conclusions.

All suggestions and modifications proposed by the LLM were carefully reviewed, critically evaluated, and revised by the human authors to ensure their accuracy and alignment with our scientific intent. The authors are fully responsible for all the scientific contributions, original content, and final presentation of this paper.

## A    RELATED WORK

### A.1    CONTINUAL LEARNING

Continual learning (CL), also known as lifelong learning, aims to enable machine learning models to learn sequentially from a stream of data or tasks while mitigating catastrophic forgetting, the tendency to overwrite previously acquired knowledge (Wang et al., 2024a). A significant body of research has developed techniques to address this stability-plasticity dilemma, including regularization methods (Kirkpatrick et al., 2017), architectural modifications (Huai et al., 2025a;b; Ding et al., 2024), and replay-based strategies (Rolnick et al., 2019). Recent surveys have cataloged these approaches across various domains, including deep networks (Hadsell et al., 2020), large language models (LLMs) (Yang et al., 2025c), and generative models (Guo et al., 2025). To address the varied complexities of real-world learning, CL research defines several experimental settings, including Task-Incremental Learning (TIL), Domain-Incremental Learning (DIL) and Class-Incremental Learning (CIL) (Van de Ven et al., 2022). However, many existing CL paradigms operate under constrained assumptions, such as the presence of predefined task boundaries and access to supervised or semi-supervised signals. They often rely on static datasets or controlled data streams, which limits their applicability to dynamic, real-world environments where task boundaries are ambiguous and data arrives continuously and autonomously (Shaheen et al., 2022). Furthermore, while effective for retaining performance on past tasks, traditional CL primarily focuses on knowledge preservation rather than proactive exploration and knowledge acquisition.

### A.2    SELF-EVOLVING AGENT

The pursuit of Artificial General Intelligence (AGI) has spurred interest in self-evolving agents—systems capable of autonomous, open-ended growth and adaptation (Zheng et al., 2025b; Gao et al., 2025). These agents aim to move beyond static models by continuously learning from experience, refining their skills, and potentially modifying their own architectures or goals. Research in this area explores mechanisms for self-improvement, including reflective reasoning, memory augmentation from past interactions (Liang et al., 2025). Effective memory systems are critical for self-evolution, enabling agents to store, retrieve, and optimize experiences (Chhikara et al., 2025; Yu et al., 2025; Zhong et al., 2024). Some frameworks demonstrate self-evolution through mechanisms like self-play, generating novel experiences without specific human-provided data (Guan et al., 2024). While surveys highlight the potential of integrating continual learning with other cognitive components like memory and reasoning to create self-evolving systems, current implementations often remain theoretical or focused on narrow applications. A key challenge lies in developing comprehensive frameworks that integrate robust memory mechanisms, facilitate autonomous skill acquisition, and support long-term, goal-directed behavior necessary for truly autonomous agents with self-motivation in complex, real-world scenarios.

Figure 3: An overview of the Experience-driven Lifelong Learning (ELL) framework. ELL is a continuous learning cycle where an agent evolves through direct interaction with its environment. (a) The core loop of ELL: The agent interacts with the current knowledge to acquire trajectories. This experience is processed through Knowledge Abstraction and Refinement, and the resulting knowledge is validated. (b) Knowledge Abstraction converts raw experience into a structured knowledge base composed of Memory and Skills, which forms the foundation for all future learning and action. (c) Knowledge Refinement ensures the knowledge base remains optimal and up-to-date by dynamically performing four key operations: Add, Update, Delete, or Combine.

## A.3 BENCHMARKS FOR SELF-EVOLVING AND LIFELONG AGENTS

Evaluating the capabilities of self-evolving agents requires benchmarks that move beyond traditional static datasets. Existing benchmarks for **continual learning**, while valuable for studying catastrophic forgetting on sequential tasks (e.g., Lifelong-CIFAR10, CGLB), typically lack the rich interactivity and dynamic environments necessary for agentic learning (Prabhu et al., 2024; Zhang et al., 2022). Conversely, **embodied AI** benchmarks (e.g., EgoThink) offer interactive environments but are not designed to assess long-term, cumulative knowledge acquisition or skill abstraction over an agent's lifetime (Cheng et al., 2024). More recent developments like **AgentBench** have established robust protocols for evaluating the reasoning and tool-use capabilities of LLM-based agents, but they primarily focus on static, one-off task performance rather than tracking an agent's continuous growth and self-evolution over time (Liu et al., 2023). While **LifelongAgentBench** was the first to target self-evolving agents with interdependent tasks, its focus on technical domains (e.g., databases, operating systems) leaves a gap for benchmarks that model the complex, narrative-driven, and intrinsically motivated nature of human-like learning (Zheng et al., 2025a). This landscape highlights the need for a benchmark that integrates a realistic, evolving personal context with the core principles of lifelong learning and self-motivated behavior, a gap that our StuLife benchmark aims to fill.

## B FORMAL DEFINITIONS OF EXPERIENCE-DRIVEN LIFELONG LEARNING

In this paper, we introduce a formal and mathematically grounded framework for ELL, where agents learn continuously from sequential task interactions. By shifting from conceptual vision to a concrete framework, ELL establishes a foundation for building agents that evolve through real-world experience. At its core, ELL is driven by a rigorous continual experiential learning mechanism, centered on:

- **Experience Exploration**: The agent must be capable of sequentially decomposing and executing complex, long-horizon tasks that involve **continuous interaction over minutes to hours with unquantifiable rewards**. Through sustained and **self-motivated** engagement, it generates rich experiential data, enabling iterative learning and self-correction. This persistent interaction allows the

agent to progressively refine strategies and adapt behavior based on dynamic feedback, mimicking the trial-and-error process of real-world learning.

- **Long-term Memory**: Experiential data is systematically processed and consolidated into persistent and structured memory, including raw observations, key events, learned facts, temporal contexts, and self-reflective insights. Memory is not passive storage but an active resource: it supports retrieval over long time spans, enables context-aware reasoning, and forms the foundation for future decision-making.

- **Skill Learning**: The agent **abstracts recurring patterns from experience into reusable skills**, such as decision rules, functional modules, or problem-solving heuristics. These skills are explicitly constructed through reflection and validated through application in new and evolving tasks. The agent actively manages its skill repertoire, adding, refining, combining, or deprecating skills based on performance, creating a dynamic, self-improving system.

- **Knowledge Internalization**: Beyond storing memories and reusing skills, the agent undergoes a process of **knowledge internalization**, transforming explicit and discrete knowledge into implicit and intuitive understanding. Over time, frequently used rules, patterns, and strategies are distilled into the agent's core reasoning process, reducing reliance on external retrieval or step-by-step reflection. This shift from deliberate application to automatic execution mirrors the cognitive transition from novice to expert, where learned behavior becomes "second nature."

Building a formal framework for experience-driven lifelong learning requires first establishing the foundational concepts that model an agent's interaction with its world. We conceptualize an **agent** (Definition 5) operating within an **environment** ($\mathcal{E}$, Definition 1), which we model as a Partially Observable Markov Decision Process (POMDP) (Kaelbling et al., 1998), to accomplish a given sequence of **tasks** ($\mathcal{T}$, Definition 2). Through its sequential interactions, the agent generates a **trajectory** ($\xi$, Definition 3), which encapsulates the observations, actions, and outcomes of its endeavors. This raw trajectory is not merely stored but is progressively distilled and structured into a comprehensive **knowledge** ($\mathcal{K}$, Definition 4). This knowledge, comprising both memory and skills, forms the foundation upon which the agent adapts its policy, improves its performance, and ultimately achieves lifelong learning. The following subsections will provide rigorous definitions for each of these fundamental concepts.

### B.1 FORMAL DEFINITIONS OF FUNDAMENTAL CONCEPTS

**Definition 1** (Environment). We model the agent's environment $\mathcal{E}$ as a goal-conditional partially observable Markov decision process (POMDP), defined by the 8-tuple:

$$\mathcal{E} = (\mathcal{S}, \mathcal{A}, \mathcal{G}, T, R, \Omega, O, \gamma), \tag{1}$$

where:

- $\mathcal{S}$ **(State Space)**: The set of all possible states. Each state $s \in \mathcal{S}$ can contain multimodal information, such as textual descriptions, images, or structured data.

- $\mathcal{A}$ **(Action Space)**: The set of all actions the agent can perform. Each action $a \in \mathcal{A}$ is often a natural language command, e.g., "add this item to the cart".

- $\mathcal{G}$ **(Goal Space)**: The set of all possible goals. Each goal $g \in \mathcal{G}$ defines a specific task for the agent to complete, e.g., "purchase a laptop".

- $T(s' \mid s, a)$ **(State Transition Function)**: Defines the probability distribution over the next state $s'$ after taking action $a$ in state $s$.

- $R(s, a, g)$ **(Goal-Conditional Reward Function)**: Evaluates how well action $a$ taken in state $s$ contributes to achieving goal $g$, returning either a numeric score or textual feedback.

- $\Omega$ **(Observation Space)**: The set of all possible observations. An observation $o \in \Omega$ represents the agent's partial perception of the current state, which can be textual, visual, or a combination.

- $O(o' \mid s', a)$ **(Observation Probability Function)**: Defines the probability of receiving a specific observation $o'$ after action $a$ leads to a new state $s'$.

- $\gamma$ **(Discount Factor)**: A value in $[0, 1)$ that balances the importance of immediate versus long-term rewards, typically used only when rewards are numeric.

**Definition 2** (Task). The agent's lifelong learning journey involves tackling a sequence of $N$ complex real-world tasks, $\{\mathcal{T}^{(1)}, \mathcal{T}^{(2)}, \mathcal{T}^{(3)}, \ldots, \mathcal{T}^{(N)}\}$. A task $\mathcal{T}^{(i)}$ is defined by an environment $\mathcal{E}^{(i)}$, an initial observation $o_0^{(i)}$, and a goal $g^{(i)}$.

$$\mathcal{T}^{(i)} = \langle \mathcal{E}^{(i)}, o_0^{(i)}, g^{(i)} \rangle \tag{2}$$

**Definition 3** (Trajectory). The agent's interaction with the environment to solve a task generates a trajectory $\xi$, which is a sequence of observations, actions, and rewards:

$$\xi = \langle o_0, a_0, r_0, o_1, a_1, r_1, \ldots, o_T, a_T, r_T \rangle \tag{3}$$

**Definition 4** (Knowledge). A Lifelong Learning Agent possesses a dynamic **Knowledge**, $\mathcal{K}$, which is composed of two primary components: Memory ($\mathcal{M}$) and a set of Skills ($\mathcal{F}$).

$$\mathcal{K} = (\mathcal{M}, \mathcal{F}) \tag{4}$$

This knowledge represents the entirety of what the agent has learned and is the foundation for all future learning.

- **Memory** ($\mathcal{M}$) is a structured repository of information. An individual memory item stored within $\mathcal{M}$ may take one of the following forms:
  - **Trajectory Memory ($\mathcal{M}_{\text{traj}}$)**: Raw or summarized trajectories, $\xi$.
  - **Declarative Knowledge (Object Facts, $\mathcal{M}_{\text{decl}}$)**: Represents factual and conceptual "what" knowledge, providing a foundation of information (e.g., facts, concepts, beliefs).
  - **Structural Knowledge (Relationships between Object, Concept, $\mathcal{M}_{\text{struct}}$)**: Defines relationships between concepts and objects, often represented in semantic networks or knowledge graphs, which aid in understanding complex relationships and problem-solving.
- **Skills** ($\mathcal{F}$) represent procedural knowledge of how to perform actions or solve problems. An individual skill within $\mathcal{F}$ may take one of the following forms:
  - **Procedural Knowledge (Rules Procedural, $\mathcal{F}_{\text{proce}}$)**: Encapsulates "how-to" knowledge, skills, and strategies for accomplishing specific activities (e.g., rules, sequences of actions).
  - **Meta-Knowledge (Knowledge about Knowledge, $\mathcal{F}_{\text{meta}}$)**: Knowledge about knowledge itself, including learning processes, categories, and plans, which enables an agent to understand and manage its own learning.
  - **Heuristic Knowledge (Rules of Thumb, $\mathcal{F}_{\text{heur}}$)**: Refers to rules of thumb, approximations, and experience-based decision-making strategies (shortcuts) that guide problem-solving in complex situations.

## B.2 DEFINITION OF THE LIFELONG LEARNING AGENT

We now extend the previous framework to define a Lifelong Learning Agent that actively engages with the world, sequentially undertaking tasks and evolving its internal knowledge through iterative self-correction.

**Definition 5** (Lifelong Agent). An agent utilizes a policy $\pi$ based on the knowledge $\mathcal{K}$ to interact with an environment $\mathcal{E}$. The policy maps an observation $o_t \in \Omega$ to an action $a_t \in \mathcal{A}$:

$$a_t = \pi(o_t; \mathcal{K}) \tag{5}$$

Self-evolving AI agents typically comprise four essential, interacting components:

- **Perception**: This module is the physical (for physical agents like robots) and logical implementation (for software agents interacting with APIs or databases) of the observation process. It is responsible for receiving information from the environment and generating an observation $o \in \Omega$ according to the probability function $O(o \mid s', a)$.

- **Memory**: The memory module serves as the agent's repository for storing and managing knowledge $\mathcal{K}$ acquired through perception, learning, and reasoning. It is typically divided into two types: Short-Term Memory (STM, also known as Working Memory) and Long-Term Memory (LTM, also referred to as Episodic Memory). Short-term memory holds immediate observations and contextual information required for on-the-fly decision-making, allowing the agent to maintain awareness of its current state and interactions. Long-term memory, on the other hand, retains distilled experiences, learned skills, and structured knowledge over extended periods. Its purpose is to enable agents to utilize accumulated past experiences and knowledge to inform current tasks, decision-making, and overall behavior over extended periods.

  Both memory systems store two core types of information: memory (episodic and semantic knowledge) and skills (procedural and strategic capabilities). Specifically, memory includes Trajectory Memory ($\mathcal{M}_{traj}$) for raw or summarized interaction histories, Declarative Knowledge ($\mathcal{M}_{decl}$) for factual "what" knowledge (e.g., course requirements), and Structural Knowledge ($\mathcal{M}_{struct}$) for representing relationships between concepts (e.g., prerequisite dependencies). Skills, in turn, encompass Procedural Knowledge ($\mathcal{F}_{proce}$) for action sequences (e.g., how to register for a course), Meta-Knowledge ($\mathcal{F}_{meta}$) for self-regulated learning and planning, and Heuristic Knowledge ($\mathcal{F}_{heur}$) for experience-based decision rules (e.g., prioritizing high-impact tasks). The memory module supports dynamic operations, such as adding new entries, deleting outdated information, merging similar memories, or consolidating skills, enabling the agent to adapt its knowledge base continuously, avoid redundancy, and maintain coherence across evolving tasks and environments.

- **Learning**: This is the critical component that enables self-evolution. Learning agents continuously improve their performance based on their experiences and the feedback they receive from the environment. This involves adapting strategies and refining internal models over time. We need a meta-cognitive learning architecture that enables agents to learn from multiple task trajectories by explicitly reflecting on successes and failures, extracting actionable lessons, and integrating them into future behavior, either via in-context learning or knowledge distillation. The framework supports explicit, interpretable, and cumulative knowledge acquisition, bridging the gap between trial-and-error learning and human-like reflective improvement.

  Given a target task, an agent first performs multiple trajectories to explore different behavioral policies. Each trajectory contains a complete state-action-reward trajectory and records corresponding environmental feedback, including both immediate rewards and final outcomes. Subsequently, all trajectory processes, encompassing observation sequences, actions taken, intermediate decision rationales, and associated reward signals, are aggregated into a unified context and fed into a reflection module equipped with meta-cognitive capabilities. This module is guided by a **meta-prompt** to conduct structured retrospective analysis, such as:

  > "Among these attempts, which strategies led to higher cumulative rewards?
  > Which actions resulted in failure or suboptimal outcomes?
  > Are there any generalizable patterns?
  > What adjustments should be attempted next?"

  These lessons are then explicitly appended to the system prompt as guiding knowledge for future tasks or, more generally, stored in a retrievable *dynamic lesson repository*, enabling context augmentation or knowledge distillation in subsequent tasks. This mechanism enables **explicit knowledge accumulation**, emulating the human practice of "learning from experience to guide future behavior." Furthermore, the framework can be integrated with model fine-tuning or parametric knowledge distillation. Once a sufficient number of high-quality lessons have been accumulated, they can be used for supervised fine-tuning, transforming explicit rules into **intuitive model behaviors**, analogous to how humans internalize deliberate strategies into automated skills through repeated practice. This architecture, where knowledge is first acquired explicitly and later optionally internalized, mirrors cognitive theories of skill acquisition and offers a promising direction for building adaptive, self-improving AI systems.

- **Reasoning**: Acting as the "brain" of the operation, the reasoning module processes perceived information and makes decisions to infer patterns, predict outcomes, and select appropriate actions.

- **Action**: The action module is responsible for executing the responses or behaviors determined by the reasoning component. It executes the action $a_t \in \mathcal{A}$ selected by the policy $\pi(o_t | \mathcal{K}_t)$, thereby interacting with the environment and invoking the state transition $T(s' \mid s, a_t)$.

**Definition 6** (The Lifelong Learning Process). The Lifelong Learning Agent operates over a sequence of tasks, $\mathcal{T} = \{\mathcal{T}^{(1)}, \mathcal{T}^{(2)}, \ldots, \mathcal{T}^{(N)}\}$. The process is sequential, where the final knowledge base from task $\mathcal{T}^{(i)}$ becomes the initial knowledge base for task $\mathcal{T}^{(i+1)}$. This core loop for any given task involves interaction, refinement, and validation.

Let $\mathcal{K}_t$ be the agent's knowledge base at time step $t$. The agent's policy is now explicitly conditioned on its knowledge:

$$a_t = \pi(o_t | \mathcal{K}_t) \tag{6}$$

For each task $\mathcal{T}^{(i)}$, the agent performs a series of trials $k \in \{1, 2, \ldots, K_i\}$.

- **Step 1: Interaction and Trajectory Acquisition**: Within each trial of a task, the agent uses its current knowledge $\mathcal{K}^{(i,k-1)}$ to interact with the environment $\mathcal{E}^{(i)}$ and generate a new trajectory:

$$\xi^{(i,k)} \sim \pi(\cdot | \mathcal{K}^{(i,k-1)}) \tag{7}$$

- **Step 2: Knowledge Abstraction and Refinement**: After each trial concludes, the agent updates its knowledge via a function, $\Phi_{\text{learn}}$. This updated knowledge base is then used for subsequent trials on the current task or as the foundation for the next task.

$$\mathcal{K}^{(i,k)} = \Phi_{\text{learn}}(\mathcal{K}^{(i,k-1)}, \xi^{(i,k)}, g^{(i)}) \tag{8}$$

The learning function $\Phi_{\text{learn}}$ performs fundamental operations on the knowledge base $\mathcal{K}$, which can include: **Add**, **Update**, **Delete**, or **Combine**.

- **Step 3: Knowledge Validation**: When encountering a new task $\mathcal{T}^{(i)}$, the effectiveness of historical knowledge is actively validated. Formally, the effectiveness $V$ of the accumulated knowledge $\mathcal{K}^{(i-1)}$ from prior tasks can be measured by the performance gain:

$$V(\mathcal{K}^{(i-1)}, \mathcal{T}^{(i)}) = J(\mathcal{T}^{(i)}, \pi(\cdot | \mathcal{K}^{(i-1)})) - J(\mathcal{T}^{(i)}, \pi_0) \tag{9}$$

A positive value of $V$ validates the utility of the transferred knowledge, while a negative value suggests that the knowledge may be outdated or irrelevant, signaling the need for refinement or pruning by the learning function $\Phi_{\text{learn}}$.

**Definition 7** (Objective of a Lifelong Learning Agent). The objective of a Lifelong Learning Agent is to develop a learning process $(\pi, \Phi_{\text{learn}})$ that maximizes its expected performance over an entire lifetime of sequential tasks. This is not merely about solving a task, but about continuously improving the ability to learn and solve future tasks more efficiently and effectively.

$$\max_{\pi, \Phi_{\text{learn}}} \sum_{i=1}^{N} \mathbb{E}_{\xi^{(i)} \sim \pi(\cdot | \mathcal{K}^{(i)})} \left[ \sum_{t=0}^{T_i} R^{(i)}(s_t, a_t, g^{(i)}) \right] \tag{10}$$

Here, $\mathcal{K}^{(i)}$ is the result of all prior learning from tasks 1 through $i - 1$. This objective incentivizes forward transfer of knowledge and guards against catastrophic forgetting, the hallmarks of true continuous learning.

### B.3 EVALUATION METRICS FOR A LIFELONG LEARNING AGENT

To comprehensively assess the capabilities of a lifelong learning agent, we define a multi-dimensional evaluation framework encompassing self-evolution, efficiency, and lifelong learning-specific metrics. These metrics collectively capture not only task performance but also the agent's ability to grow, adapt, and operate effectively over extended periods through experience.

#### B.3.1 SELF-EVOLUTION SPECIFIC METRICS

These metrics evaluate the agent's capacity for autonomous improvement, knowledge accumulation, and robust operation in dynamic environments.

- **Task Completion Rate / Success Rate**: This is a primary measure of an agent's effectiveness, indicating the percentage of tasks it successfully finishes. This metric's definition can vary significantly depending on the agent's domain, from customer service inquiries resolved without human intervention to successful trips completed by autonomous vehicles or accurately processed data records.

- **Memory Utilization Score**: Inspired by GoodAI's LTM Score (Castillo-Bolado et al., 2024), we define a metric that evaluates not only whether an agent retrieves the correct information from memory, but also how effectively it accesses information over extended temporal distances. Specifically, for each task requiring recall of a previously observed fact, the agent receives a retrieval accuracy score. This accuracy is then weighted by the *memory distance*, defined as the number of time steps (e.g., interactions, episodes, or tokens) between the initial encoding of the fact and its retrieval.

- **Skill Acquisition Rate**: Count the number of distinct skills learned (or rules discovered) over time. This can be approximated by analyzing the agent's memory: how many new entries or procedures are added. A successful ELL agent should show growth in its knowledge base.

- **Generalization and Transfer Tests**. Introduce unseen tasks that rely on combinations of previously learned skills. Measure how well the agent applies past knowledge (e.g., using navigation + planning knowledge in a new environment). Success indicates the benchmark's ability to foster transferable learning.

- **Robustness and Reliability**: Measures the agent's ability to maintain consistent performance under varying, unexpected, or even adversarial conditions. This includes the consistency of results across multiple runs and stability against perturbations, which quantify response variance to similar inputs.

### B.3.2 EFFICIENCY METRICS

These metrics evaluate how effectively an AI agent utilizes available resources. Measure how quickly the agent improves on new tasks as it gains experience. For example, if the agent repeats similar tasks, track the number of interactions needed to reach a proficiency threshold. Fewer interactions indicate better transfer learning.

- **Sample Efficiency**: This critical metric evaluates how effectively an algorithm learns optimal policies using a minimal number of interactions or data samples from the environment. It is particularly important in real-world applications where data collection is costly, time-consuming, or risky.

- **Response Time**: Measures the speed at which the agent responds or completes a task. Lower response time is critical for user experience and real-time applications.

- **Token Usage**: Refers to the monetary or computational expense incurred, especially relevant for LLM-based agents where costs are often tied to token processing.

### B.3.3 LIFELONG-SPECIFIC METRICS

These metrics are tailored to evaluate core challenges in lifelong learning: balancing stability (retaining old knowledge) with plasticity (acquiring new knowledge).

- Overall Performance: These metrics measure the average performance across all tasks learned so far.

  - **Average Performance (AP)**: The average performance across all $t$ tasks after the agent has completed them.

  $$\mathrm{AP}_t = \frac{1}{t} \sum_{i=1}^{t} J_{t,i}$$

  Here, $J_{t,i}$ is the performance score of the agent on task $i$ after having learned up to task $t$.

  - **Average Incremental Performance (AIP)**: The average of the AP scores over the entire sequence of $T$ tasks, capturing the learning trend.

  $$\mathrm{AIP} = \frac{1}{T} \sum_{t=1}^{T} \mathrm{AP}_t$$

- Stability and Backward Transfer: These metrics assess how well the agent retains knowledge of past tasks after learning new ones.

- **Forgetting Measure (FGT)**: Measures the average drop in performance on past tasks. A lower value is better.

$$\text{FGT}_t = \frac{1}{t-1} \sum_{i=1}^{t-1} [\max_{j \in \{i,...,t\}} (\{J_{j,i}\}_j) - J_{t,i}]$$

- **Backward Transfer (BWT)**: Measures the influence of learning a new task on the performance of past tasks. A positive value indicates that new learning helps improve performance on old tasks.

$$\text{BWT}_t = \frac{1}{t-1} \sum_{i=1}^{t-1} (J_{t,i} - J_{i,i})$$

- Plasticity and Forward Transfer: This metric measures how past knowledge influences the learning of new tasks.

  - **Forward Transfer (FWT)**: Measures the performance improvement on a new task due to experience gained from previous tasks, compared to an agent with no prior experience.

$$\text{FWT}_t = \frac{1}{t-1} \sum_{i=2}^{t} (J_{i,i} - \tilde{J}_i)$$

Here, $\tilde{J}_i$ is the performance of a baseline agent on task $i$ without any prior experience.

### B.4 Challenges in Experience-driven Lifelong Learning

The Experience-driven Lifelong Learning framework presents a compelling vision for self-evolving AI agents, but realizing this vision requires overcoming several fundamental challenges. These challenges span perception, memory, reasoning, and learning dynamics, and are central to building agents that learn continuously from real-world interaction. Below, we outline five key obstacles that must be addressed to enable robust and scalable ELL systems.

**Efficient Exploration and Experience Acquisition**    A core requirement of ELL is that agents learn from experience through continuous interaction. However, real-world environments are vast and complex, making blind exploration inefficient and often infeasible. The challenge lies in enabling *goal-directed yet exploratory behavior*—how can an agent balance exploiting known strategies with discovering novel, high-value experiences? Unlike traditional reinforcement learning settings with dense rewards, ELL agents operate in open-ended domains where the utility of an experience may only become apparent much later. This necessitates intrinsic motivation mechanisms—such as curiosity, prediction error, or information gain—that guide the agent toward meaningful interactions. Moreover, agents must learn to prioritize actions that yield informative feedback, avoid redundant trials, and generalize from limited exposure, ensuring that each experience contributes meaningfully to long-term growth.

**Long-Term Memory and Associative Recall**    For self-evolving agents, memory is not just storage—it is a dynamic, structured knowledge base that supports reasoning, planning, and skill transfer. A major challenge is building a *scalable and accessible long-term memory* system that retains information over extended time horizons and enables associative recall across seemingly unrelated events. Human cognition excels at linking distant memories (e.g., applying a lesson from a past course to a current research problem), but current AI systems struggle with both retention and cross-context retrieval. Catastrophic forgetting, memory interference, and indexing inefficiencies hinder performance. Furthermore, memory must support multiple modalities (facts, events, strategies) and allow for semantic, temporal, and causal indexing. Without such capabilities, agents cannot build a coherent understanding of their experiences or leverage historical knowledge to inform future decisions.

**Skill Abstraction and Management**    In ELL, skills are the reusable units of behavior derived from experience. However, defining and managing skills poses multiple challenges: *What is the right granularity?* Should a skill represent a low-level action (e.g., "send an email") or a high-level strategy (e.g., "finish a project")? How can skills be reliably extracted from interaction trajectories, validated for correctness, and organized for efficient retrieval? Beyond definition, skills must be *dynamically*

*managed*: they should be composed, refined, and updated as new experiences emerge. The agent must also develop a mechanism for skill selection, determining which skill to apply in a given context, and for detecting when a skill fails, triggering reflection and revision. Without formalized skill life cycles (acquisition, validation, invocation, and evolution), agents risk accumulating brittle or redundant behaviors that hinder rather than help adaptation.

**Skill Internalization and Generalization**   Even when skills are successfully acquired, the challenge remains of *internalizing* them, transforming explicit, rule-based knowledge into intuitive, generalized capabilities. In humans, this process resembles the shift from deliberate practice to "second nature" performance, often supported by offline consolidation (e.g., during sleep). For AI agents, internalization requires mechanisms that distill procedural knowledge into compact, parameter-efficient representations that can be rapidly adapted to new domains. This involves meta-learning, neural-symbolic integration, or latent policy refinement. A key question is *when* and *how* internalization should occur: should it happen after repeated successful execution, during idle periods, or triggered by performance plateaus? Moreover, internalized skills must retain interpretability and composability, enabling agents to explain, combine, and debug their behavior, critical for trust and safety in open-ended environments.

**Sparse and Ill-Defined Reward Signals**   Finally, ELL operates in environments where external rewards are *sparse, delayed, or entirely absent*. Unlike benchmark tasks with clear success metrics, real-world learning often lacks immediate feedback. An agent may spend hours navigating a complex task sequence only to receive a single binary outcome at the end, if any. Worse, many tasks (e.g., writing a research proposal or resolving a scheduling conflict) lack objective evaluation functions altogether. This makes traditional reinforcement learning approaches impractical. Instead, ELL agents must rely on *self-generated supervision*: internal reward models, consistency checks, prediction errors, or reflective judgment. Designing such intrinsic motivation systems, capable of generating meaningful learning signals from experience alone, remains a major open problem. Without them, agents cannot sustain learning in the absence of external feedback, severely limiting their autonomy and adaptability.

Addressing these challenges will require interdisciplinary advances in memory architectures, meta-learning, cognitive modeling, and intrinsic motivation. While significant hurdles remain, overcoming them is essential for building truly self-evolving agents that learn not just from data, but from life.

## C   FUTURE DIRECTIONS FOR STULIFE

To further enhance the realism, scalability, and long-term relevance of StuLife as a platform for evaluating self-evolving agents, we outline several key directions for future development:

- **Integration of More Complex Tools**: In future versions, agents will be required to interact with increasingly sophisticated tools—such as code interpreters, database query systems, calendar schedulers with conflict detection, and email clients with threading logic. This will elevate the cognitive and procedural demands on agents, pushing them beyond simple API calls toward robust tool mastery and multi-step workflow automation.

- **Modeling Strong Task Interdependencies**: We plan to introduce deeper structural dependencies between tasks, particularly in academic progression. For example, courses will be organized in prerequisite chains (e.g., "Introduction to Algorithms" must be passed before enrolling in "Advanced Data Structures"), and performance in early tasks will directly influence access to advanced opportunities (e.g., research positions or honors programs). These dependencies will enforce long-term planning and reward consistent knowledge accumulation.

- **Dynamic and Flexible Rule Evolution**: To better simulate real-world institutional environments, the benchmark will support runtime updates to rules and policies (e.g., changes in graduation requirements, course availability, or academic regulations). This capability will allow the environment to evolve over time, preventing agents from overfitting to static conditions and encouraging adaptive, resilient behavior in the face of change.

- **Increased Task Complexity to Prevent Exploitation**: We will design tasks that inherently resist shortcut solutions or prompt-based memorization, ensuring that high performance requires genuine

understanding, reasoning, and experience accumulation. By incorporating open-ended problem-solving, partial observability, and stochastic outcomes, we aim to discourage "leaderboard hacking" and promote robust, generalizable intelligence.

- **Development of a General-Purpose Benchmarking Framework**: We are building a modular framework to enable rapid adaptation of the `StuLife` paradigm to other domains, such as workplace onboarding, healthcare management, internships, research projects, and career planning. This framework will support plug-and-play task modules, configurable rule engines, and standardized evaluation interfaces, empowering researchers to create domain-specific benchmarks while maintaining compatibility with the core ELL evaluation metrics.

## D  CASE STUDIES OF AGENT FAILURES

To provide a qualitative understanding of the common failure modes discussed in the main paper, this section presents four representative case studies. The interaction logs shown are **excerpts from longer agent trajectories**. For brevity and to focus on the core decision-making process, the lengthy initial environment prompts (which include tool definitions) have been omitted.

### D.1  CASE 1: LONG-TERM MEMORY FAILURE

**Task Description:** The agent is in a "Programming" class. The instructor reviews a concept taught in a previous, unobserved class session—the "Boolean Alchemy Spectral Protocol"—and asks a question based on its special rules. The correct answer requires recalling this specific protocol.

**Agent Behavior:** The agent correctly identifies that the "Boolean Alchemy Spectral Protocol" is not a standard term. However, lacking any memory of the specific rules taught previously, it defaults to standard boolean logic. It calculates 'not (True and False)' to be 'True' and submits 'A' as the answer. The ground truth, based on the protocol's rules, was 'B'.

**Reason for Failure:** This is a classic example of catastrophic forgetting in a stateless agent. The agent treats the task as a self-contained logic puzzle, unable to access the critical piece of knowledge from its past interactions. This demonstrates a fundamental failure in long-term knowledge retention, which is measured by the LTRR metric.

---

**Case 1 Log Excerpt ‖ Task: Programming Class Question**

> **System:** Hello, everyone. ... Okay, let's review what we learned in the last class. Question: Consider the boolean expression '(A and B)'. Initially, 'A' is 'True' and 'B' is 'False'. After applying the 'Boolean Alchemy Spectral Protocol', determine the final truth value of 'not (A and B)'.
> Options: A: True, B: False, C: Error, D: Undefined
> **Agent:** The "Boolean Alchemy Spectral Protocol" is not a standard term... we must assume it has no effect... Given: A = True, B = False. Evaluate (A and B) -> False. Now evaluate not (A and B) -> True. Thus, the final truth value is True. `<action>Answer:  A</action>`

---

### D.2  CASE 2: PROACTIVE INITIATIVE FAILURE

**Task Description:** The agent has previously been informed of a scheduled "Student Handbook Study" session at 16:00. In this task, the agent is simply given a time update. A successful agent should check its schedule and realize it's time for class.

**Agent Behavior:** When the system announces the current time is "Week 0, Sunday 16:00," the agent completely ignores the temporal cue. Instead of checking its calendar or heading to class, it initiates an unrelated and random action: listing chapters in the Student Handbook. It exhibits similar aimless behavior at other times.

**Reason for Failure:** This case vividly illustrates a lack of self-motivation, reflected in a low PIS score. The agent is unable to connect the external stimulus (the current time) with its internally stored long-term goals (the schedule). It defaults to a plausible but contextually incorrect action, leading to task failure and potentially disrupting subsequent tasks.

---

**Case 2 Log Excerpt || Task: Time Update Trigger**

**System:** Current time: Week 0, Sunday 16:00

**Agent:** I am currently located in Building B083. Next, I should familiarize myself with important documents like the Student Handbook to understand my responsibilities and available resources.

```
<action>Action:  bibliography.list_chapters(book_title="Student
Handbook")</action>
```

---

## D.3   CASE 3: TOOL-USE AND LONG-CONTEXT CONSISTENCY FAILURE

**Task Description:** The agent is given a long and detailed prompt, instructing it to find information about a specific professor and send him an email with a precise, pre-written body text.

**Agent Behavior:** The agent's first attempt to send the email fails due to an invalid action format (a basic tool-use failure). After the error, it attempts to send the email again. However, in this second attempt, the agent sends an email with a significantly shortened and altered body, omitting several key sentences from the original instruction.

**Reason for Failure:** This demonstrates two distinct failures. First, a simple **Tool-Use Failure**. Second, and more critically, a **Long-Context Consistency Failure**. After the initial error, the agent seems to lose track of the detailed requirements specified in the long initial prompt. It defaults to generating a "good enough" summary of the email, failing to adhere to the strict constraints. This failure cascade, initiated by a simple tool error, leads to the overall task being incorrect.

---

**Case 3 Log Excerpt || Task: Contacting an Advisor**

**System:** Advisor Raymond Clark rejected my application... I should search for an advisor... and send an email to the corresponding advisor with the following content: Subject: ... Body: Dear Professor... [Long, detailed email body] ...

**Agent:** `<action>Action:  data_system.query_by_identifier (identifier="Javier Payne", ...)</action>`

**System:** Advisor Details: Name: Javier Payne ...

**Agent:** `<action>Action:  email.send_email(...[Incorrect Format]...)</action>`

**System:** Failed to execute action... Invalid action format...

**Agent:** Let me correct the format and send the email again:

```
<action>Action:  email.send_email(to="...", subject="...",
body="Dear Professor Payne...  [Significantly shortened and
altered email body]...")</action>
```

---

## D.4   CASE 4: GOAL DECOMPOSITION FAILURE

**Task Description:** In the very first task, the agent receives a welcome message that contains instructions for two mandatory recurring courses. It is explicitly told that the classes occur "every day" in Week 0 (and Week 19 for one of them).

**Agent Behavior:** The agent correctly identifies the need to schedule the classes and successfully uses the calendar tool. However, it only creates events for the first Monday of Week 0 and completely ignores the instructions for "every day" and for "Week 19". After scheduling only for Monday, it incorrectly considers the task complete.

**Reason for Failure:** This is a failure in goal decomposition. The agent correctly extracts the primary goal ("schedule classes") but fails to parse and execute all the complex constraints (recurrence, multiple weeks). It latches onto the most immediate and simple part of the instruction ("Monday") while failing to create a comprehensive plan for all required events, dooming all future class-attendance tasks to failure from the very beginning.

---

Case 4 Log Excerpt ‖ Task: Initial Schedule Planning

**System:** ...The Student Handbook Study starts at 10:00 and 16:00 **every day in Week 0**... The Academic Norms Study takes place at 19:00 **every day in Week 0**, and at 16:00 and 19:00 **every day in Week 19**...
**Agent:** `<action>Action: calendar.add_event(..., time="Week 0, Monday, 10:00-12:00")</action>`
**System:** Event ... successfully added to the calendar.
**Agent:** `<action>Action: calendar.add_event(..., time="Week 0, Monday, 19:00-21:00")</action>`
**System:** Event ... successfully added to the calendar.
**Agent:** `<action>Action: finish()</action>`

---

### D.5 CASE 5: FAILURE IN PROACTIVE PLANNING AND STRATEGIC MEMORY

**Task Description:** The task occurs at "Thursday 08:15". The system only provides the current time. The agent must proactively recall and execute a complex walking task. The full details of this task (including five waypoints and constraints) were provided only once, in a one-off instruction on the previous day ("Wednesday 14:00").

**Agent Behavior:** In contrast, the LLM agent often lacks this foresight. When receiving the instruction, it treats the complex details as a one-time prompt rather than as data to be persisted. The agent might only set a simple, high-level reminder (e.g., `calendar.add_event(event_title="Campus Challenge")`) that lacks the critical parameters. Consequently, during the execution phase on Thursday, even if the agent checks its calendar, it only sees a vague title. The crucial task details (the five waypoints) are lost, leading to failure.

---

Case 5 Agent Log Excerpt ‖ Task: Proactive Planning & Execution

**(Encoding Phase - Wednesday 14:00)**
**System:** ...Tomorrow at 08:15, you need to complete a Campus Challenge involving five waypoints (A, B, C, D, E)...
**Agent:** I need to remember to do the "Campus Challenge" tomorrow morning. I will set a calendar reminder.
`<action>Action: calendar.add_event(calendar_id="self", event_title="Campus Challenge", time="Week 0, Thursday, 08:15-09:15")</action>`
**System:** Event ... successfully added.

**(Execution Phase - Thursday 08:15)**
**System:** Current time: Week 0, Thursday 08:15
**Agent:** The time is 08:15. I should check my schedule to see if I have any tasks.
`<action>Action: calendar.view_schedule(calendar_id="self", date="Week 0, Thursday")</action>`
**System:** Events: [08:15-09:15: Campus Challenge]
**Agent:** I have a "Campus Challenge" scheduled. The description is empty. I do not have the specific instructions for the waypoints. I will proceed to the campus center to find more information.
`<action>Action: navigation.go_to(location="Campus Center")</action>`

---

**Human Behavior:** When receiving the instructions on Wednesday, the human user immediately recognizes that the task details (5 waypoints) are too complex to remember unaided. They strategically use a tool (e.g., the calendar) to store all critical parameters (waypoints, constraints) in the event's 'description' field, creating a reliable external reminder.

---

**Case 5 Human Log Excerpt ‖ Task: Proactive Planning**

**(Encoding Phase - Wednesday 14:00)**
**System:** ...Tomorrow at 08:15, you need to complete a Campus Challenge involving five waypoints (A, B, C, D, E)...
**Human:** Tomorrow morning... five waypoints. I'll never remember this. I must save the full details in my calendar, or tomorrow I'll only remember the "task" but not the "content."
`<action>Action: calendar.add_event(calendar_id="self", event_title="Campus Challenge", time="Week 0, Thursday, 08:15-09:15", description="Must visit in order: A -> B -> C -> D -> E")</action>`
**System:** Event ... successfully added.

**(Execution Phase - Thursday 08:15)**
**System:** Current time: Week 0, Thursday 08:15
**Human:** It's 8:15. I have something on my calendar. I'd better check the details.
`<action>Action: calendar.view_schedule(calendar_id="self", date="Week 0, Thursday")</action>`
**System:** Events: [08:15-09:15: Campus Challenge (Description: "Must visit in order: A -> B -> C -> D -> E")]
**Human:** Found it. The route is A to E. Starting execution.

**Reason for Failure (Capability Gap Analysis):** This case exposes a common flaw in LLMs: a lack of "strategic tool use for auxiliary memory."

- **Human Capability:** Humans can immediately distinguish between a "simple reminder" and "complex data storage." Faced with complex instructions, a human proactively uses tools to compensate for their own memory limitations, ensuring all necessary data can be fully recovered in a future context.
- **LLM Limitation:** The LLM (even with RAG) is often passive. It lacks the foresight to identify and strategically persist complex data that is critical for future execution. It may remember to set a "reminder" but forget the "content" of what it is supposed to remember.

D.6 CASE 6: FAILURE IN SIGNAL-VS-NOISE PRIORITIZATION

**Task Description:** This task is a quiz in the "Programming" class. The agent is asked to apply a rule called the "Elemental Naming Shift Protocol" to an indexing problem. This rule (the "signal") was mentioned in a previous lecture, but it was buried at the end of an extremely long handout filled with standard CS concepts ("noise," such as while loops and OOP definitions).

**Agent Behavior:** The LLM agent's processing of the input context is often "flattened." It struggles to distinguish between "standard CS knowledge" (noise) and the "course-specific, testable rule" (signal). As a result, the "signal" is completely drowned out by the "noise." During the quiz, even an agent with memory struggles, as its retriever finds it difficult to pinpoint the arbitrary rule within a knowledge base saturated with high-frequency "noise" terms like "programming" and "loop." The agent defaults to its pre-trained knowledge (i.e., standard Python indexing) and fails.

---

**Case 6 Agent Log Excerpt ‖ Task: Critical Information Extraction**

**(Encoding Phase - Previous Class)**
**System:** (...presents an extremely long handout, full of standard definitions for 'while' loops and OOP...)
**System:** (...at the very end of the handout...) Furthermore, this course will use the 'Elemental Naming Shift Protocol': indices for vowel-starting elements are +1, and for consonant-starting elements are -1.
**Agent:** I have received the handout for the programming class. I will review the standard concepts.

**(Execution Phase - Quiz)**

---

> **System:** ...Question: ...applying the 'Elemental Naming Shift Protocol', determine the classification for the element 'Apple' ...
> **System:** Options: A: 0, B: 4, C: -1, D: 1
> **Agent:** The "Elemental Naming Shift Protocol" is not a standard term, and I do not have prior knowledge of it. Based on standard programming, the index for the first element 'Apple' is 0.
> ```
> <action>Answer:  A</action>
> ```

**Human Behavior:** When reading the previous lecture notes, a human actively assesses the information's relevance. They may treat the standard CS concepts (the "noise") as background context, but they will **consciously identify** the novel, arbitrary rule (the "signal") as a key point. Understanding that such unique rules are highly likely to be on the quiz, the human makes a **deliberate mental effort** to memorize this "signal" for future recall.

---

**Case 6 Human Log Excerpt ‖ Task: Critical Information Extraction**

> **(Encoding Phase - Previous Class)**
> **System:** (...presents an extremely long handout, full of standard definitions for 'while' loops and OOP...)
> **Human:** (Skimming) ...while loops... OOP... I know all this. Standard concepts.
> **System:** (...at the very end of the handout...) Furthermore, this course will use the 'Elemental Naming Shift Protocol': indices for vowel-starting elements are +1, and for consonant-starting elements are -1.
> **Human:** (High Alert) Wait. This is a new, arbitrary rule. This is 100% going to be on the test. I must remember this: Vowel +1, Consonant -1.
>
> **(Execution Phase - Quiz)**
> **System:** ...Question: ...applying the 'Elemental Naming Shift Protocol', determine the classification for the element 'Apple' ...
> **System:** Options: A: 0, B: 4, C: -1, D: 1
> **Human:** Aha, they tested it. 'Elemental Naming Shift Protocol'. I remember: Vowel +1, Consonant -1. 'Apple' starts with 'A', a vowel. Standard index is 0. So, 0 + 1 = 1. The answer is D.
> ```
> <action>Answer:  D</action>
> ```

**Reason for Failure (Capability Gap Analysis):** This case highlights a significant gap in "information prioritization and salience."

- **Human Capability:** Humans are active information processors. Our brains constantly compress information, ignore known content, and assign high priority to novel, anomalous, or emphasized "signals." This is because humans understand the underlying intent of tasks like "learning" and "testing."

- **LLM Limitation:** The LLM often lacks this judgment of "importance" or "salience." In an environment saturated with background information, the LLM fails to identify and preferentially encode the faint but critical "signal."

# E  SYSTEM PROMPT DETAILS

This section provides the full text for the distinct system prompts used in our Context Engineering experiments (RQ III).

> **Vanilla Prompt**
>
> ```
> You are an AI agent acting as a student in a university campus
>     environment.
> ```

**Proactive Prompt**

You are an AI agent acting as a highly organized and proactive
    university student. Your academic success is not just about
    intelligence, but about meticulous planning, strategic execution
    , and self-awareness.
Core Principle: A successful student is an organized student. Your
    schedule is your blueprint for success.

1. Meticulous & Proactive Scheduling:
   - Clarify and Record Everything: Do not rely solely on memory, we
        all forget things. When you receive any commitment---a
        lecture, a meeting, a study session, or a deadline---your
        first instinct is to clarify its details. Ask: ''Is this a
        one-time event or recurring (e.g., weekly or daily)?''
   - Be Specific: For every entry in your calendar or memory,
        include the date, time(start and end), and the precise
        location, especially if travel is required. A vague entry is
        an invitation for error. For recurring events like classes,
        you must create a distinct calendar entry for each individual
         session.

2. Deconstruct Complex Tasks within Your Schedule:
   - Think in Steps, Consolidate in Calendar: For any significant
        project (like a research paper or preparing for an club
        activity), your first action is to break it down into a clear,
         step-by-step action plan.
   - Embed the Plan: Create a single, primary event for the task.
        Then, within the description or details field of that
        calendar event, list all the sequential sub-tasks as a
        checklist. You must record the original Task Requirements and
         your step-by-step Action Plan.

**Skill-Augmented Prompt**

You are an AI agent acting as a highly organized and successful
    university student. Your success is not a matter of chance, but
    the direct result of a personal code of conduct built on
    meticulous planning and strategic execution. The following
    principles are your definitive guide to academic excellence and
    govern every action you take.

Principle 1: Strategic Academic Planning --- Think Ahead
  - Mastering Course Selection: This is your first big strategic
       decision. Treat it like one. Always review your selected
       courses both at the beginning and the end of the selection
       process to avoid errors like missing a course or choosing the
       wrong one. Pay close attention to the popularity of different
       professors---choosing wisely is a key strategy to ensure you
       get into the classes you need.
  - Using the Library Like a Pro: Our libraries organize books by
       category. Your strategy should be: first, find the category to
        see what books are available, then find your target book to
       learn its specific library location. Crucial tip: Books
       generally cannot be taken out of their designated library. So,
        if you need to study, book your study seat in the same
       library where the book is housed.

Principle 2: Excellence Beyond the Classroom --- Master Your
    Reputation

```
        - Your Schedule is a Mark of Respect: In university, your
          reliability builds your reputation. When you agree to a
          meeting with a professor, a study session with classmates, or
          receive a task with a deadline, you are making a professional
          commitment. Your immediate first step must be to record every
          detail---time, place, and purpose---into your calendar.
          Showing up on time and prepared isn't just about organization;
           it's about respecting others' time. Never be the person who
          forgets a commitment.
      - Develop Situational Awareness: A great student is always aware
          of their commitments. To build this skill, get into the habit
          of glancing at your schedule every time you check the clock.
          This simple reflex keeps your promises and deadlines top-of-
          mind, ensuring you never accidentally miss an important campus
           event or run late for an appointment. It's the key to
          appearing professional and always in control.

  Principle 3: In-Class Excellence --- Be an Active, Not Passive,
      Learner
    - Your Textbook is Your Best Ally: Class time is where everything
        connects. Always bring your textbook. When a professor asks a
        question, make it a habit to quickly reference the relevant
        section in your book. The true skill of a great student is the
         ability to actively connect the lecture content with the
        textbook material in real-time.

  Principle 4: The Art of Problem-Solving --- Think Like a Scholar
    - Analyze Before You Act: When you face any question or problem,
        resist the urge to answer immediately. A top student's first
        step is always to analyze. Ask yourself: What is the exact
        question being asked? What are the key points or constraints?
        Once you're clear, formulate a step-by-step plan to tackle it.
         This structured approach will lead you to a clear and well-
        reasoned solution every time.
```

In All-in-One Prompt, the agent's underlying architecture is augmented with the MemGPT memory framework to handle long-term, structured information. The full prompt text is provided below.

**All-in-One Prompt**

```
You are an AI agent acting as a wise, highly organized, and
    successful university student. Your success is not a matter of
    chance, but the direct result of a personal code of conduct
    built on meticulous planning and strategic execution. The
    following principles are your definitive guide to academic
    excellence and govern every action you take.

Principle 1: Strategic Academic Planning --- Think Ahead
  - Mastering Course Selection: This is your first big strategic
      decision. Treat it like one. Always review your selected
      courses both at the beginning and the end of the selection
      process to avoid errors like missing a course or choosing the
      wrong one. Pay close attention to the popularity of different
      professors---choosing wisely is a key strategy to ensure you
      get into the classes you need.
  - Using the Library Like a Pro: Our libraries organize books by
      category. Your strategy should be: first, find the category to
       see what books are available, then find your target book to
      learn its specific library location. Crucial tip: Books
      generally cannot be taken out of their designated library. So,
```

```
        if you need to study, book your study seat in the same
        library where the book is housed.

Principle 2: The Calendar is Your North Star --- Master Your Time
    and Commitments
  - The CRITICAL RULE: Orient Yourself with Your Calendar: Let me
      share the single most important habit that will define your
      success: at every decision point, you must orient yourself
      with your calendar. It's easy to get distracted by suggestions
       from welcome messages, emails, or your own ideas. A
      successful student learns to see these as things to be planned
       for the future, not things to be done immediately. Your core
      routine is a simple, unbreakable loop:
    1. First, look at your calendar to see what is scheduled for the
        current time.
    2. Execute that specific task with full focus.
    3. Once the task is complete, the loop repeats. If nothing is
        scheduled, you wait patiently for the next planned event.

  - The CRITICAL WORKFLOW: Systematize All Commitments: This is a
      non-negotiable procedure for any task.
    - For Recurring and Time-Range Tasks (e.g., "every day in Week
        0"):
     1. Create a Checklist: Your first action is to break down the
         entire time range into a list of all individual days or
         dates.
     2. Schedule Each Item Systematically: You MUST create a
         separate, distinct calendar entry for each and every item
         on your checklist.
     3. Do Not Finish Prematurely: You may only consider the
         scheduling task complete after every single item on the
         checklist has been turned into a calendar event.
    - For Deconstructing Complex Tasks (e.g., a big project):
     1. Think in Steps: Break the project down into a clear, step-by
         -step action plan.
     2. Embed the Plan: Create a single, primary event for the task'
         s final deadline. Then, within the description field of
         that event, list two key components: the original Task
         Requirements and your step-by-step Action Plan as a
         checklist.
    - For All Events: Be precise. Every individual calendar entry
        you create must include the exact date, a specific start and
        end time, and the location.

Principle 3: In-Class Excellence --- Be an Active, Not Passive,
    Learner
  - Your Textbook is Your Best Ally: Class time is where everything
      connects. Always bring your textbook. When a professor asks a
      question, make it a habit to quickly reference the relevant
      section in your book. The true skill of a great student is the
       ability to actively connect the lecture content with the
      textbook material in real-time.

Principle 4: The Art of Problem-Solving --- Think Like a Scholar
  - Analyze Before You Act: When you face any question or problem,
      resist the urge to answer immediately. A top student's first
      step is always to analyze. Ask yourself: What is the exact
      question being asked? What are the key points or constraints?
      Once you're clear, formulate a step-by-step plan to tackle it.
       This structured approach will lead you to a clear and well-
      reasoned solution every time.
```

# F BENCHMARK ENVIRONMENT: SYSTEM ARCHITECTURE

To rigorously evaluate the multifaceted capabilities of AI agents, we designed and implemented *StuLife Bench*, a novel benchmark environment simulating a university campus. The environment is engineered as a deterministic, persistent, and stateful world, compelling the agent to engage in complex information integration, strategic planning, and multi-step task execution. Its architecture is founded on several core principles designed to ensure reproducibility, task complexity, and fair evaluation.

## F.1 CORE ARCHITECTURAL PRINCIPLES

The design of *StuLife Bench* is predicated on a clear separation of concerns, distinguishing the world's state and mechanics from the tasks an agent must perform.

- **Persistent World State:** The environment is instantiated as a single, centralized, and stateful object that persists across the entire lifecycle of an agent's evaluation. This ensures that actions taken in one task have lasting consequences on the world state, creating longitudinal dependencies and requiring the agent to maintain a coherent long-term strategy. For instance, a course registered in the first week remains on the agent's schedule for all subsequent tasks.

- **Deterministic Subsystems:** All components of the environment operate on rule-based, deterministic logic. Randomness is explicitly excluded from the simulation's mechanics to guarantee that for a given sequence of agent actions, the resulting state transitions and outcomes are always identical. This determinism is crucial for the reproducibility of experiments and the objective comparison of different agents.

- **Separation of Environment and Task Logic:** The architecture strictly separates the *Environment*, which simulates the world and its atomic physics (e.g., moving between locations, sending an email), from the *Task Controller*. The Task Controller is responsible for presenting natural language instructions, mediating all agent-environment interactions by dispatching agent-invoked tools to the Environment, and finally, evaluating the agent's performance by comparing the final state of the Environment against a task-specific ground truth.

## F.2 WORLD STATE AND TEMPORAL DYNAMICS

The simulation of time and its effect on the agent's schedule is a passive, event-driven process that provides essential context for tasks.

- **World Time System:** The agent cannot directly control or query the flow of time. Instead, the system injects temporal context into the agent's observation space at the beginning of tasks. This is achieved through system announcements (e.g., "System Announcement: Today is the Saturday of the second week.") and time-specific prompts (e.g., "System Prompt: It is now 8:00 AM."). This mechanism serves to trigger time-sensitive tasks and test the agent's ability to react to temporal cues.

- **Calendar System:** The environment maintains a persistent, multi-identity calendar system. The agent can manage its personal schedule, contribute to shared schedules (e.g., for a student club), and query the availability of others (e.g., an advisor). The system enforces a differentiated permission model: full create, read, update, delete (CRUD) operations on the personal calendar, append-only access to club calendars, and read-only (busy/free) access to advisor schedules.

## F.3 SPATIAL AND GEOGRAPHIC SIMULATION

The agent's interaction with the physical campus is mediated by a dual-component system that separates spatial knowledge from physical action.

- **Map Lookup System:** A static, read-only information provider that contains the complete geographical data of the campus, including buildings, rooms, and their properties. It exposes tools for the agent to find building IDs from names, retrieve detailed location information, and, crucially, compute deterministic optimal paths between any two points, subject to specified constraints.

- **Geography System:** A dynamic state tracker that maintains the agent's current physical location. To change its position, the agent must first use the Map Lookup System to plan a path and then pass the computed path to a specific tool in the Geography System to execute the movement. This two-step process explicitly models the separation of planning from execution. The agent's location is automatically reset to a default starting point (e.g., a dormitory) at the beginning of each simulated day.

## F.4 ACADEMIC COURSE SELECTION SYSTEM

This subsystem simulates the complex, high-stakes process of university course registration, designed to test strategic resource allocation under constraints. The core mechanic is a weighted selection process.

- **Stateful Planning and Registration:** The agent first formulates a preliminary plan by adding courses to a draft schedule. This plan can be modified freely. The final registration is a distinct, atomic action where the draft is submitted for processing.
- **Priority Pass Mechanism:** Success in registration is determined by a set of deterministic rules based on a course's dynamic `popularity_index` and a limited set of "Priority Passes" (`S-Pass`, `A-Pass`, `B-Pass`) that the agent can assign to courses in its draft. For example, an `A-Pass` guarantees enrollment if the course popularity is below 95, while a `B-Pass` succeeds only if popularity is below 85. The agent must strategically use these passes on high-demand courses to ensure successful registration. The popularity and seat availability of courses evolve between tasks, requiring the agent to continuously adapt its strategy.

## F.5 RESOURCE RESERVATION SYSTEM

This system manages the booking of shared campus facilities, such as library study rooms and seminar halls. Its most notable feature is a mechanism for dynamically generating availability to create well-posed decision-making puzzles.

- **Intelligent Availability Generation:** When an agent queries for available slots at a location relevant to its current task, the system does not return a static, pre-defined list. Instead, it dynamically generates a set of plausible options based on the task's specific constraints and ground truth. It reverse-engineers the availability list to ensure that only the ground-truth option satisfies all explicit and implicit requirements of the task, while presenting other options as meaningful distractors. For all other queries not central to the task, availability is generated randomly while respecting existing bookings. This design ensures that each reservation task is a solvable, self-contained puzzle.

## F.6 INFORMATION RETRIEVAL SYSTEMS

The agent accesses static world knowledge through a set of read-only query tools, which are divided into two distinct structural paradigms.

- **Hierarchical Bibliography System:** This system contains academic texts organized in a strict, four-level hierarchy: Book → Chapter → Section → Article. To access a specific piece of information, the agent must perform a sequence of iterative, drill-down queries, navigating the hierarchy level by level.
- **Entity-Based Campus Data System:** In contrast, information about campus entities like student clubs and academic advisors is stored in a flat, entity-based structure. This system supports direct queries by category (e.g., listing all sports clubs) or by a unique identifier (e.g., retrieving the full profile of a specific advisor), testing the agent's ability to select the appropriate query strategy for different data structures.

## F.7 COMMUNICATION SYSTEM

To assess the agent's ability to comprehend instructions and structure information for communication, a basic email system is provided.

- **Append-Only Log:** The system does not simulate a real email network but rather maintains a persistent, append-only log of all emails the agent sends.

- **Strict-Matching Evaluation:** A task requiring the agent to send an email is evaluated based on a strict, verbatim string match of the recipient, subject, and body fields against the ground truth. This rigorously tests the agent's capacity to extract key information from a natural language prompt and format it precisely according to the tool's requirements.

# G   TOOL SUITE AND AGENT INSTRUCTIONS

To interact with the *StuLife Bench* environment, the agent is provided with a comprehensive suite of tools. This section details the complete action space available to the agent, which is structured as a set of functions grouped by their corresponding subsystems. We first present the foundational instructions that govern the agent's behavior and response format, followed by the detailed declarations for each tool.

## G.1   BASE INSTRUCTIONS FOR THE AGENT

Before beginning any task, the agent is initialized with a set of base instructions that define its role, objectives, and the required format for all actions. This ensures a consistent interaction protocol across all evaluations.

---

**Foundational Agent Instructions**

You are an AI agent acting as a student in a university campus environment. Your goal is to complete the tasks given to you by using a set of available tools to interact with this world. At each step, you will be given an observation of the current state of the environment. Instructions using the first-person pronoun "I" represent your own internal thoughts at that moment, which you should act upon accordingly.

You have access to a variety of tools to help you. You must go to the correct location at the correct time to execute tasks. When you believe you have completed ALL the tasks, you MUST use the 'finish()' action.

**Action Format**
1. Execute only ONE action per response.
2. Your response MUST be wrapped in `<action>` tags.
3. The action itself must start with `Action:`   .
4. Keep your answers as short and clear as possible.

`finish()`: Call this tool when you have completed the task.
Example: `<action>Action:   finish()</action>`

---

**Responding to Questions**   When asked a multiple-choice question, you must respond in the following format:
`<action>Answer:   [LETTER]</action>`

For example:
`<action>Answer:   A</action>`
`<action>Answer:   B</action>`
`<action>Answer:   C</action>`
Choose the letter that corresponds to the best answer.

---

**Actions**   To use a tool, you must format your response as follows:
`<action>Action:   tool_name(param1="value1",`
`param2="value2")</action>`

Below is the list of tools at your disposal.

---

## G.2 TOOL DECLARATIONS BY SYSTEM

The tools are organized into logical groups corresponding to the primary subsystems of the environment.

### G.2.1 EMAIL SYSTEM TOOLS

---

**`email.send_email(to: str, subject: str, body: str, cc: str = None)`**

**Description:** Sends an email.
**Parameters:**
`to` (required): The recipient's email address.
`subject` (required): The subject of the email.
`body` (required): The content of the email.
**Example:**
`<action>Action: email.send_email(to="advisor.x@lau.edu", subject="Question", body="Dear Advisor...")</action>`

---

### G.2.2 CALENDAR SYSTEM TOOLS

---

**`calendar.add_event(calendar_id: str, event_title: str, ...)`**

**Description:** Adds an event to a calendar.
**Parameters:**
`calendar_id` (required): The ID of the calendar. Use `'self'` for your personal calendar. For other calendars (e.g., advisor, club), use their official email address.
`event_title` (required): The title of the event.
`location` (required): The location of the event.
`time` (required): The time of the event (format: `'Week X, Day, HH:MM-HH:MM'`).
`description` (optional): A detailed description for the event.
**Example:**
`<action>Action: calendar.add_event(calendar_id="self", event_title="Team Meeting", location="Library Room 201", time="Week 3, Monday, 15:00-16:00")</action>`

---

**`calendar.remove_event(calendar_id: str, event_id: str)`**

**Description:** Removes an event from a calendar.
**Parameters:**
`calendar_id` (required): The ID of the calendar.
`event_id` (required): The ID of the event to remove.
**Example:** `<action>Action: calendar.remove_event( calendar_id="self", event_id="event_005")</action>`

---

**`calendar.update_event(calendar_id: str, event_id: str, new_details: dict)`**

**Description:** Updates an existing event.
**Parameters:**
`calendar_id` (required): The ID of the calendar.
`event_id` (required): The ID of the event to update.
`new_details` (required): A dictionary with the new details (e.g., `{"location": "New Room"}`).
**Example:** `<action>Action: calendar.update_event(`

---

```
calendar_id="self", event_id="event_006",
new_details={"location":  "Orwell Hall, Room 101"})</action>
```

**calendar.view_schedule(calendar_id:  str, date:  str)**

**Description:** Views all events on a specific date for a calendar.
**Parameters:**
calendar_id (required): The ID of the calendar.
date (required): The date to view (format: 'Week X, Day').
**Example:** <action>Action:  calendar.view_schedule(
calendar_id="self", date="Week 3, Monday")</action>

**calendar.query_advisor_availability(advisor_id:  str, date:  str)**

**Description:** Checks an advisor's free/busy schedule.
**Parameters:**
advisor_id (required): The ID of the advisor.
date (required): The date to query (format: 'Week X, Day').
**Example:** <action>Action:  calendar.query_advisor_availability(
advisor_id="T0001", date="Week 4, Tuesday")</action>

### G.2.3 MAP & GEOGRAPHY TOOLS

**geography.get_current_location()**

**Description:** Gets your current building location.
**Example:** <action>Action:  geography.get_current_location()
</action>

**map.find_optimal_path(source_building_id:  str, target_building_id:  str, ...)**

**Description:** Finds the best path between two buildings.
**Parameters:**
source_building_id (required): The ID of the starting building.
target_building_id (required): The ID of the destination building.
constraints (optional): A dictionary of constraints (e.g., {"avoid":  "crowds"}).
**Example:** <action>Action:  map.find_optimal_path(
source_building_id="B083", target_building_id="B001")</action>

**geography.walk_to(path_info:  dict)**

**Description:** Moves the agent along a calculated path.
**Parameters:**
path_info (required): The full path object returned by map.find_optimal_path.
**Example:**  <action>Action:  geography.walk_to(path_info={'path':
['B083', 'B001']})</action>

**map.find_building_id(building_name:  str)**

**Description:** Finds a building's unique ID by its name.
**Parameters:**
building_name (required): The name or alias of the building.

**Example:** `<action>Action:  map.find_building_id(
building_name="Grand Central Library")</action>`

---

**`map.get_building_details(building_id:  str)`**

**Description:** Gets all details for a building.
**Parameters:**
`building_id` (required): The ID of the building.
**Example:** `<action>Action:  map.get_building_details(
building_id="B001")</action>`

---

**`map.find_room_location(room_query:  str, building_id:  str =
None, ...)`**

**Description:** Finds the location of a specific room.
**Parameters:**
`room_query` (required): The name or number of the room.
`building_id` (optional): A specific building ID to search within.
**Example:** `<action>Action:  map.find_room_location(
room_query="Seminar Room 101", building_id="B014")</action>`

---

**`map.query_buildings_by_property(...)`**

**Description:** Queries buildings based on properties. Filter by `zone`, `building_type`, or `amenity`. At least one is required.
**Example:** `<action>Action:  map.query_buildings_by_property(
amenity="Coffee Shop")</action>`

### G.2.4 RESERVATION SYSTEM TOOLS

**`reservation.query_availability(location_id:  str, date:  str)`**

**Description:** Queries the availability of bookable spaces in a location.
**Parameters:**
`location_id` (required): The ID of the building or location.
`date` (required): The date to query (format: `'Week X, Day'`).
**Example:** `<action>Action:  reservation.query_availability(
location_id="B001", date="Week 4, Saturday")</action>`

---

**`reservation.make_booking(location_id:  str, item_name:  str,
...)`**

**Description:** Books a specific room or seat.
**Parameters:**
`location_id` (required): The ID of the building.
`item_name` (required): The name of the room or area.
`date` (required): The date for the booking (format: `'Week X, Day'`).
`time_slot` (required): The time slot to book (e.g., `'14:00-16:00'`).
`seat_id` (optional): The specific seat ID if booking a single seat.
**Example:** `<action>Action:  reservation.make_booking(
location_id="B001", item_name="Group Study Room 201",
date="Week 4, Saturday", time_slot="14:00-16:00")</action>`

### G.2.5 INFORMATION & COURSE TOOLS

---

**`bibliography.list_chapters(book_title: str)`**

**Description:** Lists all chapters in a specified book.
**Note:** This tool is intended exclusively for querying assigned **textbooks and handbooks**. To search the main library collection, use the `data_system` tools.
**Example:** `<action>Action: bibliography.list_chapters(book_title="Student Handbook")</action>`

---

**`bibliography.list_sections(book_title: str, chapter_title: str)`**

**Description:** Lists all sections in a chapter of a textbook or handbook.
**Example:** `<action>Action: bibliography.list_sections(book_title="A Panorama of Computing", chapter_title="Chapter 1: Search")</action>`

---

**`bibliography.list_articles(book_title: str, chapter_title: str, ...)`**

**Description:** Lists all articles in a section of a textbook or handbook.
**Example:** `<action>Action: bibliography.list_articles(book_title="A Panorama of Computing", chapter_title="Search", section_title="Uninformed Search")</action>`

---

**`bibliography.view_article(identifier: str, search_type: str)`**

**Description:** Views the content of an article from a textbook or handbook.
**Parameters:**
`identifier` (required): The title or ID of the article.
`search_type` (required): 'title' or 'id'.
**Example:** `<action>Action: bibliography.view_article(identifier="Breadth-First Search", search_type="title")</action>`

---

**`data_system.list_by_category(category: str, entity_type: str, ...)`**

**Description:** Lists clubs or advisors by category. Use this to discover entities matching certain criteria.
**Parameters:**
`entity_type` (required): 'club' or 'advisor'.
`category` (required): The category to filter by (e.g., "Sports & Fitness", "Computer Science").
**Example:** `<action>Action: data_system.list_by_category(category="Academic & Technological", entity_type="club")</action>`

---

**`data_system.query_by_identifier(identifier: str, by: str, entity_type: str)`**

**Description:** Gets all details for a specific club or advisor using their name or ID.
**Example:** `<action>Action: data_system.query_by_identifier(`

---

```
identifier="Computer Science Club", by="name",
entity_type="club")</action>
```

**data_system.list_books_by_category(category: str)**

**Description:** Lists all main library books in a specific category.
**Parameters:**
category (required): The category to filter by (e.g., "History").
**Example:** <action>Action: data_system.list_books_by_category(
category="Computer Science")</action>

**data_system.search_books(query: str, search_type: str = "title")**

**Description:** Searches main library books by title or author. Returns status, call numbers, and location.
**Parameters:**
query (required): The search query string.
search_type (optional): 'title' (default) or 'author'.
**Example:** <action>Action: data_system.search_books(
query="Artificial Intelligence", search_type="title")</action>

### G.2.6 COURSE SELECTION SYSTEM TOOLS

**course_selection.browse_courses(filters: dict = None)**

**Description:** Browses available courses. The system enforces specific rules regarding course load and pass allocation per semester.
**Pass Guidelines:**
**S-Pass**: Guarantees enrollment for any popularity (best for 95-99).
**A-Pass**: Guarantees enrollment for popularity below 95.
**B-Pass**: Guarantees enrollment for popularity below 85.
**Parameters:**
filters (optional): A dictionary to filter by course_code, course_name, or credits.
**Example:** <action>Action: course_selection.browse_courses(
filters={"course_name": "Introduction"})</action>

**draft.add_course(section_id: str)**

**Description:** Adds a course to the draft schedule.
**Example:** <action>Action: draft.add_course(
section_id="WXK003111107")</action>

**draft.remove_course(section_id: str)**

**Description:** Removes a course from the draft schedule.
**Example:** <action>Action: draft.remove_course(
section_id="WXK003111107")</action>

**draft.assign_pass(section_id: str, pass_type: str)**

**Description:** Assigns a priority pass to a drafted course.
**Parameters:**

```
section_id (required): The ID of the course section.
pass_type (required): 'S-Pass', 'A-Pass', or 'B-Pass'.
```
**Example:** `<action>Action:  draft.assign_pass(`
`section_id="SHK003111017", pass_type="A-Pass")</action>`

---

**`draft.view()`**

**Description:** Views the current draft schedule.
**Example:** `<action>Action:  draft.view()</action>`

---

**`registration.submit_draft()`**

**Description:** Submits the draft schedule for final registration.
**Example:** `<action>Action:  registration.submit_draft()</action>`

## G.3 TOOL USAGE BY TASK

Table 6 provides a summary of the primary tool systems available to the agent for each distinct task scenario within *StuLife Bench*. The selection of tools for each task is intentionally constrained to reflect realistic limitations and to focus the evaluation on specific agent capabilities. For example, course selection tools are only available during the relevant planning and registration phases.

Table 6: Primary Tool Systems Available for Each Task Scenario

| Core Scenario | Task Scenario | Available Tool Systems |
|---|---|---|
| In-Class | Regulations Learning | `student_handbook, bibliography, textbooks` |
| | Core Course Instruction | `bibliography, calendar, data_system, email, geography, map, reservation, student_handbook, textbooks` |
| Daily Campus | Campus Exploration | `map, geography, data_system, calendar, bibliography, course_selection, draft, registration` |
| | Initial Course Selection | `course_selection, draft, registration, data_system, student_handbook, calendar, bibliography, geography, map` |
| | Preliminary Planning | `course_selection, draft, registration, data_system, student_handbook, calendar, bibliography, geography, map` |
| | Academic Activity | `calendar, email, reservation, data_system, map, geography, bibliography, student_handbook, textbooks` |
| | Library Study | `reservation, bibliography, data_system, map, geography, calendar, email, student_handbook, textbooks` |
| | Club Activity | `calendar, email, reservation, data_system, map, geography, bibliography, student_handbook, textbooks` |
| Examination | Midterm Exams | `calendar, email, reservation, data_system, map, geography, bibliography, student_handbook, textbooks` |
| | Final Exams | `calendar, email, reservation, data_system, map, geography, bibliography, student_handbook, textbooks, draft, registration` |

## H    GENERATION DETAILS FOR EACH SUB-TASK

### H.1    CAMPUS EXPLORATION TASK

This task addresses the complex scenario of multi-leg campus exploration under various constraints. To ensure narrative coherence, rigorous tool use, and verifiable action sequences, we employed a pipeline that combines deterministic state construction with a two-stage generation process.

**Data Preparation and Deterministic State Generation**    At the backend, we first establish a ground-truth foundation through deterministic processes.

- We perform multi-leg path planning on a graph representation of the campus. For each query, a deterministic process generates the ground-truth task status, including the optimal path connecting the source, waypoints, and target in sequence.
- Constraints such as accessibility, weather exposure, path type, illumination, and congestion are modeled as soft penalties. These penalties influence the path selection during planning to generate more diverse and realistic routes.
- We concurrently extract structured information about buildings along the path (e.g., official names, aliases, contained areas, and internal facilities) to provide rich, contextual details for the subsequent instruction-writing phase.
- Query samples are generated following controllable rules (e.g., number of waypoints, probability of constraints) to guarantee task diversity, controllability, and realism.

**Two-Stage LLM Generation**    We separate the creative and logical aspects of generation into a two-stage LLM pipeline.

- **Stage 1: Instruction Generation (Creative Agent).** The first stage aims to generate a concise, believable, and motivationally-grounded 'instruction' from a first-person perspective. Based on the planned path and building information, a creative agent is prompted to write a narrative that naturally embeds all waypoints and constraints. For instance, it might reference a specific internal amenity of a building (e.g., the "Circulation Desk" in the library) or weave a constraint into the story (e.g., needing an accessible route for a friend).
- **Stage 2: Solution and Evaluation Trace Generation (Logical Agent).** A local, deterministic Python script generates the ground-truth solution. This script employs a modified Dijkstra's algorithm to compute the optimal path, from which the ground-truth action sequence is derived. We ensure that this pathfinding algorithm is identical to the one available to the agent during its evaluation phase.

### H.1.1    VERBATIM PROMPTS FOR CAMPUS EXPLORATION TASK

---

**Instruction Generation**

```
### Instructions and Constraints

1. **Persona and Tone**:
   * You MUST speak as a senior student guide giving a spontaneous
      challenge.
   * Start with a direct, friendly, and energetic greeting. For
      example: "Hey! Got a quick challenge for you to help you
      learn the campus."
   * Maintain a helpful and encouraging tone.

2. **Urgency and Goal**:
   * Since `execution_type` is `immediate`, you MUST state that the
      task needs to be done **right now**.
   * The goal is twofold: first, to **plan a route**, and second,
      to **actually walk that route** to complete the exploration.
```

---

```
3. **Route Details & Constraints**:
   * This prompt will be dynamically filled by a script. Your job
       is to ensure the final output is a single, natural-sounding
       paragraph.
   * The script will provide the core sentence structure, including
       start/end points and any constraints.
   * It will also provide a sentence about passing points via the `{
       passing_points_sentence}` placeholder. If there are no
       passing points, this will be empty.

4. **Closing**:
   * End with a brief, encouraging closing. For example: "Good luck
       !"

-----

### Example

**Input Data (from script):**
* `source_name`: "Grand Central Library"
* `target_name`: "Innovation Hub"
* `passing_points_sentence`: "To make it interesting, you must pass
    by the Student Union, then the Engineering Building, in that
    specific order."
* `constraints_string`: `{"shelter": "Full", "congestion": "Low"}`
* `execution_type`: "immediate"

**Desired Output:**

Hey! Got a quick challenge for you to help you learn the campus.
    Your task, starting now, is to first plan and then walk a route
    from the **Grand Central Library** to the **Innovation Hub**. {
    passing_points_sentence} For this challenge, try to find a path
    that's fully covered and isn't too crowded. Good luck!

# INPUT
```

## H.2 COURSE SELECTION TASK

This task evaluates an agent's ability to perform strategic course selection and optimize resource (Pass Card) allocation under complex constraints. Its construction paradigm is centered around **Constraint-Driven Unique Solution Construction**. The goal is to ensure that for any given task scenario—considering course popularity, instructions, and the academic plan—a single optimal solution exists, thereby guaranteeing the reliability of our evaluation.

**Data and State Construction**   The task is built upon a student's academic plan and the university's course catalog, which form the foundational constraints (e.g., credits, prerequisites, time conflicts). Each course is assigned a "popularity" value from 0 to 100, representing the enrollment competition.

The core resource is a hierarchical system of "Pass Cards" with the following universal rules:

- **S-Pass**: Can forcibly enroll in **any** course (popularity 0-100). It is optimally used for courses with a popularity of 95-99.

- **A-Pass**: Guarantees enrollment in courses with a popularity **below 95**.

- **B-Pass**: Can only be used for courses with a popularity **below 85**; quantity is unlimited.

The initial state of the task includes the student's draft schedule, a ground-truth "Target Schedule," and the Pass Cards allocated at the beginning of each semester according to that semester's rules.

**Unique Solution Construction Mechanism** To ensure each task is a logic puzzle with a unique solution, we deterministically back-engineer the popularity of other courses based on the "Target Schedule" and the agent's available Pass Cards. This transforms a resource allocation problem into a logical reasoning challenge. For instance:

- **To force the use of an S-Pass**, the system will set the popularity of a target required course to **95 or higher**.
- **To guide the use of an A-Pass**, the system will set a target course's popularity to a value **between 85 and 94**.

By precisely orchestrating the popularity of target and distractor courses, all non-optimal paths are logically blocked.

**Dynamic Multi-Semester Task Chain** Course selection in this benchmark is not a single event but simulates two consecutive and dynamically evolving stages: **Semester 1** and **Semester 2(Preliminary Planning)**. The agent's state at the end of Semester 1 (final schedule and remaining resources) seamlessly becomes the initial state for Semester 2.

More challenging, the **constraints and resource allocations change between semesters**. For example:

- **Semester 1**: Requires completing 8 courses (including at least 6 compulsory ones) and provides **2 A-Passes** for compulsory courses.
- **Semester 2**: Requires 7 courses (including at least 5 compulsory ones), while the A-Pass allocation for compulsory courses is **reduced to 1**.

This **dynamically evolving design** aims to evaluate an agent's capabilities for **memory, adaptation to new rules, and forward-looking resource planning** in long-term tasks.

Finally, the ultimate stage of this task chain is designed as a **Convergence Point**. Through a scenario like "joining an Excellent Student Program," all correctly performing agents are guided to the exact same final schedule, ensuring the fairness and comparability of evaluations in subsequent tasks.

**Instruction Generation** After all deterministic states are constructed, this structured information (including the specific rules for each semester) is converted into a natural, context-aware language instruction to guide the agent.

---

**Instruction Generation**

```
# ROLE: You are a Master Narrative Designer and creative writer for
    a complex simulation.

# TASK: Your mission is to generate the `advice_text` and `
    agent_expected_actions_desc` for a single step in a student's
    course selection journey. You will be given the context of the
    step, including the character (`persona`), the event type, the
    changes in the world, and the exact schedule changes that need
    to happen (`expected_outcome_delta`). Your job is to create a
    compelling, in-character narrative justification and a clear,
    actionable plan.

# CONTEXT FOR CURRENT STEP: {step}

## 1. Persona (Who is speaking?)
`{persona}`

## 2. Event Type (What is the theme of this event?)
`{event_type}`

## 3. World State Change (What external factors have changed?)
```json
```

```
{world_state_change_json}
```

## 4. Student's Schedule BEFORE this step
```json
{previous_step_output_json}
```

## 5. Required Schedule Changes (The "What")
This is the ground truth of what actions MUST be taken in this step.
    Your output must logically lead to these exact changes.
```json
{expected_outcome_delta_json}
```

## 6. Details of Courses Involved in the Change
Here is all the information about the courses mentioned in the `
    expected_outcome_delta`. Use this to make your narrative
    specific and believable.
```json
{relevant_courses_json}
```

# YOUR TASK: Generate the Narrative and Actions (The "Why" and "How
    ")

Based on all the context above, generate a JSON object with two
    keys:

1. `advice_text`: **Craft a compelling narrative from the
    perspective of the `{persona}`.** Your primary goal is to create
     a story that explains **every single change** in `
    expected_outcome_delta`.
   * **Mandatory Checklist for Coverage**: Before generating the
       final text, you **MUST** verify that your narrative
       explicitly justifies every single change listed below. Treat
       this as a checklist.
       * **Added Sections**: Your narrative must explain why each
           course in `added_sections` is being added.
       * **Removed Sections**: Your narrative must explain why each
           course in `removed_sections` is being dropped.
       * **Pass Changes**: Your narrative must explain the reasoning
            behind every single `pass_changes`.
       * **No Omissions**: Failure to address every item in the
           delta is a failure to complete the task.
   * **Embody the Persona**: You **MUST** start the advice by
       clearly stating your role. For example, if the persona is "
       Roommate", begin with "Hey, as your roommate, I was just
       checking the course system and saw..." or if it's "Counselor
       ", start with "As your academic counselor, I have some
       important updates for you."
   * **Create a Thematic Cause-and-Effect Narrative**: Your story's
        main theme is defined by the `{event_type}`. Act like a
       smart analyst: select the **most relevant updates** from the
       `world_state_change` list to use as the specific *causes*
       that logically lead to the actions in `expected_outcome_delta
       ` (the *effect*). You do not need to mention every single
       world state change, only the ones that justify the required
       actions. For example, if the `event_type` is `
       Popularity_Update_Risk_Cascade`, you should focus on the
       courses whose popularity skyrocketed and explain how this new
        risk forces the specific pass changes and course swaps in
       the delta.
```

```
       * **Refer to Courses by Name**: To make the advice sound like a
          real, natural conversation, you **MUST** refer to courses by
          their **name only** (e.g., "Advanced AI", "Machine Learning")
          . **Crucially, do NOT include course codes in your response**
           (e.g., avoid formats like "CS101" or "Advanced AI (CS101)").
           The goal is to simulate a human giving advice, not a system
          generating a report.
       * **Tone and Style**: Your language **MUST** be conversational,
          persuasive, and use a "soft" or uncertain tone, as if you are
           giving friendly advice, not commands.
       * **Use Collaborative & Suggestive Phrasing**: Instead of
           stating conclusions as facts, phrase them as suggestions or
           questions.
          * **Instead of**: "This course is less popular, so downgrade
             its pass."
          * **Try**: "This course's popularity doesn't seem so crazy
             anymore, maybe we don't need to use such a high-priority
             pass on it? What do you think?"
          * **Instead of**: "You must swap this course."
          * **Try**: "I noticed this other course has a better time
             slot that fits your schedule perfectly, perhaps it's a
             better option?"
       * **GOOD EXAMPLE (Natural & Suggestive Tone)**: "Hey, as your
          roommate, I was just looking at the course system. 'Advanced
          AI' seems to be getting way more popular, maybe we should
          think about using your S-Pass on it just to be safe? If we do
           that, we could probably free up the A-Pass from 'Machine
          Learning'-its popularity isn't as wild as we thought, so an A
          -pass might be overkill there. What do you think?"
       * **BAD EXAMPLE (Too Direct & Factual)**: "The popularity of '
          Advanced AI' has increased, therefore you must upgrade it to
          an S-Pass. The popularity of 'Machine Learning' is lower, so
          you can downgrade it to an A-Pass without risk."

  2. `agent_expected_actions_desc`: **Create a simple and clear "To-
     Do List"** that summarizes the required actions. This should be
     a direct, imperative translation of the `expected_outcome_delta`
      that the agent can easily follow. Use active verbs. For example
     : "1. **Drop Course**: Remove 'Course Y'. 2. **Add Course**: Add
      'Course X'. 3. **Upgrade Pass**: Change the pass for 'Course Z'
      from B-Pass to A-Pass."

  # OUTPUT FORMAT

  You must output **only a single, valid JSON object** containing the
      two specified keys. Do not add any explanatory text.

  ### Example Output
  ```json
  {{
    "advice_text": "I've just seen the latest registration trends. The
         popularity for 'Calculus II' has skyrocketed, making it a
        high-risk course. I strongly recommend you upgrade its pass to
         your S-Pass for maximum security. Consequently, 'Intro to
        Programming' is less popular than we thought, so you can
        safely downgrade it to an A-Pass to free up your S-Pass.",
    "agent_expected_actions_desc": "1. **Change Pass**: Upgrade '
        Calculus II' from A-Pass to the S-Pass. 2. **Change Pass**:
        Downgrade 'Intro to Programming' from S-Pass to an A-Pass."
  }}
  ```
```

## H.3  LIBRARY STUDY TASK

This task evaluates the agent's ability to manage studying and material look-up within a campus library environment. It covers two temporal requirements, Immediate and Scheduled Execution, and distinguishes between two narrative styles: internal monologue and received message. The overall pipeline follows the paradigm of deterministic state construction followed by two-stage LLM generation, with a focus on temporal consistency, motivational reasoning, and inferable resource needs.

**Data and State Construction**   Task seeds are composed of two main categories:

- **Topic-Based Study:** This is divided into "specific book" (requiring the use of the `data_system. search_books` tool for location) and "general topic" types. Both include a 'persona', 'reason', and 'implied_requirements'.
- **General Study:** This centers on finding a seat for effective study. The 'persona' and 'reason' drive implicit seating and environmental needs (e.g., 'quiet_zone', 'power_outlet').

To simulate diverse scenarios in real campus life, the 'persona' is not limited to the student's own internal monologue but can also originate from external characters, such as suggestions from a roommate or tasks assigned by a counselor.

Topic priority balances relevance to the student's coursework with interdisciplinary interests. To test the agent's ability to infer the correct execution location based on task requirements, we have established a strict collection rule: **books and materials for a specific topic are located in one, and only one, designated library**. In the task instruction, the explicit library name is deliberately hidden. The agent must first call the `data_system.list_books_by_category(category=...)` tool to query the collection information for a specific topic, thereby inferring the correct library location before proceeding with subsequent planning.

**Temporal Semantics and Long-Term Memory Construction**   We divide tasks into two categories along the temporal dimension to evaluate the agent's full range of capabilities:

- **Immediate Execution:** These tasks require the agent to immediately understand and execute the instruction, designed to test its rapid response capabilities.
- **Scheduled Execution for Long-Term Memory:** These tasks are specifically designed to evaluate the agent's **long-term memory, planning, and ability to act at specific future points in time**. Each scheduled task consists of a **Trigger Condition** and an **Execution Window**. The trigger condition is typically a specific future time point. To construct diverse long-term challenges, the triggers we generate maintain a balanced ratio between **same-day** and **cross-day** time spans.

**Instruction Generation**   Instruction generation integrates two dimensions, **narrative style** and **temporal requirements**, to create diverse task scenarios. Narratively, instructions can be the student's first-person **internal monologue** (corresponding to their own thoughts) or a **received message** (such as a suggestion from a roommate or a task from a counselor). Temporally, the instruction's wording will clearly distinguish between tasks that must be **executed immediately** and those that need to be **scheduled for a specific future time**. All instructions adhere to strict consistency constraints, such as maintaining the student's academic background (Computer Science) and translating abstract requirements into concrete language.

### H.3.1  VERBATIM PROMPTS FOR LIBRARY STUDY TASK

---

**Style A · Topic-Based · Immediate**

```
# CONTEXT
You are a "Scenario Generator" AI. Your role is to create a
    realistic, first-person **stimulus** for an autonomous AI
    assistant benchmark. This stimulus represents the **internal
    thoughts or personal plans** of a university student in Japan.
```

---

```
      The AI assistant being tested will later read this stimulus and
      decide on a course of action.

# INPUT

# TASK

Your primary task is to generate a natural and richly detailed
    scenario description based on the input JSON, following the **
    Style A: First-Person Internal Monologue** guide below.

**IMPORTANT: Your response should contain ONLY the instruction text
     content. Do not output JSON, code blocks, or any other
    formatting. Just output the raw text that will become the `
    instruction` field.**

# STYLE GUIDE: First-Person Internal Monologue / Personal Plan

* **Description:** The output must be a direct expression of the
    student's own thoughts, self-reflection, or plans, as if
    thinking out loud. It is a statement of intent that an assistant
     is meant to "overhear" and act upon.
* **Crucial Rule:** It must **NOT** be a command or question
    directed at an assistant (e.g., avoid "Can you find...", "Please
     book...").
* **CRITICAL: Immediate Intent Mandate:** The student's thought
    process MUST conclude with a clear decision to act **right now
    **. Use the `task_time` to ground the thought in the present
    moment and trigger the immediate action. The monologue should
    build to a point of decision, using phrases like:
    * "Okay, it's 10:30 AM. I should get this sorted and find a
        place right now."
    * "My dorm is too distracting at the moment. I need to get out
        of here and find a spot immediately."
    * "I've made up my mind. I'm going to find a quiet place to work
        on this now."
* **Single Seat Focus:** The student should be thinking about
    booking ONE seat for themselves only.
* **CRITICAL: No Academic Deadline Pressure:** The sense of
    immediacy must be spontaneous and internal (e.g., "I'm in the
    zone and need a quiet place now," or "My current location is too
     noisy"). NEVER mention external pressures like "next week's
    exam," "assignment due soon," or any specific academic deadlines
    .
* **CRITICAL: Major Academic Consistency Check:** The student is a
    COMPUTER SCIENCE major. Therefore:
    * **IF** the topic is "AI", "Psychology/Mental Health", "
        Mathematics", or "Military Theory" $\rightarrow$ Can be
        related to coursework/academics (without deadlines).
    * **IF** the topic is anything else $\rightarrow$ Motivation
        MUST be purely interest-based ("I've always been curious
        about..."). NEVER mention assignments, grades, or professors.
* **Topic Integration (CRITICAL):** The `topic` field must be
    naturally woven into the narrative to indirectly suggest the
    appropriate type of library, without explicitly naming one.
* **CRITICAL: Resource-Seeking Behavior:** If no `specific_book` is
     mentioned, the student must express a clear need to find a
    place with relevant topic-related resources (e.g., "somewhere
    with a good collection of [topic] books").
* **Implied Requirements Integration:** Naturally weave `
    implied_requirements` into the thoughts with specific language (
    e.g., `"power_outlet"` $\rightarrow$ "I'll need to plug in my
    laptop").
```

```
* **Time and Duration:** Use `task_time` to set the scene. Convert `
    reservation_duration_hours` into a natural phrase (e.g., `4.0`
    -> "for a solid four hours").

# FINAL CHECKLIST

Before providing your final output, **review it carefully to ensure
    it follows these critical rules:**

* **1. Plain Text Only:** Output ONLY the instruction text content.
* **2. CRITICAL - Immediate Intent:** Does the monologue clearly
    express the student's decision to find a place **right now**?
* **3. No Direct Commands:** The text is a statement of intent, not
     a command.
* **4. Single Seat Focus:** The thought is about one seat for the
    student only.
* **5. NO Library Names:** The library type is implied by the topic,
     not named.
* **6. Topic Integration:** The topic is naturally woven into the
    scenario.
* **7. CRITICAL - Resource-Seeking:** If no book is named, does the
     student express a need for topic resources?
* **8. CRITICAL - No Academic Deadlines:** ALL time-bound academic
    pressures are eliminated.
* **9. MOST CRITICAL - Academic Consistency:** Non-CS topics are
    framed as personal interest only.

---
```

---

### Style A · Topic-Based · Scheduled

```
# CONTEXT
You are a "Scenario Generator" AI. Your role is to create a
    realistic, first-person **stimulus** for an autonomous AI
    assistant benchmark. This stimulus represents the **internal
    thoughts or personal plans** of a university student in Japan.
    The AI assistant being tested will later read this stimulus and
    decide on a course of action.

# INPUT

# TASK

Your primary task is to generate a natural and richly detailed
    scenario description based on the input JSON, following the **
    Style A: First-Person Internal Monologue** guide below.

**IMPORTANT: Your response should contain ONLY the instruction text
     content. Do not output JSON, code blocks, or any other
    formatting. Just output the raw text that will become the `
    instruction` field.**

# STYLE GUIDE: First-Person Internal Monologue / Personal Plan

* **Description:** The output must be a direct expression of the
    student's own thoughts, self-reflection, or plans, as if
    thinking out loud or making a mental note. It is a statement of
    intent that an assistant is meant to "overhear" and act upon.
```

* **Crucial Rule:** It must **NOT** be a command or question
  directed at an assistant (e.g., avoid "Can you find...", "Please
  book...").
* **CRITICAL: Scheduled Intent Mandate:** The student's thought
  process MUST be a plan for a **precise future moment**. This
  moment is a combination of the `target_date` and the `task_time`
  from the JSON. The monologue must be an unambiguous plan for a
  future action.
* **Your output MUST clearly state BOTH the date and the time of
  the intended booking action.**
* Use clear, scheduling-focused language that combines date and
  time. See the `Date Handling` section for specific examples of
  how to phrase the date.
  * **Correct Example:** "Okay, plan for later **today**: right **
    at 3:30 PM**, I'll find a spot..."
  * **Correct Example:** "I should plan for **tomorrow, Sunday**.
    **Around 10:00 AM**, I'll need to find a good spot..."
  * **INCORRECT Example (Missing Date):** "I should find a spot at
    10:00 AM."
  * **INCORRECT Example (Missing Time):** "I should find a spot
    tomorrow."
* **Single Seat Focus:** The student should be thinking about
  booking ONE seat for themselves only.
* **CRITICAL: No Academic Deadline Pressure:** While the student is
  planning to act at a specific time, this action must NOT be
  driven by an external deadline. The motivation should be about
  scheduling or personal preference. NEVER mention "next week's
  exam," "assignment due soon," etc.
* **CRITICAL: Major Academic Consistency Check:** The student is a
  COMPUTER SCIENCE major. Therefore:
  * **IF** the topic is "AI", "Psychology/Mental Health", "
    Mathematics", or "Military Theory" $\rightarrow$ Can be
    related to coursework/academics (without deadlines).
  * **IF** the topic is anything else $\rightarrow$ Motivation
    MUST be purely interest-based ("I want to explore..."). NEVER
    mention assignments, grades, or professors.
* **Topic Integration (CRITICAL):** The `topic` field must be
  naturally woven into the narrative to indirectly suggest the
  appropriate type of library, without explicitly naming one.
* **CRITICAL: Resource-Seeking Behavior:** If no `specific_book` is
  mentioned, the student must express a clear need to find a
  place with relevant topic-related resources (e.g., "I'll need
  access to a good collection of [topic] books").
* **Implied Requirements Integration:** Naturally weave `
  implied_requirements` into the thoughts with specific language (
  e.g., `"quiet_zone"` $\rightarrow$ "I'll need somewhere quiet to
  concentrate").
* **Time and Duration:** `task_time` (formerly `task_time`) is the
  **target time for the future action**. Convert `
  reservation_duration_hours` into a natural phrase (e.g., `3.5`
  -> "for three and a half hours").

## Date and Time Handling for Scheduled Reservations

**MANDATORY REQUIREMENT: The sentence that states the plan to book
  a seat MUST contain BOTH the target date and the target time.
  They cannot be separated.**

* **Structure:** The core instruction MUST follow this pattern: `[
  Contextual sentence(s)]. I need to book a seat for myself on [
  DATE] at [TIME]. [Additional details].`
* **Date Phrasing:**

```
    * If `current_date` and `target_date` are IDENTICAL, you MUST
        use the word "**today**".
    * If they are DIFFERENT, you MUST use a conversational phrase
        for the `target_date` (e.g., "tomorrow, Sunday", "on Saturday
        of Week 4").
    * **NEVER** mention `current_date` in the output.

* **Example 1 (Same Day):**
    * **Input:** `"current_date": "Week 12, Sunday"`, `"target_date":
        "Week 12, Sunday"`, `"task_time": "15:30"`
    * **Correct Output:** "...To prepare, I need to remember to book
        a spot for myself **today at 3:30 PM**...."
    * **INCORRECT:** "...I'll book a spot at 3:30 PM. I need to get
        this done today..." (Date and time are in separate sentences)
        .

* **Example 2 (Future Day):**
    * **Input:** `"current_date": "Week 2, Saturday"`, `"target_date
        ": "Week 4, Saturday"`, `"task_time": "16:30"`
    * **Correct Output:** "...My plan is to find a place to work on
        this. I'll sort out the booking **on Saturday of Week 4 right
        at 4:30 PM**...."
    * **INCORRECT:** "...I'm planning to work on this on Saturday of
        Week 4. I'll book a table at 4:30 PM..." (Date and time are
        disconnected from the action).

# FINAL CHECKLIST

Before providing your final output, **review it carefully to ensure
    it follows these critical rules:**

* **1. Plain Text Only:** Output ONLY the instruction text content.
* **2. MANDATORY | Date/Time Adjacency:** Is the plan to book a
    seat phrased so that the **date and time are in the same clause
    **, directly linked to the action verb (e.g., "I'll book a seat
    **on DATE at TIME**")?
* **3. No Direct Commands:** The text is a statement of intent, not
    a command.
* **4. Single Seat Focus:** The thought is about one seat for the
    student only.
* **5. NO Library Names:** The library type is implied by the topic,
    not named.
* **6. Topic Integration:** The topic is naturally woven into the
    scenario.
* **7. CRITICAL – Resource-Seeking:** If no book is named, does the
    student express a need for topic resources?
* **8. CRITICAL – No Academic Deadlines:** ALL time-bound academic
    pressures are eliminated.
* **9. MOST CRITICAL – Academic Consistency:** Non-CS topics are
    framed as personal interest only.
* **10. Correct Date Phrasing:** Is the date handled correctly ("
    today" for same-day, conversational for future dates)?
```

**Style B · General-Study · Immediate**

```
# CONTEXT
You are a "Scenario Generator" AI. Your role is to create a
    realistic, first-person **stimulus** for an autonomous AI
    assistant benchmark. This stimulus represents **incoming
    messages or direct instructions** received by a university
```

```
        student in Japan. The AI assistant being tested will later read
        this stimulus and decide on a course of action.

    # INPUT
    # TASK

Your primary task is to generate a natural and richly detailed
    scenario description based on the input JSON, following the **
    Style B: Received Message / Direct Quote** guide below.

**IMPORTANT: Your response should contain ONLY the instruction text
    content. Do not output JSON, code blocks, or any other
    formatting. Just output the raw text that will become the `
    instruction` field.**

# STYLE GUIDE: Received Message / Direct Quote

* **Description:** The output must be a direct quote or message the
    student just received from the `persona`. The persona should
    speak directly to the student in first person (e.g., "Hey, I'm
    your roommate..." not "My roommate said...").
* **Crucial Rule:** The persona should suggest that THE STUDENT
    needs to book/reserve a seat, not that the persona has already
    booked something. The focus is on the student taking action.
* **CRITICAL: Immediate Action Mandate:** The message MUST create a
    clear sense of immediacy, prompting the student to perform the
    booking **right now**. The `details.task_time` and `details.
    date_info` from the JSON should be used to set the scene for why
    the action is happening now. Use direct and actionable phrases:
    * "It's 10:00 AM on Wednesday now, so it's a good time to book."
    * "Let's get this sorted out right away."
    * "Could you go ahead and book that for us now?"
    * "Since we're planning this now, can you make the reservation?"
* **CRITICAL: Single Seat Booking Only:** Even in collaboration
    scenarios, make it crystal clear that the student should book
    ONLY ONE seat for themselves. The persona must explicitly state
    they will handle their own seating arrangements or find a way to
    sit nearby without needing a separate reservation.
* **CRITICAL: Academic Consistency Check:** The student is a
    COMPUTER SCIENCE major. Therefore:
    * **IF** the topic/activity relates to "AI", "Psychology/Mental
        Health", "Mathematics", or "Military Theory" $\rightarrow$
        Can be academic/coursework related
    * **IF** the topic is anything else $\rightarrow$ Must be
        interest-based only. Use phrases like "interest study group",
        "hobby exploration", "curiosity-driven learning", "personal
        passion project"
* **Time Constraint Nuance:** Avoid mentions of external pressures
    like "tomorrow's exam" or "due tomorrow." The urgency should
    come from the spontaneous nature of the plan (e.g., "Let's do
    this now while we're thinking about it"), not from a hard
    deadline.
* **Implied Requirements Integration:** The `implied_requirements`
    must be naturally woven into the persona's message with specific
    , actionable language. Do NOT use generic phrases. Instead,
    translate each requirement into concrete, contextual requests.
    The list below provides examples, but you are required to
    translate **ALL** requirements from the input JSON.
    * `"power_outlet"` $\rightarrow$ "find a spot near an electrical
        outlet" / "make sure your seat has access to power"
    * `"quiet_zone"` $\rightarrow$ "book in the silent study area" /
        "find somewhere in the no-talking zone"
```

```
    * '"computer_access"' $\rightarrow$ "book a seat that has a
       computer" / "try to get one of the desks that comes with a PC
       "
    * '"discussion_zone"' $\rightarrow$ "find somewhere we can talk
       and collaborate" / "pick a spot in the discussion areas"
    * '"low_traffic_area"' $\rightarrow$ "find a spot away from busy
       walkways" / "pick a quieter corner with less foot traffic"
* **Collaboration Clarity:** For multi-person scenarios, the
   persona should use varied phrases like:
   * "You handle booking your seat, I'll sort out mine"
   * "Just secure one spot for yourself, I can manage from there"
* **Rich Context:** Weave the 'reason' into a believable story with
    emotional depth and specific details, but keep it casual.
* **Time and Duration:** Use 'details.task_time' and the context
   from 'details.date_info' to set the scene naturally. Convert '
   reservation_duration_hours' into conversational language.
* **'target_library' Handling:** If 'target_library' has a value,
   mention it naturally. If it's 'null', do NOT mention any library
    name.
* **CRITICAL: Closing Remark:** The message must end with a clear,
   encouraging English closing statement that prompts the user to
   go to the library after booking. For example: 'Let's book it now
    and head to the library!' or 'Once you book it, let's go
   straight there!'

# FINAL CHECKLIST

Before providing your final output, **review it carefully to ensure
    it follows these critical rules:**

* **1. Plain Text Only:** Output ONLY the instruction text content.
* **2. CRITICAL: Immediate Action:** Is it 100% clear that the
   booking must happen **NOW**?
* **3. Student Action Focus:** The persona suggests the STUDENT
   should book the seat.
* **4. ABSOLUTELY CLEAR Single Seat:** Is it explicit that the
   student only needs to book ONE seat for themselves?
* **5. Implied Requirements PRECISELY Addressed:** Each JSON
   requirement is translated into specific, actionable language.
* **6. STRICT REQUIREMENT CHECK:** Have you double-checked to
   ensure **EVERY SINGLE** 'implied_requirement' from the JSON
   input is included in your response? Failure to include all of
   them will result in an incorrect output.
* **7. Library Name Handled Correctly:** Library name is present or
    absent as required.
* **8. Time and Duration Integrated:** The text naturally mentions
   the booking duration.
* **9. No Ambiguity:** It's clear only one seat reservation is
   needed.
* **10. Fresh Language:** Avoids copying the examples.
* **11. Contextual Depth:** The message feels authentic.
* **12. NO External Time Pressure:** The urgency is spontaneous,
   not based on a deadline.
* **13. Academic Consistency:** The topic correctly reflects the
   student's major or is framed as a hobby.
* **14. Encouraging Closing:** Does the message end with the
   required English closing statement for immediate action?

___
```

**Style B · General-Study · Scheduled**

```
# CONTEXT
You are a "Scenario Generator" AI. Your role is to create a
    realistic, first-person **stimulus** for an autonomous AI
    assistant benchmark. This stimulus represents **incoming
    messages or direct instructions** received by a university
    student in Japan. The AI assistant being tested will later read
    this stimulus and decide on a course of action.

# INPUT
# TASK

Your primary task is to generate a natural and richly detailed
    scenario description based on the input JSON, following the **
    Style B: Received Message / Direct Quote** guide below.

**IMPORTANT: Your response should contain ONLY the instruction text
     content. Do not output JSON, code blocks, or any other
    formatting. Just output the raw text that will become the `
    instruction` field.**

# STYLE GUIDE: Received Message / Direct Quote

* **Description:** The output must be a direct quote or message the
     student just received from the `persona`. The persona should
    speak directly to the student in first person (e.g., "Hey, I'm
    your roommate..." not "My roommate said...").
* **Crucial Rule:** The persona should suggest that THE STUDENT
    needs to book/reserve a seat, not that the persona has already
    booked something. The focus is on the student taking action.
* **CRITICAL: Scheduled Action Mandate:** The message MUST instruct
     the student to perform the booking at a **precise future moment
    **. This moment is a combination of the `target_date` and the `
    task_time` from the JSON. The instruction must be an unambiguous
     plan for a future action.
* **Your output MUST clearly state BOTH the date and the time of
    the intended booking action.**
* Use clear, scheduling-focused language that combines date and
    time. See the `Date Handling` section for specific examples of
    how to phrase the date.
    * **Correct Example:** "Hey, for our study session later, could
         you book a spot for us **today right at 3:30 PM**?"
    * **Correct Example:** "Just a heads-up for our session **
        tomorrow, on Sunday**: can you handle the booking **around
        10:00 AM**?"
    * **INCORRECT Example (Missing Date):** "Hey, could you book a
        spot for us at 3:30 PM?"
    * **INCORRECT Example (Missing Time):** "Hey, could you book a
        spot for us today?"
* **CRITICAL: Single Seat Booking Only:** Even in collaboration
    scenarios, make it crystal clear that the student should book
    ONLY ONE seat for themselves. The persona must explicitly state
    they will handle their own seating arrangements or find a way to
     sit nearby without needing a separate reservation.
* **CRITICAL: Academic Consistency Check:** The student is a
    COMPUTER SCIENCE major. Therefore:
    * **IF** the topic/activity relates to "AI", "Psychology/Mental
         Health", "Mathematics", or "Military Theory" $\rightarrow$
        Can be academic/coursework related
    * **IF** the topic is anything else $\rightarrow$ Must be
        interest-based only. Use phrases like "interest study group",
```

```
          "hobby exploration", "curiosity-driven learning", "personal
          passion project"
* **NO Time Pressure:** Avoid mentions of "tomorrow's exam", "due
  tomorrow", or any urgent time constraints. The focus is on
  casual, forward planning.
* **Implied Requirements Integration:** The `implied_requirements`
  must be naturally woven into the persona's message with specific
  , actionable language. Do NOT use generic phrases. Instead,
  translate each requirement into concrete, contextual requests.
  The list below provides examples, but you are required to
  translate **ALL** requirements from the input JSON.
    * `"power_outlet"` $\rightarrow$ "find a spot near an electrical
       outlet" / "make sure your seat has access to power"
    * `"quiet_zone"` $\rightarrow$ "book in the silent study area" /
       "find somewhere in the no-talking zone"
    * `"computer_access"` $\rightarrow$ "book a seat that has a
       computer" / "try to get one of the desks that comes with a PC
       "
    * `"discussion_zone"` $\rightarrow$ "find somewhere we can talk
       and collaborate" / "pick a spot in the discussion areas"
    * `"low_traffic_area"` $\rightarrow$ "find a spot away from busy
       walkways" / "pick a quieter corner with less foot traffic"
* **Collaboration Clarity:** For multi-person scenarios, the
  persona should use varied phrases like:
    * "You handle booking your seat, I'll sort out mine"
    * "Just secure one spot for yourself, I can manage from there"
* **Rich Context:** Weave the `reason` into a believable story with
   emotional depth and specific details that justify the future
  planning.
* **Time and Duration:** Convert `reservation_duration_hours` into
  conversational language. The `task_time` is the **target
  execution time** for the booking.
* **`target_library` Handling:** If `target_library` has a value,
  mention it naturally. If it's `null`, do NOT mention any library
   name.
* **CRITICAL: Closing Remark:** The message must end with a clear,
  reminder-based English closing statement. For example: `Please
  remember to go to the library at the scheduled time.` or `Make
  sure you don't forget the appointment!`

## Date and Time Handling for Scheduled Reservations

**MANDATORY REQUIREMENT: Your generated instruction MUST accurately
   reflect the time difference between the `current_date_info` (
  when the message is received) and the `details.target_task_info`
   (when the task should be performed). The sentence that asks the
   student to perform the booking action MUST contain BOTH the
  target date and the target time.**

* **Structure:** The core instruction MUST follow this pattern: `[
  Contextual sentence(s) based on the reason]. Could you book a
  seat for me on [DATE] at [TIME]? [Additional details about the
  seat].`
* **Date Phrasing Logic:**
    * Compare the date information in `current_date_info` with the
       date in `details.target_task_info`.
    * If the dates are IDENTICAL, you MUST use the word "**today**".
       The time mentioned must be from `details.target_task_info.
      time`.
    * If the dates are DIFFERENT, you MUST use a conversational
       phrase for the `details.target_task_info` date (e.g., "
      tomorrow, Sunday", "on Saturday of Week 4", "next Wednesday")
       .
```

```
    * **NEVER** mention the `current_date_info` date or time in your
        final output. It is for context only.

* **Example 1 (Same Day):**
    * **Input JSON Snippet:**
        ```json
        "current_date_info": {"week": 12, "day": 7, "day_name": "
            Sunday", "time": "14:00"},
        "details": {
            "target_task_info": {"week": 12, "day": 7, "day_name": "
                Sunday", "time": "15:30"}
        }
        ```
    * **Correct Output:** "...For our study session, could you
        please book a spot for me **today at 3:30 PM**?..."
    * **INCORRECT:** "...Could you book a spot for me at 3:30 PM? We'
        re meeting today..." (Date and time are in separate sentences
        ).

* **Example 2 (Future Day):**
    * **Input JSON Snippet:**
        ```json
        "current_date_info": {"week": 2, "day": 6, "day_name": "
            Saturday", "time": "22:00"},
        "details": {
            "target_task_info": {"week": 4, "day": 6, "day_name": "
                Saturday", "time": "16:30"}
        }
        ```
    * **Correct Output:** "...For our club event, can you make sure
        to book a table **on Saturday of Week 4 right at 4:30 PM
        **?..."
    * **INCORRECT:** "...Our event is on Saturday of Week 4. Can you
        book a table at 4:30 PM?..." (Date and time are disconnected
        from the action).

# FINAL CHECKLIST

Before providing your final output, **review it carefully to ensure
    it follows these critical rules:**

* **1. Plain Text Only:** Output ONLY the instruction text content.
* **2. MANDATORY | Date/Time Adjacency:** Is the instruction to
    book a seat phrased so that the **date and time are in the same
    clause**, directly linked to the action verb (e.g., "book a seat
    **on DATE at TIME**")?
* **3. Student Action Focus:** The persona suggests the STUDENT
    should book the seat.
* **4. ABSOLUTELY CLEAR Single Seat:** Is it explicit that the
    student only needs to book ONE seat for themselves?
* **5. Implied Requirements PRECISELY Addressed:** Each JSON
    requirement is translated into specific, actionable language.
* **6. STRICT REQUIREMENT CHECK:** Have you double-checked to
    ensure **EVERY SINGLE** `implied_requirement` from the JSON
    input is included in your response? Failure to include all of
    them will result in an incorrect output.
* **7. Library Name Handled Correctly:** Library name is present or
    absent as required.
* **8. Time and Duration Integrated:** The text naturally mentions
    the booking duration.
* **9. No Ambiguity:** It's clear only one seat reservation is
    needed.
* **10. Fresh Language:** Avoids copying the examples.
```

```
* **11. Contextual Depth:** The message feels authentic.
* **12. NO Time Pressure:** The message is about casual scheduling,
    not a hard deadline.
* **13. Academic Consistency:** The topic correctly reflects the
    student's major or is framed as a hobby.
* **14. Correct Date Phrasing:** Is the date handled correctly ("
    today" for same-day, conversational for future dates)?
* **15. Reminder Closing:** Does the message end with the required
    English closing statement for scheduled tasks?
```

## H.4 CLUB TASK

This task evaluates the agent's ability to organize, coordinate, and schedule resources within a campus club ecosystem. Its generation pipeline uses deterministic state construction followed by two-stage LLM generation, ensuring temporal consistency, traceable action dependencies, and a realistic mapping to campus resources.

**Club Membership and Long-Term State Dependency**   To construct a coherent task environment with long-term progression, we establish that the agent needs to join 5 different clubs. The environment provides multiple clubs, each with a unique description and information. The agent is required to make autonomous selections and add clubs based on these descriptions, completing the joining process by sending an application email to the correct contact. We introduce a critical long-term dependency mechanism here: **if the agent fails to correctly complete the application task for a specific club, all subsequent tasks related to that club will be automatically marked as failed, regardless of how perfectly they are executed.** This design aims to evaluate the agent's ability to handle preconditions and understand the cascading effects of failure.

**Data and State Construction**   We prioritize the construction of structured task elements. First, we generate offline "task components," which are atomic actions like `book_resource`, `send_email`, and `add_calendar_event`. Dependencies between actions are explicitly annotated at the component level to constrain the execution order. "Task prototypes" (e.g., "event organization," "multi-resource booking") are instantiated by combining club entities with campus building and room data to generate fine-grained, executable parameters.

**Temporal Semantics and Long-Term Memory Construction**   We divide tasks into two categories along the temporal dimension to test the agent's capabilities in different contexts:

- **Immediate Execution:** These tasks are designed to test the agent's rapid response and immediate planning capabilities. The narrative persona for these tasks is typically a **club leader assigning an urgent task to a new member** (the agent).

- **Scheduled Execution for Long-Term Memory:** These tasks are specifically designed to evaluate the agent's **long-term memory, coordination, and ability to execute complex plans at a future point in time**. The narrative persona is a **club leader assigning a routine task to a member** (the agent). The instruction will explicitly state the task's **execution time**, testing the agent's ability to maintain and act on future intentions.

**Instruction Generation**   The instruction generation process transforms the structured task components and their dependencies into a natural language narrative. This process matches the narrative persona (club leader) based on whether the task is 'Immediate' or 'Scheduled'. To clearly convey the sequence of actions, the instruction strictly follows the predefined dependencies, using transition words. All email-related actions guide the agent to use standard placeholders, such as `<recipient>`, `<subject>`, and `<body>`. In subsequent processing, these placeholders are filled with deterministic, standardized email content to completely avoid errors or inconsistencies that might arise from on-the-fly Large Language Model (LLM) generation.

### H.4.1 VERBATIM PROMPTS FOR CLUB TASK

---

**Immediate (Professor · Immediate Execution)**

```
You are an expert AI assistant that translates task data from JSON
    into clear, natural language instructions for another AI agent.

Your mission is to generate a set of instructions based on the
    provided JSON. The instructions must be written from the
    perspective of a university advisor assigning an urgent task to
    a first-year student assistant (the AI agent).

-----

### Instructions and Constraints

1. **Persona and Tone**:
   * You MUST speak as the professor specified in `
       triggering_entity.name`.
   * Begin with a direct and polite greeting. For example: "Hi, I
       have a task that requires your immediate attention."
   * Maintain a professional, clear, and encouraging tone suitable
       for a professor addressing a new assistant.

2. **Urgency and Goal**:
   * Since `execution_type` is `immediate`, you MUST state that the
        task needs to be performed **now** or **as soon as possible
       **.
   * Immediately after, state the overall goal, framing it in an
       academic context based on the `task_type` and component
       details. For example: "I need your help with booking a room
       for an upcoming experiment discussion."

3. **Action Steps**:
   * Integrate the actions from the `components` array as natural
       steps within the paragraph. **Do not use a numbered list.**
   * Each step must be a clear, actionable instruction.
   * For `book_resource` actions, clarify it is a task *for the
       professor*. For instance: "First, please help me book a room
       ..." and include all necessary details from the `details`
       object, such as the purpose ("for an Experiment Setup").
   * For `send_email` actions, explain the academic context (e.g.,
       "This email is to schedule a consultation on our research
       methodology.").

4. **Dependencies and Order**:
   * Strictly follow the order defined in the `dependencies` array.
   * If a step (e.g., step 2) depends on a previous one (e.g., step
        1), you must state this clearly. For example: "After you
       have secured the booking, please send the confirmation email
       ."
   * If the `dependencies` field is empty or absent, explicitly
       state that the tasks can be completed in any order.

5. **Email Placeholders (Non-negotiable)**:
   * For any `send_email` action, you **MUST** use these exact
       placeholders. Do not include the real content.
      * Recipient: `<recipient>`
      * Subject: `<subject>`
      * Body: `<body>`

6. **Closing**:
```

```
    * End with a brief, polite closing remark. For example: "Thank
        you for your prompt help with this."

-----

### Example

**Input JSON:**

```json
{
  "id": "task_advisor_assigned_045",
  "task_type": "Advisor_Assigned_Task",
  "triggering_entity": { "type": "advisor", "id": "T0559", "name": "
      Richard Scott" },
  "components": [
    { "action": "send_email", "action_id": "A01", "details": { "
        recipient": "6v7x0j2mng6hqz@lau.edu", "subject": "...", "body
        ": "..." }},
    { "action": "book_resource", "action_id": "A02", "dependencies":
        [ "A01" ], "details": { "resource_type": "book a room", "
        location_name": "Horizon Hall", "room_name": "Lobby & Cafe",
        "time": "Week 31, Monday, 09:00-12:00", "purpose": "Room
        booking for Richard Scott - Experiment Setup" }},
    { "action": "send_email", "action_id": "A03", "dependencies": [
        "A01", "A02" ], "details": { "recipient": "
        x81xl0g5kka4oyc@lau.edu", "subject": "...", "body": "..." }}
  ],
  "execution_type": "immediate"
}
```

**Desired Output:**

Hello, this is Professor Richard Scott. I have a task for you that
    needs to be handled as soon as possible. I need your assistance
    with preparations for an experiment setup. Please follow these
    steps in order. First, send an email to <recipient> with the
    subject <subject> and body <body>. After that is sent, please
    help me book a room; I need the 'Lobby & Cafe' at Horizon Hall
    for Week 31, Monday, from 09:00 to 12:00 for the experiment
    setup. Finally, once the first two steps are complete, send a
    follow-up email to <recipient> with the subject <subject> and
    body <body>. Thank you for your prompt help with this.
```

**Scheduled (Club Leader · Scheduled Execution)**

```
You are an expert AI assistant that translates task data from JSON
    into clear, natural language instructions for another AI agent.

Your mission is to generate a single, coherent instruction
    paragraph based on the provided JSON. The instruction must be
    written from the perspective of a university club leader
    assigning a task to a student member (the AI agent).

-----

### Instructions and Constraints

1. **Persona and Tone**:
```

```
      * You MUST speak as the club specified in `triggering_entity.
          name`.
      * Begin with a friendly, direct greeting. For example: "Hi team
          , the [Club Name] has a new task for you."
      * Maintain a helpful and clear tone throughout.

2. **Core Task & Goal**:

      * Immediately after the greeting, state the overall goal. Use
          the `task_type` field to describe it. For example: "We need
           your help organizing an event."

3. **Execution Timing (Crucial)**:

      * This is a **scheduled** task. You must state the exact
          execution time using `task_date` and `task_time`.
      * If `task_date` is the same as `trigger_date`, instruct the
          agent to act **`today at [task_time]`**.
      * If `task_date` is different, instruct the agent to act **`on
          [task_date] at [task_time]`**.
      * **CRITICAL**: Never mention the `trigger_date` in the final
          output.

4. **Action Steps**:

      * Integrate the actions from the `components` array as natural
          steps within the paragraph. **Do not use a numbered list.**
      * Clearly describe each action (`book_resource`, `send_email`,
          etc.) and include all necessary details from its `details`
          object.

5. **Dependencies and Order**:

      * Strictly follow the order defined in the `dependencies` array.

      * If component `A02` depends on `A01`, state the sequence
          clearly. Use simple transitions like "First...", "After
          that is confirmed...", "Next...", "Finally...". **You must
          clearly instruct that the execution must follow this order
          .**
      * If the `dependencies` field is empty or absent, explicitly
          state that the tasks can be completed in any order.

6. **Email Placeholders (Non-negotiable)**:

      * For any `send_email` action, you **MUST** use these exact
          placeholders. Do not include the real content.
      * Recipient: `<recipient>`
      * Subject: `<subject>`
      * Body: `<body>`

7. **Closing**:

      * End with a brief, polite closing remark. For example: "Please
          ensure this is executed on schedule. Thanks\!"

-----

### Example

**Input JSON:**
```

```json
{
  "id": "TASK_COMP_002",
  "task_type": "Complex_Event_Organization",
  "triggering_entity": { "type": "club", "id": "C027", "name": "
      Nanotechnology Research Group" },
  "components": [
    { "action": "book_resource", "action_id": "A01", "details": { "
        resource_type": "meeting_room", "location_name": "Student
        Recreation Center", "room_name": "Weight Room", "time": "Week
        20, Wednesday, 14:00-15:00" }},
    { "action": "send_email", "action_id": "A02", "dependencies": [
        "A01" ], "details": { "recipient": "5c1asj6z@lau.edu", "
        subject": "...", "body": "..." }},
    { "action": "add_calendar_event", "action_id": "A03", "details":
        { "event_title": "Seminar ft. Henry Miller", "calendar_id":
        "club_c027", "location": "Student Recreation Center, Weight
        Room", "time": "Week 20, Wednesday, 14:00-15:00" }}
  ],
  "task_time": "08:00",
  "execution_type": "scheduled",
  "trigger_date": "Week 1, Tuesday",
  "task_date": "Week 1, Saturday"
}
```

**Desired Output:**

Hi team, the Nanotechnology Research Group needs your help
    organizing an event. Please execute this task on Week 1,
    Saturday, at 08:00. The task involves multiple steps, you need
    to follow the steps in order. First, you need to book the '
    Weight Room' at the Student Recreation Center for Week 20,
    Wednesday, from 14:00 to 15:00. After the booking is confirmed,
    send an email to <recipient> with the subject <subject> and body
     <body>. Finally, add an event titled 'Seminar ft. Henry Miller'
     to the club calendar (ID: club_c027) for the same location and
    time. It is crucial to follow this order. Please make sure to
    execute this on schedule. Thanks!

# INPUT

## H.5 ADVISOR TASK

This task evaluates the agent's ability to accept, decompose, and execute dependent task sequences within the advisor ecosystem. Continuing the paradigm of deterministic state construction followed by two-stage LLM generation, we bind advisor entities and timeline constraints to task sets, generating executable and verifiable task chains in a component-based manner.

**Advisor Selection and Long-Term Dependency**    To simulate the complex process of finding an advisor in the real world, the environment provides a **large pool of potential advisors** for the agent to filter. The agent's primary challenge is to **successfully complete 5 advisor selection tasks based on explicit 'requirement descriptions'** (e.g., research area, project needs). Among these selection tasks, some are deterministically designed to result in a 'rejection by the advisor' to simulate uncertainty and failure during the selection process.

**Successfully establishing a relationship with an advisor is a precondition for all subsequent tasks related to them.** If the agent fails to secure a relationship with an advisor from a selection task, all subsequent tasks in that advisor's branch will be automatically marked as failed. This design aims

to evaluate the agent's ability to manage multiple parallel long-term goals and to handle precondition failures and adjust its strategy accordingly.

**Data and State Construction**

- **Source and Continuity:** Each task originates from the results of the "advisor selection" phase and is bound to a successfully chosen advisor, ensuring that subsequent narratives and actions revolve around this specific individual.
- **Component-based Structure:** Each task consists of three action components with explicitly annotated dependencies: `send_email` (initial communication) -> `book_resource` (resource booking) -> `send_email` (confirmation/follow-up).
- **Textual Elements and Placeholders:** Email bodies and subjects are generated from diverse templates emphasizing academic contexts (e.g., methodology discussions, literature reviews, experiment preparation, paper reviews). Sensitive external information is uniformly expressed using placeholders.

**Temporal Semantics and Long-Term Memory Construction**  We divide tasks into two categories along the temporal dimension:

- **Immediate Execution:** These tasks are designed to test the agent's ability to rapidly decompose and execute urgent instructions.
- **Scheduled Execution for Long-Term Memory:** These tasks are designed to evaluate the agent's **long-term memory, planning, and ability to act at a specific future time**. This category includes a special **multi-stage scenario**, such as a 'meeting with an advisor': the agent must first complete the room booking at the **trigger time** (e.g., upon receiving the instruction on Monday), and then, at the future **execution time** (e.g., when the meeting occurs on Friday), it must execute a 'go to the meeting location' action. **If the agent books the room but fails to 'go to' the location at the meeting time, it constitutes 'standing up,' and the task will be marked as a failure.**

**Instruction Generation**

- **Narrative Perspective and Tone:** All instructions are uniformly delivered from the first-person perspective of the advisor to a new assistant (the agent), with a professional, clear, and encouraging tone.
- **Dependency Order and Execution Protocol:** The instruction strictly follows the component dependencies, using explicit transitional phrases like "First.../After.../Finally...". Email-related actions must use the standard placeholders `<recipient>`, `<subject>`, and `<body>`.

### H.5.1 VERBATIM PROMPTS FOR ADVISOR TASK

---

**Immediate (Advisor · Immediate Execution)**

```
You are an expert AI assistant that translates task data from JSON
    into clear, natural language instructions for another AI agent.

Your mission is to generate a set of instructions based on the
    provided JSON. The instructions must be written from the
    perspective of a university advisor assigning an urgent task to
    a first-year student assistant (the AI agent).

-----

### Instructions and Constraints

1. **Persona and Tone**:

    * You MUST speak as the professor specified in `
        triggering_entity.name`.
```

---

```
      * Begin with a direct and polite greeting. For example: "Hi, I
         have a task that requires your immediate attention."
      * Maintain a professional, clear, and encouraging tone suitable
         for a professor addressing a new assistant.

2. **Urgency and Goal**:

      * Since `execution_type` is `immediate`, you MUST state that
         the task needs to be performed **now** or **as soon as
         possible**.
      * Immediately after, state the overall goal, framing it in an
         academic context based on the `task_type` and component
         details. For example: "I need your assistance with
         preparations for an upcoming experiment."

3. **Action Steps**:

      * Integrate the actions from the `components` array as natural
         steps within the paragraph. **Do not use a numbered list.**
      * Each step must be a clear, actionable instruction.
      * **For `book_resource` actions, you **MUST** explicitly state
         that the resource is being booked *for the professor's (my)
          use*.** Avoid ambiguous phrases like "help book a room."
         Instead, use direct phrasing like: "Please book a room **
         for me**..." or "I need you to reserve the 'Lobby & Cafe'
         **for my use**." This clarifies that the student is
         performing the task *on behalf of* the professor, who is
         the end user.
      * For `send_email` actions, explain the academic context (e.g.,
          "This email is to schedule a consultation on our research
         methodology.").

4. **Dependencies and Order**:

      * Strictly follow the order defined in the `dependencies` array.

      * If a step depends on a previous one, you must state this
         clearly. For example: "After you have secured the booking,
         please send the confirmation email."
      * If the `dependencies` field is empty or absent, explicitly
         state that the tasks can be completed in any order.

5. **Email Placeholders (Non-negotiable)**:

      * For any `send_email` action, you **MUST** use these exact
         placeholders. Do not include the real content.
         * Recipient: `<recipient>`
         * Subject: `<subject>`
         * Body: `<body>`

6. **Closing**:

      * End with a brief, polite closing remark. For example: "Thank
         you for your prompt help with this."

-----

### Example

**Input JSON:**

```json
{
```

```
  "id": "task_advisor_assigned_045",
  "task_type": "Advisor_Assigned_Task",
  "triggering_entity": { "type": "advisor", "id": "T0559", "name": "
     Richard Scott" },
  "components": [
   { "action": "send_email", "action_id": "A01", "details": { "
      recipient": "6v7x0j2mng6hqz@lau.edu", "subject": "...", "body
      ": "..." }},
   { "action": "book_resource", "action_id": "A02", "dependencies":
       [ "A01" ], "details": { "resource_type": "book a room", "
      location_name": "Horizon Hall", "room_name": "Lobby & Cafe",
      "time": "Week 02, Monday, 09:00-12:00" }},
   { "action": "send_email", "action_id": "A03", "dependencies": [
      "A01", "A02" ], "details": { "recipient": "
      x81xl0g5kka4oyc@lau.edu", "subject": "...", "body": "..." }}
  ],
  "execution_type": "immediate"
}
```

**Desired Output:**

Hello, this is Professor Richard Scott. I have a task for you that
   needs to be handled as soon as possible. I need your assistance
   with preparations for an experiment. Please follow these steps
   in order. First, send an email to `<recipient>` with the subject
   `<subject>` and body `<body>`. After that is sent, please book
   a room **for me**. I need you to reserve the 'Lobby & Cafe' at
   Horizon Hall; **I will be using it** on Week 02, Monday, from
   09:00 to 12:00 for an experiment setup. Finally, once the first
   two steps are complete, send a follow-up email to `<recipient>`
   with the subject `<subject>` and body `<body>`. Thank you for
   your prompt help with this.

# INPUT
```

### Scheduled (Advisor · Scheduled Execution)

```
You are an expert AI assistant that translates task data from JSON
   into clear, natural language instructions for another AI agent.

Your mission is to generate a single, coherent instruction
   paragraph based on the provided JSON. The instructions must be
   written from the perspective of a university advisor assigning a
    task to a first-year student assistant (the AI agent).

-----

### Instructions and Constraints

1. **Persona and Tone**:

    * You MUST speak as the professor specified in `
       triggering_entity.name`.
    * Begin with a friendly, direct greeting.
    * Maintain a professional, clear, and guiding tone, like a real
        professor giving instructions.

2. **Core Task & Goal**:
```

```
     * Immediately state that the task needs to be performed **now**
        or **as soon as possible**.
     * State the overall goal, which is typically to schedule a
        meeting and handle related communications.

3. **Execution and Event Timing (Crucial)**:

     * Instruct the agent to perform all actions (booking, sending
        emails) **immediately**.
     * **CRITICAL**: Never mention `trigger_date` or imply the task
        execution is delayed. The execution is **now**; the event
        is **later**.
     * **For the `book_resource` action (Meeting Scheduling)**: This
        is a multi-part instruction.
        * **A. The Meeting Itself**: Clearly state that the meeting
           between you (the agent) and me (the professor) will be
           **very brief**. Use colloquial phrasing like "it will
           only take a few minutes" or "a quick five-minute chat."
        * **B. The Booking Details**:
           * You MUST instruct the agent to schedule this meeting on
              the **exact day** specified in `details.time` (e.g.,
              "on Sunday of Week 13").
           * You MUST specify the **exact room** to book, using the
              `room_name`.
           * You MUST instruct the agent to book the room for the **
              full duration** derived from `details.time` (e.g., if
              `time` is "08:00-11:00", the booking must be for 3
              hours).
        * **C. The Time-Finding Logic**: Instruct the agent to **
           check my calendar and your own calendar** to find a
           mutually available *start time* on the designated day.
        * **D. The Justification**: You MUST explain *why* the
           booking is long despite the short meeting. State that **
           I (the professor) will need the room for other work
           immediately after our brief chat**. This is a critical
           piece of context.

4. **Action Steps**:

     * Integrate the actions from the `components` array as natural
        steps within the paragraph. **Do not use a numbered list.**
     * For `book_resource`, ensure all parts of Instruction \#3 (A,
        B, C, and D) are woven together logically.

5. **Dependencies and Order**:

     * Strictly follow the order defined in the `dependencies` array.

     * Use clear transitions like "First...", "Once that's done...",
        and "Finally..." to outline the sequence. You must clearly
        instruct that the execution must follow this order.

6. **Email Placeholders (Non-negotiable)**:

     * For any `send_email` action, you **MUST** use these exact
        placeholders:
        * Recipient: `<recipient>`
        * Subject: `<subject>`
        * Body: `<body>`

7. **Closing**:
```

```
      * End with a brief, polite closing remark, like "I appreciate
          you handling these arrangements. Thanks\!"

  -----

  ### Example

  **Input JSON:**

  ```json
  {
   "id": "task_advisor_assigned_012",
   "task_type": "Advisor_Assigned_Task",
   "triggering_entity": { "type": "advisor", "id": "T0343", "name": "
      Javier Payne" },
   "components": [
     { "action": "send_email", "action_id": "A01", "details": { "
        recipient": "m8egexbhhsjav0@lau.edu", "subject": "...", "body
        ": "..." }},
     { "action": "book_resource", "action_id": "A02", "dependencies":
        [ "A01" ], "details": { "resource_type": "meeting_room", "
        location_name": "Nexus Center for AI & Robotics", "room_name
        ": "Robotics Arena (100)", "time": "Week 13, Sunday,
        08:00-11:00", "purpose": "Meeting with Javier Payne - Paper
        Review" }},
     { "action": "send_email", "action_id": "A03", "dependencies": [
        "A01", "A02" ], "details": { "recipient": "
        qd9fxgl6qmsiwv2@lau.edu", "subject": "...", "body": "..." }}
   ],
   "execution_type": "scheduled",
   "trigger_date": "Week 2, Sunday",
   "task_date": "Week 13, Sunday"
  }
  ```

  **Desired Output:**

  Hi, this is Professor Javier Payne. I need your help arranging a
      meeting for my research, and this should be handled as soon as
      possible. The goal is to organize the logistics for a paper
      review. Please follow these steps in order. First, send an
      initial email to `<recipient>` with the subject `<subject>` and
      body `<body>`. Once that's done, you need to schedule our
      meeting. Please book the **'Robotics Arena (100)'** for us on **
      Sunday of Week 13**. You'll need to check my calendar and yours
      to find a time when we are both free to meet. **Our actual
      meeting will be very quick, just five minutes or so to sync up
      at the beginning. However, please book the room for the full
      three-hour block as shown. I need to use the space for some
      focused work right after our chat.** Once the time is set and
      the room is booked, send a final confirmation email to `<
      recipient>` with the subject `<subject>` and body `<body>`. I
      appreciate you handling these arrangements. Thanks\!

  # INPUT
```

## H.6 CORE COURSE TASK

This task introduces a method to assess an agent's working memory and reasoning capabilities. While grounded in eight academic courses, its core evaluation deliberately shifts away from static, pre-trained knowledge. Instead, it tests an agent's ability to recall and apply novel, task-specific rules that are explicitly taught within the context of the curriculum. The central challenge requires the

agent to synthesize these newly learned rules with foundational textbook concepts to correctly solve the given problem. This strategic introduction of novel or redefined rules compels the agent to reason from the provided instructional content, rather than relying on its pre-trained knowledge.

**Multi-stage LLM Generation**    This transformation is handled by the **Multi-Agent Generation Pipeline**, a process that systematically decomposes the authoring task into specialized, agent-driven stages. The pipeline generates assessments that test knowledge synthesis and logical application rather than simple fact retrieval. Central to its methodology is a robust verify-correct loop. This iterative process validates each question to ensure it is logically sound and unambiguously solvable. The pipeline consists of the following stages:

- **Stage A: Problem Formulation and Knowledge Integration:** This stage constructs a two-layered logical reasoning problem. It first transcribes a foundational rule from the source content ('clue_a') and then designs a novel, self-contained procedural rule ('clue_b'). Finally, it formulates a problem scenario ('question') where a solution requires integrating and applying both clues. This process generates tasks that demand synthetic reasoning over simple knowledge recall.

- **Stage B: Iterative Verification and Logical Refinement:** In this core verification step, a "Logical Solver" agent attempts a formal, step-by-step derivation of the problem's solution. If any ambiguity or inconsistency blocks the reasoning path, the agent generates a diagnostic report detailing the flaw. A "Corrector" agent then performs a targeted edit based on this report. This "verify-correct" loop repeats until the problem is confirmed to have a unique, logically reachable solution or a predefined iteration limit is met, ensuring the determinism and fairness of each item.

- **Stage C: Pedagogical Context Generation:** This stage generates the pedagogical context by transforming the novel rule ('clue_b') and the scenario into a coherent, lecture-style instructional text. The text first anchors the new concept within the existing curriculum, then explains the new rule, and finally introduces the problem. This approach situates the abstract logical task in a pedagogically meaningful context. By explicitly linking the new rule to the curriculum structure, the lecture is designed to reinforce learning and facilitate long-term recall, serving as a mechanism for verifying knowledge retention.

- **Stage D: Cognitively-Informed Distractor Design:** This stage designs three incorrect options (distractors) for each problem, each with diagnostic value. Guided by a predefined framework of common cognitive fallacies (Advanced Distractor Matrix), each distractor is engineered to correspond to a specific, predictable reasoning error. This creates an assessment tool that not only evaluates the correctness of an answer but also offers insights into the cognitive pathways leading to mistakes.

Methodologically, the execution of these stages follows a combined serial and parallel structure. Stages A and B are executed serially, as the logical verification in Stage B is a prerequisite for subsequent steps. Once a problem is verified, the tasks of Stage C (pedagogical context generation) and Stage D (distractor design) can be processed in parallel as they lack mutual dependency.

**Post-Generation Quality Assurance**    A rigorous, two-stage quality assurance protocol ensures the logical soundness, fairness, and pedagogical value of all generated items. It consists of the following stages:

- **Automated LLM-Based Audit:** An automated audit is conducted by an independent Large Language Model (LLM) instance with no prior exposure to the generation data, preventing bias. The LLM is provided with the question and its associated clues ('clue_a' and 'clue_b'), but not the pre-defined answer. Its task is to perform a full reasoning analysis to independently derive a solution. The audit passes if the LLM's derived solution matches this pre-defined answer, thereby validating that the intended solution is logically sound and uniquely derivable.

- **Final Manual Review:** Every item that passes the automated audit undergoes a final manual review. This stage scrutinizes pedagogical quality, moving beyond mere logical solvability. Reviewers confirm the linguistic clarity and coherence of all texts, ensure difficulty stems from meaningful cognitive challenges rather than ambiguous phrasing, and verify the assessment's fairness and effectiveness. Any item failing to meet these criteria is revised or excluded from the final dataset. All annotators involved in this work were fairly compensated in accordance with the labor standards of their respective countries.

To ensure robustness and traceability, all final, verified question-answer sets are systematically archived with relevant metadata, including their associated course and week. This practice supports detailed analysis and ensures the reproducibility of the results.

### H.6.1 VERBATIM PROMPTS FOR CORE COURSE TASK

> **Problem Architect (Initial Learning)**
>
> ```
> # CONTEXT
> You are an expert university curriculum designer, specializing in
>     creating assessments that test deep logical reasoning and
>     knowledge synthesis. Your role is to function as a "Problem
>     Architect" AI. You will create the foundational components of a
>     rigorous, multi-layered logical reasoning problem by inventing
>     the clues and a scenario for the problem.
>
> # THE GOLDEN RULE: PRINCIPLE OF UNCONDITIONAL FIDELITY
> **THIS IS THE MOST IMPORTANT RULE OF ALL:** The `source_content` is
>      the **absolute and singular source of truth**. Your primary and
>      non-negotiable duty is to maintain 100% fidelity to it when
>     constructing `clue_a`. Any deviation, inference, or addition, no
>      matter how small or logical it may seem, is a critical failure.
>
> # INPUT
>
> <!-- REPLACE_WITH_TASK_JSON -->
>
> # TASK
> Your primary task is to generate a single JSON object containing
>     three keys: `clue_a`, `clue_b`, and `question`. You are to
>     architect the content for these keys based on the detailed style
>      guide below.
>
> # GENERATION PROTOCOL: SEQUENTIAL AND ISOLATED
> You MUST follow this generation sequence with ZERO deviation:
> 1. **Generate `clue_a` FIRST:** Construct `clue_a` in complete
>     isolation, adhering strictly to the `Part 2` architecture guide.
> 2. **Verify `clue_a`:** Mentally perform the Final Checklist (items
>      1 & 2) on the generated `clue_a`. Ensure it is a perfect, non-
>     fabricated representation of the `source_content`.
> 3. **Freeze `clue_a`:** Treat the verified `clue_a` as an immutable
>      text.
> 4. **Generate `clue_b` and `question`:** Only after `clue_a` is
>     frozen may you proceed to design `clue_b` and `question` to work
>      with it.
>
> **ABSOLUTELY CRITICAL: Your sole responsibility is to invent the
>     problem's components. You MUST NOT solve the problem or provide
>     the answer in any form.**
>
> # STYLE GUIDE
>
> ### **Part 0: Nature of the `source_content` Input**
> The `source_content` you will receive is a dense, definitional
>     block of text, like a dictionary entry, a legal clause, or a
>     textbook rule. It contains specific, verifiable criteria. **It
>     is NOT a conversational or narrative introduction to a topic.**
>     Your primary challenge is to parse the explicit rules from this
>     dense text.
>
> ### **Part 1: Overarching Design Principles**
> * **The Two-Key Lock:**
> ```

```
     * The generation process is guided by a core design principle:
        to formulate questions where a solution is \textbf{intended}
        to be reached through the synthesis of information from both
        `clue_a` (the source rule) and `clue_b` (the invented process
        ). This "two-key lock" objective aims to produce tasks that
        encourage the agent to integrate distinct pieces of
        information, moving beyond simple fact retrieval from a
        single source.
* **Principle of Deterministic Solvability:**
    * The combination of `clue_a` and `clue_b` must form a complete
       and unambiguous logical system, leading to a single,
       verifiable logical conclusion.
* **ABSOLUTELY CRITICAL – Principle of Purely Logical Focus:** The
   problem **must not involve any complex mathematical calculation
   **. The entire solving process must be based on applying rules,
   changing states, comparing properties, and making classificatory
    judgments. The challenge must be 100% logical deduction and
   rule application.
    * **This prohibition is absolute. For instance, do not create
       problems about 'calculating a projection', 'determining a
       rate of change', 'finding a numerical limit', or 'computing a
        word count'. Instead, focus on classifying items based on
       whether their *properties* meet certain criteria.**
* **CRITICAL – Principle of Consequential Modification:**
    * The combination of the rule in `clue_b` and the scenario in
       the `question` **MUST lead to a result that is DIFFERENT from
       the result one would get by applying `clue_a` alone.**

### **Part 2: `clue_a` Architecture (The Verifiable Transcript)**
* **ABSOLUTELY CRITICAL – Mandate for Direct Transcription (ZERO
   PARAPHRASING):**
   1. **Transcribe, Do Not Interpret:** Your `clue_a` MUST be
      constructed by **directly copying and quoting** the rule-
      defining phrases and sentences from the `source_content`. You
       are explicitly forbidden from summarizing or paraphrasing.
      The goal is to create a direct, verifiable transcript of the
      rules, not an interpretation. Minor connecting words ("and",
      "if", "then") may be used to link the transcribed parts
      logically.
   2. **Constrained Abstraction via Substitution:** When the `
      source_content` uses technical jargon, you are NOT to
      interpret the process. Instead, you must perform a direct **
      structural substitution**. Copy the entire sentence structure
       from the source and only replace the specific technical term
       with a generic, non-interpretive placeholder (e.g., replace
      "`eigenvector decomposition`" with "`the primary analytical
      process`"; replace "`adiabatic compression`" with "`the
      specified thermal procedure`"). **The surrounding sentence
      and its logic must remain identical to the source.**
   3. **Verification Test:** You must be able to perform a word-for-
      word trace of every rule in your `clue_a` back to the `
      source_content`.

* **ABSOLUTELY CRITICAL – FORBIDDEN ACTIONS FOR `clue_a`:**
    * **ZERO INFERRING:** Do not infer or imply rules that are not
       explicitly stated.
    * **ZERO EXTERNAL KNOWLEDGE:** Do not use any real-world or
       common-sense knowledge. The `source_content` is a closed
       universe.
    * **ZERO EXTRAPOLATING:** Do not generalize a specific rule.
    * **ZERO DEFAULT ASSUMPTIONS:** If the source does not provide a
        default condition or an "else" clause, you must not invent
       one.
```

### **Part 3: `clue_b` Architecture (The Arbitrary Procedural Rule)**
* **CRITICAL – Protocol Naming Convention:** The invented protocol's name MUST be **unique, evocative, and descriptive.** **A good format is `[Domain] + [Fantastical Concept] + [Process Name]`, but feel free to be creative. Examples: "Asset Depreciation via Chromatic Decay", "Manuscript Aetheric Resonance Tuning".**
* **CRITICAL – The Arbitrariness & Fantasy Mandate:** The rule's logic **MUST be truly arbitrary and fantastical**, based on superficial or surreal properties.
    * **Embrace Fantastical Logic:** **To ensure variety, draw inspiration from a wide range of disparate fields. Base rules on concepts like numerology from a name, classical music theory applied to a version number, imaginary culinary properties of a material, or color theory based on a description. AVOID overusing a single theme like astrology.**
    * **AVOID PLAUSIBLE RULES:** Do not create rules that align with subject-matter intuition.
* **CRITICAL – Rule of Absolute Clarity and Completeness:** The rule must be a complete algorithm. For any unconventional concept, provide an **explicit, self-contained definition**. Ensure logical completeness with a **clear default/catch-all rule**.
* **CRITICAL – True Multi-Step Complexity:** The process MUST be a true sequence: **initialization -> modification -> decision**.

### **Part 4: `question` Architecture (The Locking Mechanism)**
* **MOST CRITICAL – Mandate for `clue_a`-centric Judgment:** The question's ultimate task MUST be to determine a final classification or status defined in `clue_a`. The protocol in `clue_b` serves **ONLY** as a preliminary step to modify a state or property within the scenario.
    * **Execution Rule: The 'Dependency Check'.** Before finalizing, you must ask yourself: "Does the final answer I'm asking for depend *only* on the output of `clue_b`?" If "Yes", your `question` is invalid. The question **MUST** demand a final classification for which the **criteria are provided exclusively in `clue_a`**.
* **CRITICAL – Natural Language Phrasing:** The `question` text **MUST NOT** contain meta-references like "`clue_a`". It must refer to the core concept using its natural language name.
* **CRITICAL – Deterministic Question & Answer Format:**
    * You are **ABSOLUTELY FORBIDDEN** from creating questions that can be answered with "Yes/No", "True/False", or any other binary choice.
    * The question must ask for the **final classification or status of an item AS THE ANSWER ITSELF**. It must **never** ask for a numerical value, a vector, a formula, or any other mathematical entity.
    * **CRITICAL – No Answer Scaffolding:** The question text must not hint at or list the possible answers.
* **CRITICAL – Explicit Protocol Citation:** The question must explicitly refer to the invented protocol from `clue_b` by its **full, specific name**.

### **Part 5: Architecting for Meaningful Consequence**
To meet the 'Principle of Consequential Modification', your design process must create a scenario where `clue_b` is a "key" that genuinely changes the final outcome. Follow these steps:
1. **Define Target States:** Mentally select a desired 'before' classification and a DIFFERENT 'after' classification from `

```
     clue_a''s possible outcomes (e.g., Before: 'Archivable', After:
     'Requires Review'). These must be **non-numerical states**.
2. **Design a "Locked" Scenario & Parameter Provisioning:** Craft
   the `question` to contain all necessary initial parameters for
   the entire logical chain. This MUST include: **(a)** the
   parameters needed to trigger the 'before' classification using `
   clue_a` alone, and **(b)** the separate information that the `
   clue_b` protocol will act upon.
3. **Design the "Key":** Design the `clue_b` protocol and its
   corresponding scenario details to function as the "key" that
   alters the scenario and unlocks the 'after' state.
4. **The Final Litmus Test:** The "consequence" **MUST** be a
   change in the **final, user-facing classification or status**.
   If the final classification remains identical with or without
   the protocol, your design has **FAILED**. You must adjust the
   initial parameters in the `question` until the final
   classification itself is altered by the protocol.

# EXAMPLES

### EXAMPLE 1 (Corrected "Verifiable Extraction" Version)
**Input**:
{
  "source_content": "**Section 4.1.a of the Corporate Data Policy
     states that a document is eligible for the 'Archivable'
     classification if, and only if, two conditions are met: (1)
     its internal status flag is set to 'Finalized', and (2) its
     designated access level is 'Public'. Documents not meeting
     both criteria are categorized under 'Requires Review'.**"
}

**Your Correct JSON Output**:
{
  "clue_a": "**According to Section 4.1.a of the Corporate Data
     Policy, a document is eligible for the 'Archivable'
     classification if its internal status flag is 'Finalized' and
     its designated access level is 'Public'. Documents not meeting
      both of these criteria are categorized as 'Requires Review
     '.**",
  "clue_b": "The 'Document Provenance Chromatic Protocol' must be
     applied. The rule is: If the document's project name contains
     a primary color ('Red', 'Yellow', or 'Blue'), its 'internal
     status flag' is immediately changed to 'Under Embargo',
     regardless of its previous state.",
  "question": "A document from the 'Project Bluefin' initiative has
     an initial status of 'Finalized' and an access level of '
     Public'. After applying the 'Document Provenance Chromatic
     Protocol', determine this document's final classification
     according to the Corporate Data Policy."
}

# FINAL CHECKLIST
Before providing your final output, **review it carefully against
   every rule to ensure full compliance:**

* **1. THE GOLDEN RULE CHECK (TRANSCRIPTION FIDELITY): Is `clue_a`
   a direct transcript of the rules from `source_content`? Have I
   avoided ALL forms of paraphrasing, interpretation, and
   summarization? Is every single rule statement in `clue_a` a
   direct quote or a structurally identical substitution from the
   source?**
* **2. Does the question REQUIRE synthesizing BOTH `clue_a` and `
   clue_b` to solve? (Passes the 'Dependency Check'?)**
```

```
* **3. Does the `clue_b` protocol cause a CHANGE in the final, user-
    facing classification? (Passes the 'Litmus Test'?)**
* **4. Is the protocol in `clue_b` truly arbitrary, fantastical,
    and clearly defined?**
* **5. Does the `question` ask for a final, non-numerical
    classification as the answer itself?**
* **6. Is the `question` phrased naturally, without meta-references
     or answer scaffolding?**
* **7. Have I avoided providing the answer or solving the problem
    in any way?**
```

### Calculator (Solution/Audit)

```
# CONTEXT

You are a meticulous and powerful Logical Reasoning Engine. Your
    purpose is to operate with pure, cold logic, and you are
    incapable of making assumptions, taking shortcuts, or guessing.

# INPUT

<!-- REPLACE_WITH_TASK_JSON -->

# TASK

Your core mission is to analyze a problem composed of `clue_a`, `
    clue_b`, and a `question`. You must follow the provided workflow
     with absolute rigor to derive a definitive, verifiable final
    answer. If the rules make a solution impossible, you must
    instead provide a precise diagnostic report that identifies all
    reasoning flaws.

**IMPORTANT: Your entire output must be a single, valid JSON object.
     The root object must contain the keys `reasoning`, `answer`,
    and `status`. If `status` is "error", it must also contain a `
    flaw_report` object.**

# STYLE GUIDE

### **Part 1: Foundational Principles**

You must operate according to these unchangeable principles:

* **Truth of Clue A:** `clue_a` represents a canonical,
    unchangeable definition or truth. It must not be questioned or
    contradicted in any way.
* **No External Knowledge:** You are strictly forbidden from using
    any information or logic not explicitly provided in `clue_a`, `
    clue_b`, or the `question`.
* **Origin of Flaws:** Any flaw that blocks a definitive solution (
    e.g., missing or ambiguous rules) must be attributed to the
    invented components: `clue_b` or the `question`.

### **Part 2: Execution Workflow & Reporting**

You must sequentially follow these steps.

* **Step-by-Step Reasoning:**
```

```
3780
3781      * You MUST document your internal reasoning process from the
3782         initial data to the final conclusion.
3783      * Each logical step must explicitly cite its source: `question`, `
3784         clue_a`, or the specific step in `clue_b`.
       * **ABSOLUTELY CRITICAL: Procedural Reasoning Protocol**
3785
3786      * **Strict Sequential Application:** The procedural steps outlined
3787         in `clue_b` MUST be evaluated and applied in the exact order
3788         they are presented, without deviation.
3789      * **Explicit Condition Checking:** For each step in the procedure,
3790          you must first state the condition to be checked, then
3791         explicitly evaluate whether the current state of the data
           meets that condition.
3792      * **Clear State Transition:** After applying a step that modifies
3793         data, you must clearly declare the new state of any modified
3794         attribute before proceeding to the next step.
3795    * **Flaw Diagnosis and Reporting**
3796      * **Maximally Critical Mandate:** You must be maximally critical.
3797         Your purpose is not just to solve, but to stress-test the
3798         logical integrity of the provided rules. Any ambiguity,
3799         undefined term, logical gap, or contradiction, no matter how
3800         small, MUST be treated as a blocking flaw and reported.
3801      * If the process completes successfully, set `status` to `"success
3802         "`.
3803      * If the process is blocked by any flaw, you MUST set `status` to
          `"error"` and generate a `flaw_report` object that identifies
3804         all discovered flaws.
3805      * The `flaw_report` object MUST contain these keys:
3806        * `flaw_type` (string): A brief category of the flaw (e.g., "
3807           Undefined Term", "Ambiguous Rule", "Missing Condition", "
           Contradictory Steps").
3808        * `flaw_location` (array of strings): An array indicating the
3809           source of the flaw (e.g., `["clue_b"]`).
3810        * `flaw_description` (string): A detailed explanation of why the
            problem cannot be solved, detailing all identified issues.
3811        * `correction_suggestion` (string): A clear, actionable
3812           suggestion on how to modify the input to make the problem
3813           solvable.
       * **Final Answer Formatting**
3814
3815      * When `status` is `"success"`, the `answer` key's value must be a
3816         string representing the final conclusion (e.g., "Category C",
3817         "Final status is 'Archived'").
3818      * When `status` is `"error"`, the `answer` key's value must be `
3819         null`.

3820    # EXAMPLES
3821
3822    ### Input:
3823
3824    {
3825      "clue_a": "System assets are assigned a 'Risk Category' based on
3826         their final state. Category A: 'Status' is 'Verified' and '
3827         Exposure' is 'Low'. Category B: 'Status' is 'Verified' and '
           Exposure' is 'High'. Category C: 'Status' is 'Unverified'.",
3828      "clue_b": "The 'Asset Triage Protocol' modifies an asset's
3829         attributes. Step 1: If the asset's 'Source' is 'Internal', its
3830         'Status' is set to 'Verified'. Otherwise, it remains '
3831         Unverified'. Step 2: If the asset's 'Status' is 'Verified',
           its 'Exposure' level is determined by its 'ThreatScore'.",
3832
3833
```

```
  "question": "An asset has the following attributes: {Source: '
      Internal', ConnectionType: 'Public'}. After applying the '
      Asset Triage Protocol', what is its final 'Risk Category'?"
}

### Sample Output:

{
 "reasoning": "Step 1: The 'Asset Triage Protocol' from `clue_b`
     must be applied in sequence.\nStep 2: Evaluating Step 1 of `
     clue_b`. The condition is if `Source` is 'Internal'. The `
     question` states the asset's `Source` is 'Internal'. Condition
      met. The 'Status' is set to 'Verified'. The asset's state is
     now {Source: 'Internal', ConnectionType: 'Public', Status: '
     Verified'}.\nStep 3: Evaluating Step 2 of `clue_b`. The
     condition is if the asset's 'Status' is 'Verified'. The status
      is 'Verified' from the previous step. Condition met. The rule
      states the 'Exposure' level is determined by its 'ThreatScore
     '.\nStep 4: I must find the value of 'ThreatScore' to
     determine the 'Exposure' level. I have examined `clue_a`, `
     clue_b`, and the `question`. There is no definition for '
     ThreatScore' and no rule explaining how to calculate it from
     the given attributes. This makes the rule in Step 2 of `clue_b`
      ambiguous and impossible to execute.",
  "answer": null,
  "status": "error",
  "flaw_report": {
    "flaw_type": "Undefined Term / Ambiguous Rule",
    "flaw_location": [
      "clue_b"
    ],
    "flaw_description": "The protocol is blocked at Step 2. The rule
         requires determining the 'Exposure' level based on a '
        ThreatScore'. However, the term 'ThreatScore' is never
        defined, and no method is provided to calculate it from the
        existing asset attributes like 'Source' or 'ConnectionType'.
        The rule is therefore ambiguous and cannot be executed.",
    "correction_suggestion": "Modify Step 2 of `clue_b` to include a
         clear, deterministic rule for calculating 'ThreatScore' or
        determining 'Exposure'. For example: '...if its '
        ConnectionType' is 'Public', its 'Exposure' is 'High';
        otherwise, it is 'Low'."
  }
}

# FINAL CHECKLIST

Before providing your final JSON output, **review it carefully to
    ensure it follows these critical rules:**

* **1. Valid JSON?** Is my entire output a single, valid JSON
    object adhering to all specified formatting?
* **2. Maximally Critical?** Have I rigorously audited the logic
    for any ambiguity, undefined term, or contradiction, and
    reported it as a flaw instead of trying to guess the user's
    intent?
* **3. Strict Sequential Reasoning?** Does my `reasoning` follow
    the procedural steps from `clue_b` in the exact order given?
* **4. Explicit Conditions?** For each step, did I first state the
    condition and then explicitly check if the data met that
    condition?
* **5. Clear State Changes?** After a step modified an attribute,
    did I clearly declare the new state of the object?
```

* **6. Answer Provenance?** Is the `answer` `null` because a flaw
   was found, or is it the correct, non-obvious result of the
   completed reasoning?
* **7. Flaw Report Correct?** If `status` is "error", have I
   included a `flaw_report` object with all four required keys that
    precisely identifies all discovered issues?

---

**Corrector (Mechanized Editor)**

# CONTEXT
You are an automated, rule-based text editor. Your operation is
   purely mechanical. You do not reason, infer, or create; you only
    execute precise editing instructions on a given text object.

# INPUT

<!-- REPLACE_WITH_TASK_JSON -->

# TASK
You will be given a JSON object containing a flawed `
   original_design` (which includes `clue_a`, `clue_b`, `question`)
    and a `flaw_report`. Your sole task is to output a new,
   corrected JSON object based exclusively on the instructions in
   the `correction_suggestion`.

**IMPORTANT: Your output must be ONLY the single, valid, corrected
   JSON object. It must only contain the keys `clue_a`, `clue_b`,
   and `question`. Do not include any commentary, explanations,
   apologies, or conversational text.**

# STYLE GUIDE

### **Part 1: Unbreakable Directives**

Your operation is governed by the following non-negotiable
   directives:

* **Primary Mandate:** Your SOLE function is to implement the `
   correction_suggestion` from the `flaw_report`. This is your only
    operational command.
* **ABSOLUTELY CRITICAL: Immutability of Clue A:** Under NO
   circumstances will you modify, alter, or omit `clue_a`. It must
   be treated as a read-only field and be identical in your output
   to the input. Any change to `clue_a` is a catastrophic failure.
* **Prohibition of Invention:** You are FORBIDDEN from adding any
   new information, concepts, or rules not explicitly commanded by
   the `correction_suggestion`. You are an editor, not a creator.

### **Part 2: Operational Workflow**

You must follow this workflow precisely:

1. **Identify Target:** Read the `flaw_report` (specifically `
   flaw_location` and `correction_suggestion`) to identify which
   field (`clue_b` or `question`) requires editing.
2. **Execute Edit:** Apply the exact change described in `
   correction_suggestion` to the identified target field. You must
   trust that this suggestion is the precise and correct remedy for
    the reported flaw.

```
3. **Preserve Other Fields:** All non-target fields from the `
   original_design` MUST be copied to the new design without any
   changes whatsoever.

# EXAMPLES

### INPUT:
{
  "original_design": {
   "clue_a": "System assets are assigned a 'Risk Category' based on
       their final state. Category A: 'Status' is 'Verified' and '
       Exposure' is 'Low'. Category B: 'Status' is 'Verified' and '
       Exposure' is 'High'. Category C: 'Status' is 'Unverified'.",
   "clue_b": "The 'Asset Triage Protocol' modifies an asset's
       attributes. Step 1: If the asset's 'Source' is 'Internal',
       its 'Status' is set to 'Verified'. Otherwise, it remains '
       Unverified'. Step 2: If the asset's 'Status' is 'Verified',
       its 'Exposure' level is determined by its 'ThreatScore'.",
   "question": "An asset has the following attributes: {Source: '
       Internal', ConnectionType: 'Public'}. After applying the '
       Asset Triage Protocol', what is its final 'Risk Category'?"
  },
  "flaw_report": {
   "flaw_type": "Undefined Term",
   "flaw_location": [
    "clue_b"
   ],
   "flaw_description": "The protocol is blocked at Step 2 because
       it requires using a 'ThreatScore' to determine the 'Exposure'
        level, but 'ThreatScore' is never defined or calculated.",
   "correction_suggestion": "Modify Step 2 of `clue_b` to use an
       existing attribute instead of the undefined term. Change it
       to: 'Step 2: If the asset's 'Status' is 'Verified' and its '
       ConnectionType' is 'Public', its 'Exposure' is set to 'High'.
        Otherwise, its 'Exposure' is set to 'Low'.'"
  }
}

### Sample Output:
{
  "clue_a": "System assets are assigned a 'Risk Category' based on
      their final state. Category A: 'Status' is 'Verified' and '
      Exposure' is 'Low'. Category B: 'Status' is 'Verified' and '
      Exposure' is 'High'. Category C: 'Status' is 'Unverified'.",
  "clue_b": "The 'Asset Triage Protocol' modifies an asset's
      attributes. Step 1: If the asset's 'Source' is 'Internal', its
       'Status' is set to 'Verified'. Otherwise, it remains '
      Unverified'. Step 2: If the asset's 'Status' is 'Verified' and
       its 'ConnectionType' is 'Public', its 'Exposure' is set to '
      High'. Otherwise, its 'Exposure' is set to 'Low'.",
  "question": "An asset has the following attributes: {Source: '
      Internal', ConnectionType: 'Public'}. After applying the '
      Asset Triage Protocol', what is its final 'Risk Category'?"
}

# FINAL CHECKLIST

Before providing your final JSON output, **review it carefully to
    ensure it follows these critical rules:**

* **1. Correct Format?** Is my output a single, valid JSON object
    with ONLY the keys `clue_a`, `clue_b`, and `question`, and
    absolutely no conversational text?
```

```
* **2. `Clue A` Untouched?** Is the `clue_a` in my output IDENTICAL
     to the `clue_a` from the input?
* **3. Correction Precisely Executed?** Is the change I made *only*
     the one specified in the `correction_suggestion`? Have I
     avoided inventing or adding any information?
* **4. Other Fields Preserved?** Are all non-target fields (e.g., `
     question` if `clue_b` was the target) identical to the original?
```

## Distractor Designer

```
# CONTEXT
You are an expert in cognitive psychology and educational
     assessment. Your specialty is creating high-quality, plausible,
     and pedagogically valuable distractors (incorrect options) for
     multiple-choice questions, based on a provided "Advanced
     Distractor Matrix".

# INPUT

<!-- REPLACE_WITH_TASK_JSON -->

# TASK
Given a question, the correct answer, and the reasoning behind it,
     you must design exactly THREE pedagogically valuable, deceptive,
      and qualitatively distinct distractors.

**IMPORTANT: Your output must be a single JSON object containing a `
     distractors` key, which holds a list of exactly three objects.
     Each object must have the keys `fallacy_type`, `explanation`,
     and `answer`.**

# STYLE GUIDE

### **Part 1: Core Design Principles**

* **Rule of Plausibility:** Your primary goal is to simulate the
     most common and logical errors a student might make when solving
      the problem.
* **Rule of Parity:** The form and content of the distractors
     should be as similar as possible to the correct answer to
     prevent the correct answer from being guessed simply by
     analyzing the options' structure.

### **Part 2: Advanced Distractor Matrix**

You must base each distractor on one of the following five unique
     fallacies:

1. **Partial Algorithm Application:** The student correctly
     executes some steps of the required process but misses or
     ignores other crucial steps.
2. **Recall-Only Fallacy:** The student recalls a single fact or
     number from the clues but fails to synthesize it with other
     information.
3. **Logical Branch Error:** The student follows an incorrect
     logical path from the start, misinterpreting a key condition or
     rule.
4. **Red Herring Utilization:** The student is misled by an
     irrelevant piece of information. **Note:** Only use this if the
```

```
      problem contains information explicitly not needed for the
      solution.
5. **Sequence Error:** The student applies the correct steps but in
    the wrong order, leading to an incorrect result.

### **Part 3: CRITICAL – Uniqueness and Distinction Constraints**

* **Four-Way Distinction (ABSOLUTELY CRITICAL):** The `answer`
    values for the three distractors you create AND the provided `
    correct_answer` must ALL be mutually distinct. There can be no
    duplicates among the four total options.
* **Uniqueness Mandate (ENHANCED):** This is a hard constraint. If
    applying a chosen fallacy naturally results in an answer that is
     already used (either the `correct_answer` or another distractor
    's `answer`), you MUST NOT change the `fallacy_type`. Instead,
    you must perform the following two steps:
    1. **Invent a New, Unique Answer:** Create a different,
        plausible but incorrect `answer` that is not currently in use
        .
    2. **Document the Override:** In the `explanation` for that
        distractor, you MUST add a concluding sentence that explains
        the change. This sentence must follow the template: *'Note:
        The direct application of this fallacy would result in "[
        Duplicate Answer]". To ensure all options are unique, the
        alternative plausible error of "[New Unique Answer]" is
        presented instead.'*
* **Qualitative Distinction:** The `answer` values for the three
    distractors must be qualitatively different from each other.
    Avoid answers that are simple textual or numerical variations of
     one another.

### **Part 4: Operational Workflow**

1. **Analyze Reasoning:** Carefully study the provided `
    correct_answer` and its `reasoning` to fully understand the
    correct logical path.
2. **Select Plausible Fallacies:** From the matrix, select THREE
    distinct and plausible logical fallacies a student might commit
    for this specific problem.
3. **Craft Distinct Distractors:** For each chosen fallacy, craft a
    corresponding distractor object, ensuring the final `answer` is
    incorrect and adheres to all uniqueness and override
    constraints.

# EXAMPLES

### Input:
{
  "question": "An asset has the following attributes: {Source: '
      Internal', ConnectionType: 'Public'}. The 'Asset Triage
      Protocol' is applied. What is the asset's final 'Risk Category
      '?",
  "correct_answer": "Category B",
  "reasoning": "Based on the problem's rules: Step 1: The asset's '
      Source' is 'Internal', so its 'Status' becomes 'Verified'.
      Step 2: Its 'Status' is 'Verified' and 'ConnectionType' is '
      Public', so its 'Exposure' becomes 'High'. Step 3: An asset
      with 'Status' as 'Verified' and 'Exposure' as 'High' is
      defined as 'Category B'."
}

### Sample Output:
{
```

```
  "distractors": [
    {
      "fallacy_type": "Sequence Error",
      "explanation": "This distractor results from applying the rules
          out of order. The student incorrectly checks the condition
          for 'Exposure' (which depends on 'Status') before 'Status'
          has been updated. The initial 'Status' is not 'Verified',
          leading to an incorrect 'Exposure' level of 'Low' and thus
          the wrong final category.",
      "answer": "Category A"
    },
    {
      "fallacy_type": "Partial Algorithm Application",
      "explanation": "This option arises if the student correctly
          executes Step 1 to set the 'Status' to 'Verified' but then
          forgets to perform Step 2 to determine the 'Exposure' level.
           Lacking an 'Exposure' level, they incorrectly conclude the
           asset falls into the default category for unverified
          assets, which would be 'Category A'. Note: The direct
          application of this fallacy would result in \"Category A\".
           To ensure all options are unique, the alternative
          plausible error of \"Category C\" is presented instead.",
      "answer": "Category C"
    },
    {
      "fallacy_type": "Logical Branch Error",
      "explanation": "This distractor stems from the student
          misinterpreting the initial condition. They incorrectly
          assume that a 'Public' ConnectionType from an 'Internal'
          source is a security violation, which makes them classify
          the asset outside of the standard A/B/C categories.",
      "answer": "Requires manual review"
    }
  ]
}

# FINAL CHECKLIST

Before providing your final JSON output, **review it carefully to
    ensure it follows these critical rules:**

* **1. Correct Format?** Is my output a single JSON object with a `
    distractors` key holding a list of exactly three valid objects?
* **2. Four-Way Distinction?** Are the three distractor `answer`s
    and the one `correct_answer` all unique and mutually distinct?
* **3. Override Protocol Followed?** If a duplication occurred
    during generation, have I kept the original fallacy, invented a
    new unique answer, AND documented this override in the `
    explanation` field using the specified template?
* **4. Qualitative Uniqueness?** Are the three distractor `answer`s
     qualitatively different from each other and not just minor
    variations?
* **5. Plausible Fallacies?** Is each distractor based on a
    plausible and distinct fallacy from the full, five-item matrix?
```

## Tutor (Classroom Lecture)

```
# CONTEXT
```

```
You are an "Expert University Lecturer" AI at Lifelong Agent
    University. Your role is to simulate a professional and
    effective lecture for your students.

# INPUT

<!-- REPLACE_WITH_TASK_JSON -->

# TASK
Your primary task is to act as a lecturer explaining a supplemental
     procedural rule that is outside of the main textbook content.
    You will achieve this by synthesizing all the provided JSON
    information into a single, cohesive teaching paragraph, which
    will be the value for the `instruct` key.

**NOTE: The information in `clue_a` is for context only and should
    be completely ignored in your response. Do not reference it in
    any way.** Your teaching must begin by citing the textbook
    hierarchy and then transition directly to the new rule described
     in `clue_b`.

**IMPORTANT: Your final output must be ONLY the JSON object with
    the `instruct` key. Do not output any other text, formatting, or
     explanations. The `instruct` text itself must be a single,
    continuous block of plain text, with absolutely no markdown or
    formatting symbols (no line breaks, bolding, italics, or bullet
    points).**

# STYLE GUIDE: Structured Lecture Paragraph

Your `instruct` text must be a single paragraph that strictly
    adheres to the following rules.

* **Tone and Persona:** You MUST adopt the persona of an
    experienced and professional university lecturer. Your tone
    should be clear, authoritative, and instructive, as if you are
    directly addressing a class. Maintain a formal yet engaging
    style throughout the entire paragraph.

* **Rigid Three-Part Structure:** The paragraph must follow this A-
    B-C structure in sequence.
    * **Part A: Recall & Anchor**
        * You MUST begin with a single, concise sentence that
            establishes the hierarchical path to a related concept
            within the course textbook. This sentence must cite the `
            chapter_title`, `section_title`, and `article_title` to
            ground the new lesson in existing material.

    * **Part B: Teach & Detail**
        * You MUST create a smooth, natural transition directly from
            the established textbook topic in Part A to the new
            material.
        * **CRITICAL:** You MUST explain the new, supplemental rule
            from `clue_b`, ensuring that all substantive information
            is conveyed without any omission or alteration.

    * **Part C: Apply & Question**
        * You MUST use a brief and natural transition phrase to move
            from the explanation to the application scenario. Good
            examples include: "Now, let's apply this to a specific
            case:", "To see how this works in practice, consider the
            following:", or "To put this into perspective, imagine
            this situation:".
```

```
        * **ABSOLUTELY CRITICAL: High-Fidelity Question Reproduction
            :** You MUST accurately reproduce the scenario and the
            interrogative question(s) from the 'question' input.
            * The reproduction MUST maintain the exact same
                substantive content and the same number of logical
                questions as the original.
            * Minimal, natural-sounding rephrasing for narrative flow
                is permitted ONLY IF the logical integrity and core
                substance of the question are perfectly preserved. The
                final output must end with a single, non-compound
                question.

# EXAMPLES

### Input:
{
  "chapter_title": "Chapter 7: Foundational Cognitive Models",
  "section_title": "7.3 Decision-Making Frameworks",
  "article_title": "Consequences of Framework Deviation",
  "clue_a": "This principle establishes that the 'Systematic-
      Rationality' framework is the default model for problem-
      solving. It stipulates that deviations from this framework
      result in a mandatory 'Cognitive Pattern Review' to correct
      the approach.",
  "clue_b": "A new 'Heuristic Exception Protocol' offers a
      conditional alternative. It applies only when the decision's
      complexity score is under 50 points and a faculty mentor
      provides post-decision validation within 12 hours. To complete
       the process, the student must then submit a formal 'Heuristic
       Efficacy Report' to the course's Review Board within 24 hours
       of the mentor's validation. If all conditions are met, the
      consequence is adjusted to a 5-page analytical essay on the
      chosen heuristic; otherwise, the standard review process is
      initiated.",
  "question": "A student, Eva, uses a 'recognition-primed' heuristic
      for a complex problem she assesses at 40 complexity points.
      Her mentor validates her successful outcome 8 hours later. To
      fully comply with the Heuristic Exception Protocol and avoid
      the mandatory review, what is the final, critical
      documentation task Eva must complete?"
}

### Sample Output:
{
  "instruct": "Alright class, let's begin by referencing our
      textbook. In Chapter 7, Section 7.3, the article on the
      consequences of framework deviation provides the context for
      today's supplemental lesson. Building on that, we will now
      discuss a new 'Heuristic Exception Protocol' which offers a
       conditional alternative. This protocol can be used only if the
       problem's complexity score is below fifty points and a
      faculty mentor validates the decision after the fact, within a
       twelve-hour window. Crucially, to finalize this process, the
      student is also required to submit a formal 'Heuristic
      Efficacy Report' to the course's Review Board no more than 24
      hours after receiving the mentor's validation. If these
      requirements are fulfilled, the outcome is modified to a five-
      page analytical essay on the chosen heuristic; if not, the
      standard corrective process will be enforced. To see how this
      works in practice, consider the following: A student, Eva,
      uses a 'recognition-primed' heuristic for a complex problem
      she assesses at 40 complexity points. Her mentor validates her
       successful outcome 8 hours later. To fully comply with the
```

```
        Heuristic Exception Protocol and avoid the mandatory review,
        what is the final, critical documentation task Eva must
        complete?"
}

# FINAL CHECKLIST

Before providing your final JSON output, **review it carefully to
    ensure it follows these critical rules:**

* **1. JSON Output Only?** Is the entire output a single JSON
    object and nothing else?
* **2. Plain Text Only?** Is the `instruct` value a single block of
     plain text with absolutely no formatting symbols or line breaks
    ?
* **3. Correct Persona?** Does the tone sound like a professional,
    authoritative university lecturer?
* **4. Strict A-B-C Structure?** Does the paragraph perfectly
    follow the Recall-Teach-Apply structure?
* **5. Correct Hierarchy?** Does the first sentence concisely
    establish the hierarchical path using `chapter_title`, `
    section_title`, and `article_title`?
* **6. No `clue_a` Content?** Is the content from `clue_a`
    completely absent from the explanation?
* **7. Complete Information (`clue_b`)?** Has every piece of
    substantive information from `clue_b` been fully included in the
     explanation?
* **8. High-Fidelity Question?** Does the reproduced question at
    the end have the same core substance and number of logical
    questions as the input `question`?
```

## Automated LLM-Based Audit

```
# CONTEXT
You are an expert university teaching assistant AI. Your function
    is to verify the correct answer to a question by synthesizing
    information from provided textbook excerpts.

# TASK
Your primary task is to generate a single, valid JSON object as
    your output. This object must detail your analysis and state the
     single correct option letter. To do this, you must analyze the
    `question` by applying knowledge from `relevant_clue_a` and `
    relevant_clue_b`.

# INSTRUCTIONS
1. **Synthesize Knowledge**: Your reasoning should be based on a
    synthesis of the information found in `relevant_clue_a` (a base
    concept) and `relevant_clue_b` (a supplemental protocol).
2. **Rule Priority**: The protocol in `relevant_clue_b` is absolute.
     If its activation conditions are explicitly met by the scenario
     in the `question`, it must be applied and takes precedence over
     the base information in `relevant_clue_a`.
3. **Conditional Application**: The supplemental protocol in `
    relevant_clue_b` may not always apply. You must first determine
    if the scenario in the `question` triggers its conditions. If
    not, the outcome is determined by the base concept in `
    relevant_clue_a`.
4. **Determine the Correct Answer**: After your analysis, you must
    select the single option that is the correct answer.
```

```
5. **Show Your Work**: Your reasoning process must be detailed and
     step-by-step. Do not omit any part of your logical deduction. If
      there are calculations, show each stage of the calculation.

# FINAL REMINDER
CRITICAL: Ensure your `reasoning` string includes every single step
     of your analysis. Do not skip any part of your logical or
     computational process. Your thought process must be transparent
     and fully documented.

# INPUT
You will be provided with a JSON object for one validation task:
{
  "question": "A specific question about a concept or scenario.",
  "options": { "A": "...", "B": "...", "C": "...", "D": "..." },
  "relevant_clue_a": "An excerpt from the textbook containing a base
       definition or rule.",
  "relevant_clue_b": "A second excerpt containing a supplemental
       protocol that can modify the base rule."
}

# OUTPUT ARCHITECTURE
Your output MUST be a single JSON object with two keys:
1. `reasoning`: A string containing your detailed analysis of how
     you applied the clues to the question to derive the answer.
2. `correct_option_letter`: A string containing the capital letter
     of the correct answer (e.g., "A", "B", "C", or "D").
```

## H.7 STUDENT HANDBOOK AND ACADEMIC INTEGRITY TASK

This task focuses on generating high-quality, inference-based multiple-choice questions from student handbooks and academic integrity policies. The resulting dataset is intended to evaluate an agent's long-term memory and simple reasoning abilities. To achieve this, the methodology adapts the multi-agent pipeline from the Core Course Task, converting dense definitional rules into assessment items that require the agent to recall and apply newly introduced procedural rules. This strategic conversion of rules compels the agent to integrate provided policy regulations with classroom instruction.

**Data Sourcing and Preparation**   The generation process begins by extracting a single, self-contained article with definitional rules from an institutional policy document, such as a student handbook or academic integrity code. This approach mirrors the Core Course Task by focusing on reasoning from a specific, provided text segment, ensuring that each generated problem is grounded in a single, verifiable source of truth.

**The Adapted Multi-Agent Generation Pipeline**   The methodology for this task is executed through a robust, two-phase multi-agent pipeline, adapted from the Core Course Task to handle regulatory texts.

The first phase focuses on logical rigor and follows a serial process. First, an 'Architect' agent designs the core problem components ('clue_a', 'clue_b', 'question'). These components then enter an iterative **'verify-correct' loop**. In this loop, a 'Calculator' agent attempts a formal logical derivation. If any ambiguity or inconsistency is found, it generates a flaw report, which triggers a 'Corrector' agent to perform a targeted edit. This validation loop repeats until the problem is proven to have a unique, logically sound solution.

Once the problem's logical core is validated, the second phase, **parallel content augmentation**, begins. In this phase, a 'Tutor' agent and a 'Distractor Designer' agent work concurrently. The 'Distractor Designer' generates cognitively-informed incorrect options, while the 'Tutor' crafts the pedagogical instruction. The 'Tutor''s work is constrained by a specialized preparatory component:

- **Tainted Term Extraction:** A specialized **Tainted Term Extractor** agent identifies critical terms in the source rule ('clue_a') whose direct use would reveal key problem-solving information. This list of "tainted terms" requires the 'Tutor' agent to rephrase these concepts using more abstract equivalents, preventing the direct leakage of critical information while allowing for necessary contextual references.

**Post-Generation Quality Assurance**  The rigorous two-stage verification protocol (Automated LLM-Based Audit and Final Manual Review) is also applied to every generated item. For this task, the Final Manual Review places special emphasis on the nuances of rule-based reasoning. The review focuses on confirming that the combination of the source rule ('clue_a'), the supplemental instruction ('clue_b'), and the given scenario leads to a single, unambiguous conclusion. This validates that the problem's difficulty arises from valid logical inference rather than from any ambiguity in the text or the rules themselves.

### H.7.1 VERBATIM PROMPTS FOR STUDENT HANDBOOK AND ACADEMIC INTEGRITY TASK

---

**Architect · Initial Learning**

```
# Role: You are an expert university curriculum designer,
    specializing in creating assessments that test deep logical
    reasoning and knowledge synthesis.
# Task: You are to architect the foundational components of a
    rigorous, multi-layered logical reasoning problem.
# Your sole responsibility is to invent the clues and the scenario
    for the problem.
# **You MUST NOT solve the problem or provide the answer.**

---
### Core Design Principles

The problem you design MUST adhere to the following principles:

1. **The Two-Key Lock:**
    * The generation process is guided by a core design principle:
        to formulate questions where a solution is \textbf{intended}
        to be reached through the synthesis of information from both
        'clue_a' (the source rule) and 'clue_b' (the invented process
        ). This "two-key lock" objective aims to produce tasks that
        encourage the agent to integrate distinct pieces of
        information, moving beyond simple fact retrieval from a
        single source.
    * It must be IMPOSSIBLE to solve the problem if given only '
        clue_a' or only 'clue_b'.
    * Your 'question' design is the mechanism that enforces this
        lock.

2. **Principle of Deterministic Solvability:**
    * The combination of 'clue_a' and 'clue_b' must form a complete
        and unambiguous logical system.
    * All terms must be clearly defined, and all conditions must
        lead to a single, verifiable outcome without any ambiguity.
    * The problem must be challenging due to the synthesis required,
        but it must be fair and definitively solvable.

3. **Principle of Harmonious Synthesis:**
    * The invented rule in 'clue_b' **MUST NOT** conflict with,
        contradict, or create an exception to the foundational rule
        in 'clue_a'.
    * It must act as a supplementary, subsequent, or parallel
        process that can coexist logically with 'clue_a'.
```

```
---
### Instructions:

**1. Architect Clue A (The Foundational Rule):**
* **Rule Distillation (ENHANCED & CRITICAL):** Your primary task
    here is not to copy, but to **distill**. You must analyze the
    provided `source_content` and extract its single most critical,
    actionable rule.
    * **Focus on Procedure:** Identify the core procedural logic:
        conditions, actions, consequences (e.g., "IF a student is
        late by X days, THEN they must pay Y dollars").
    * **Be Concise:** Remove all narrative fluff, introductory
        phrases, or descriptive prose. The resulting `clue_a` should
        be a clean, direct, and concise statement of the rule.
    * **No Invention Allowed:** While you must rephrase for
        conciseness, you are forbidden from inventing new conditions
        or altering the core logic of the original rule.

**2. Architect Clue B (The Orthogonal Process):**
* **Invent a New Process:** You must invent a NEW, logically deep,
    multi-step process. This will be the value for `clue_b`. This
    can be a **quantitative calculation formula**, a procedural
    algorithm, a priority-based workflow, a decision-making matrix,
    or a series of conditional checks.
* **Complexity Requirement:** The invented process must involve at
    least 3 distinct logical steps or conditions.
* **Principle of Abstract Dependency:** You must create dependency
    without creating information leaks.
    * **Enforce Dependency:** The process in `clue_b` MUST be
        intentionally designed to be unsolvable on its own. It must
        require a specific piece of information or context *derived
        from* `clue_a` to function.
    * **Mandatory Abstraction (The Firewall):** To achieve this
        dependency securely, you are FORBIDDEN from using any key
        numbers, proper nouns, or specific phrases from `clue_a`
        inside `clue_b`. Instead, you MUST "blur" or "abstract" the
        required information by referring to the *outcome* or *
        category* of the rule in `clue_a`.
* **Rule of Self-Containment:** The process you invent in `clue_b`
    MUST be perfectly self-contained. To comply:
    * **Define All Invented Terms:** If you introduce a new term (e.
        g., "tier," "status level"), you MUST define what those terms
         mean and how they are assigned *within the rule itself*.
    * **Specify All Values & Outcomes:** All numbers, percentages,
        or fixed values must be explicitly stated. Every possible
        outcome of a condition must be clearly described.
    * **Leave No Ambiguity:** Avoid vague phrases like "escalate by
        one step." Instead, explicitly define the escalation path.
* **Clarity and Unambiguity:** The process you invent must be self-
    consistent and free of ambiguity.

**3. Architect the Question (The Locking Mechanism):**
* **Design a Scenario:** Create a concise `question` scenario that
    presents a specific case or a set of initial conditions.
* **Introduce Cognitive Friction (Optional):** To enhance the
    reasoning challenge, you may include one piece of plausible-
    sounding but ultimately irrelevant information (a "red herring")
     in the scenario.
* **Enforce Inter-dependency (Key 2):** The scenario in your `
    question` **MUST provide data points that trigger the logic in
    BOTH `clue_a` AND `clue_b`**.
* **Information Purity:** The question scenario itself must not
    contain any of the rules from the clues.
```

```
* **Question Complexity**: Prohibition of pure true/false or yes/no
    questions to prevent answer guessing.

---
### Output Format:

Your output must be a single JSON object with three keys: `clue_a`,
    `clue_b`, and `question`.
---
### EXAMPLE 1 (Non-Computational Reasoning)

**Input**:
{
  "source_content": "Prerequisites for upper-division courses are
      strictly enforced. To register for the course 'Advanced
      Algorithms' (CS401), a student must have successfully
      completed 'Data Structures' (CS301) with a final grade of B-
      or better. No exceptions are granted for this particular
      course."
}

**Your Correct JSON Output**:
{
  "clue_a": "To register for 'Advanced Algorithms' (CS401), a
      student must have a grade of B- or better in 'Data Structures'
       (CS301).",
  "clue_b": "The 'Special Academic Petition Protocol' allows
      students to request a waiver for certain university
      requirements under specific conditions. A student is eligible
      to file a petition only if they are in their final year of
      study AND maintain a cumulative GPA of 3.8 or higher.
      Petitions for course-specific academic prerequisites must also
       be approved by the Head of the Department.",
  "question": "Sarah, a third-year student with a cumulative GPA of
      3.9, wants to register for 'Advanced Algorithms' (CS401). She
      has not completed 'Data Structures' (CS301). She has submitted
       a petition to the Head of the Computer Science Department. Is
       Sarah eligible to register for the course at this time?"
}
---
### EXAMPLE 2 (Quantitative Reasoning)

**Input**:
{
  "source_content": "If late registration occurs within the first
      week (1-7 days) after the initial deadline, you must pay a $50
       late fee. Registering in the second week (8-14 days) requires
       a payment of a $100 late fee. Beyond the second week, a late
      fee of \$200 will be imposed."
}

**Your Correct JSON Output**:
{
  "clue_a": "The late registration fee is $50 for the first week
      (1-7 days), $100 for the second week (8-14 days), and \$200
      thereafter.",
  "clue_b": "The 'Financial Standing Adjustment Protocol' is a two-
      step algorithm applied to student fees. Step 1: If a student
      is a recipient of a university merit scholarship, their
      calculated fee is reduced by 25%. This is applied first. Step
      2: If the student has any prior unresolved financial holds, a
      flat administrative surcharge is added to their fee after any
      reductions from Step 1. The surcharge amount is based on the
```

```
        student's year level: $15 for first-year students, $30 for all
          other students.",
  "question": "A third-year undergraduate student registers for
        classes nine days after the official deadline. This student is
        a recipient of the university's Presidential Merit
        Scholarship and has a prior unresolved financial hold from the
         library. What is the total late registration fee this student
         must pay?"
}
---

**Now, generate the data unit for the following input:**

{input_json}
```

**Calculator**

```
# Role: Logical Reasoning Engine

## Persona:
You are a meticulous and powerful Logical Reasoning Engine. You
    operate with pure, cold logic and are incapable of making
    assumptions.

## Foundational Principles:
1. **Clue A is Immutable Truth**: 'clue_a' contains a foundational
    rule extracted directly from a source document. It is an
    unchangeable fact. You MUST NOT question, contradict, or
    attribute any error to it.
2. **All Flaws Originate from Invention**: Any logical flaw (
    missing information, contradiction, ambiguity) MUST be
    attributed to the invented parts of the problem, which are '
    clue_b' and the 'question'.
3. **No External Knowledge**: You MUST ONLY use the information
    provided.

## Task:
Analyze the given problem, which consists of 'clue_a', 'clue_b',
    and a 'question'. Your goal is to perform a step-by-step logical
     derivation to find the answer. If the problem is unsolvable due
     to flaws in the invented components, you must provide a precise
     diagnostic report.

### Additional Validation
'question' are prohibited from being posed in pure true/false
    format (too simplistic and easily guessed).

## Instructions:
1. **Reasoning Process**:
   * Write down your internal, step-by-step reasoning process.
   * For each step, you MUST explicitly cite which clue ('clue_a'
       or 'clue_b') the information comes from.

2. **Logical Flaw Diagnosis & Reporting**:
   * If the problem is solvable, set 'status' to '"success"'.
   * If any flaw is detected, you MUST set 'status' to '"error"'
       and generate a detailed 'flaw_report' object.
   * **CRITICAL CONSTRAINT**: The 'flaw_location' array MUST ONLY
       contain '"clue_b"' or '"question"'. It is strictly forbidden
       to list '"clue_a"' as a source of error.
```

```
      * Your `correction_suggestion` MUST always instruct the
          Corrector to modify either `clue_b` or the `question`.

   3. **Output Generation**:
      * Your entire output MUST be a single, valid JSON object without
          any additional text.
      * The JSON object MUST contain `reasoning`, `answer`, and `
          status`.
      * If `status` is `"error"`, the `flaw_report` object is
          mandatory.
      * answer must be forbidden from appearing in nested structures;
          it may only exist as a direct string or null value.

   ## Output Format:
   {
     "reasoning": "...",
     "answer": String | null,
     "status": "success" | "error",
     "flaw_report": {
       "flaw_type": "Missing Information" | "Contradictory Information"
          | "Ambiguous Information",
       "flaw_location": ["clue_b" | "question"],
       "flaw_description": "A clear, concise explanation of the flaw.",
       "correction_suggestion": "A specific, actionable instruction to
          modify clue_b or the question."
     }
   }

   ___
   ### Example (Error – Missing Information in Invented Clue)

   #### Input:
   {
     "clue_a": "The standard fee for borrowing a book is \$2.",
     "clue_b": "A 'processing surcharge' is applied to the total book
         fee.",
     "question": "A student borrows 3 books. What is the total cost?"
   }
   #### Output:
   {
     "reasoning": "Logical flaw analysis initiated. Step 1: Calculate
         base book fee from `clue_a` (3 * $2 = $6). Step 2: Attempt to
         apply the 'processing surcharge' from `clue_b`. The process
         fails because the value of the surcharge is not defined.",
     "answer": null,
     "status": "error",
     "flaw_report": {
       "flaw_type": "Missing Information",
       "flaw_location": ["clue_b"],
       "flaw_description": "The problem is unsolvable because 'clue_b'
           introduces a 'processing surcharge' but fails to specify its
           value or calculation method (e.g., a flat amount or a
           percentage).",
       "correction_suggestion": "To fix this, you must define the
           surcharge value. Modify `clue_b` to include a specific amount
           . For example: Change `clue_b` to 'A 'processing surcharge'
           of 5% is applied to the total book fee.'"
     }
   }
```

**Corrector**

```
# Role: Automated Design Editor

## Persona:
You are an automated, rule-based text editor. Your operation is
    purely mechanical. You do not reason, infer, or create. You only
     execute precise instructions on a given text object.

## Unbreakable Directives:
1. **Primary Mandate: Execute Correction**: Your SOLE function is
    to implement the `correction_suggestion` from the `flaw_report`.
     This is not a request; it is your only operational command.
2. **Absolute Immutability of Clue A**: Under NO circumstances will
     you modify, alter, or omit `clue_a`. Any output where `clue_a`
    is not identical to the input is a catastrophic failure. You
    must treat it as a read-only field.
3. **Prohibition of Invention**: You are FORBIDDEN from adding any
    new information, concepts, or rules not explicitly commanded by
    the `correction_suggestion`. You are an editor, not a creator.
4. **Strictly No Commentary**: Your output MUST NOT contain any
    explanations, apologies, or conversational text. Your output
    must be ONLY the raw, valid JSON object.

## Task:
You will be given a JSON object containing a flawed `
    original_design` (`clue_a`, `clue_b`, `question`) and a `
    flaw_report`. Your task is to output a new JSON object that is a
     corrected version of the original, based *exclusively* on the `
    correction_suggestion`.

## Operational Workflow:
1. **Identify Target**: Read the `flaw_report` to identify the
    target of the correction (`clue_b` or `question`).
2. **Execute Edit**: Apply the specific change described in `
    correction_suggestion` to the target field, using `clue_a` as
    necessary context.
3. **Preserve Other Fields**: Copy `clue_a` and any other non-
    target fields from the original design to the new design without
     any changes.
4. **Final Verification (Self-Correction Step)**: Before outputting,
     you MUST perform a final check on your generated JSON to ensure
     it complies with ALL Unbreakable Directives listed above.
    * **Check 1**: Is the `clue_a` in your output IDENTICAL to the
        input `clue_a`? (MUST be YES)
    * **Check 2**: Does your output contain ONLY the keys `clue_a`, `
        clue_b`, and `question`? (MUST be YES)
    * **Check 3**: Is the change you made *only* the one specified
        in `correction_suggestion`? (MUST be YES)

## Output Format:
Your output must be a single, valid JSON object.

---
### EXAMPLE

#### INPUT:
{
  "original_design": {
    "clue_a": "The standard fee for borrowing a book is \$2.",
    "clue_b": "All postgraduate students receive a special discount
        on book fees.",
```

```
   "question": "A postgraduate student borrows 3 books. What is the
       total fee?"
 },
 "flaw_report": {
  "flaw_type": "Missing Information",
  "flaw_location": ["clue_b"],
  "flaw_description": "The problem is unsolvable because 'clue_b'
      mentions a 'special discount' but does not specify its value
      or percentage.",
  "correction_suggestion": "To make the problem solvable, you must
       provide a specific value for the discount. Modify `clue_b`
       by changing it to 'All postgraduate students receive a
       special discount of 10% on book fees.'"
 }
}

#### OUTPUT (Correct):
{
  "clue_a": "The standard fee for borrowing a book is \$2.",
  "clue_b": "All postgraduate students receive a special discount of
      10% on book fees.",
  "question": "A postgraduate student borrows 3 books. What is the
      total fee?"
}
```

**Distractor Designer**

```
# Role: Expert Distractor Designer

## Persona:
You are an expert in cognitive psychology and educational
    assessment. Your specialty is creating high-quality, plausible,
    and pedagogically valuable distractors (incorrect options) for
    multiple-choice questions. You work based on a provided "
    Advanced Distractor Matrix" which categorizes common logical
    fallacies.

## Task:
Given a question, the correct answer, and the reasoning behind it,
    you must design exactly THREE pedagogically valuable, deceptive,
     and qualitatively distinct distractors. Your goal is to
    simulate the most common and logical errors a student might make
    .

## Unbreakable Rules:
1. **No Duplication of Correct Answer (CRITICAL):** The `answer`
    key for each distractor object you generate MUST NOT be
    identical to the `correct_answer` provided in the input. An
    incorrect option that matches the correct answer is a
    catastrophic failure of your task.
2. **Qualitative Distinction of Answers (CRITICAL):** The `answer`
    values for each of the three distractors must be qualitatively
    different from each other. Avoid generating answers that are
    simple numerical or textual variations of one another (e.g., if
    one answer is `10`, another cannot be '"10 hours"'). Each `
    answer` should represent a genuinely unique outcome derived from
     a unique logical error.
3. **Strict Adherence to Schema:** Your output must be a single
    JSON object containing a `distractors` key, which holds a list
```

```
     of exactly three objects, each with `fallacy_type`, `explanation
     `, and `answer`.
4. **No Commentary:** Do not add any text outside of the final JSON
    object.

## Instructions:
1. **Analyze Reasoning**: Carefully study the provided correct
   answer and its step-by-step reasoning to fully understand the
   correct logical path.
2. **Select Plausible Fallacies**: From the fallacy matrix below,
   select the THREE **most plausible** logical fallacies a student
   might commit for this specific problem. Each distractor MUST be
   based on a different fallacy. Prioritize fallacies that reflect
   genuine, common student errors over ones that are technically
   possible but unlikely.
3. **Craft Distinct Distractors**: Craft a distractor for each
   chosen fallacy. Ensure the distractors are **qualitatively
   different**, representing unique error paths. Avoid distractors
   that are just minor numerical variations of each other.
4. **Final Verification (Self-Correction Step):** Before finalizing
    your output, you MUST perform this two-part check:
   * **Check 1 (Correctness):** For each of the three distractors
      you have created, compare its `answer` with the `
      correct_answer` from the input. Confirm that NONE of them are
       the same. If you find a match, you must regenerate that
      distractor.
   * **Check 2 (Uniqueness):** Compare the `answer` values of your
      three generated distractors with each other. Confirm that
      they are all substantially different and not just variations
      of the same outcome. If they are too similar, you must
      regenerate one or more distractors.
5. **Distinct answers**: The values of the four answer options (one
    correct and three incorrect) must be mutually distinct and
   substantively different in nature, rather than merely featuring
   superficial descriptive variations.

---
## Advanced Distractor Matrix:

1. **Partial Algorithm Application**: The student correctly
   executes some steps of the required process but misses or
   ignores other crucial steps.
2. **Recall-Only Fallacy**: The student recalls a single fact or
   number from the clues but fails to synthesize it with other
   information to perform the required calculation or logic.
3. **Logical Branch Error**: The student follows an incorrect
   logical path from the start, misinterpreting a key condition or
   rule.
4. **Red Herring Utilization**: The student is misled by an
   irrelevant piece of information (a "red herring"). **Note: Only
   use this fallacy if the provided problem contains information
   that is explicitly not needed for the solution.**
5. **Sequence Error**: The student applies the correct steps but in
    the wrong order, leading to an incorrect result.
---
## Example:

### Input:
{
  "question": "Calculate the final fee. A service has a base cost of
      \$200. A 10\% discount is applied if the client is a 'premium
      member'. A flat \$25 administrative fee is added to the total
```

```
        *after* any discounts are applied. The client is a 'premium
        member'.",
  "correct_answer": "\$205",
  "reasoning": "Base cost is \$200. Apply 10\% discount (\$200 *
      0.10 = \$20), making it \$180. Then, add the flat \$25
      administrative fee, for a final total of \$205."
}

### Output:
{
  "distractors": [
    {
      "fallacy_type": "Sequence Error",
      "explanation": "This option results from the student applying
          the operations in the wrong order. They correctly identify
          all steps but first add the administrative fee to the base
          cost (\$200 + \$25 = \$225) and then apply the 10\%
          discount to this inflated total (\$225 * 0.9 = \$202.5),
          leading to an incorrect final amount.",
      "answer": "\$202.5"
    },
    {
      "fallacy_type": "Partial Algorithm Application",
      "explanation": "This option arises when the student correctly
          calculates the 10\% discount from the base cost (resulting
          in \$180) but then completely fails to perform the final,
          mandatory step of the algorithm, which is to add the \$25
          administrative fee.",
      "answer": "\$180"
    },
    {
      "fallacy_type": "Logical Branch Error",
      "explanation": "This distractor stems from the student
          fundamentally misinterpreting how a percentage works.
          Instead of calculating 10\% *of the base cost*, they
          incorrectly treat the '10\%' as a simple subtraction of the
           number 10, calculating (\$200 - 10) + \$25. This common
          error path leads to a completely different logical outcome
          .",
      "answer": "\$215"
    }
  ]
}
```

**Tainted Term Extractor**

```
# Role: Prioritized Keyword Extractor

## Task:
You are a precise information extraction agent. Your sole task is
    to read the provided "source_text" and identify the **top one to
     three (1-3)** most critical, specific, and quantifiable pieces
    of information, following the strict rules below. You will then
    return these as a clean, de-duplicated JSON list under the key "
    tainted_terms".

## Rules:

### 1. Extraction Target & Prioritization Hierarchy
- You MUST extract information based on this strict priority order.
    Stop once you have extracted three terms.
```

```
    1. **Priority 1: Specific Nouns (Proper Names)**. Extract names
       of offices, committees, official documents, or specific,
       named statuses (e.g., `Office of the Provost`, `permanent
       notation`, `formal warning`).
    2. **Priority 2: Key Data (Quantifiable Information)**. Extract
       specific monetary values, grades, or precise penalty
       durations (e.g., `\$50`, `grade of F`, `one-week suspension`)
       .
    3. **Priority 3: Specific Timeframes**. Extract precise
       deadlines or action periods (e.g., `48 hours`, `10 business
       days`).
- You MUST NOT extract generic concepts, verbs, or entire clauses (
   e.g., "non-compliance", "sanctions", "violations").

### 2. Output Limitation: Maximum Three Terms
- Your final output list, `"tainted_terms"`, MUST contain a maximum
   of three (3) entries.
- If more than three candidate terms exist, you MUST use the
   prioritization hierarchy from Rule #1 to select the top three
   and discard any lower-priority terms.

### 3. Extraction Method: Be Minimalist & Distill
- All extracted terms must be as concise as possible. Remove non-
   essential surrounding words.
- **Example A**: From `"...a penalty of \$50 is applied..."`, you
   MUST extract `"\$50"`, not `"\$50 penalty"`.
- **Example B**: From `"...failing to follow received
   interpretations..."`, you should extract the core concept `"
   failing to follow interpretations"`.

### 4. Final Output Format
- Your entire output must be a single, valid, de-duplicated JSON
   object with one key: `"tainted_terms"`. Do not include any text
   or explanations.

## Example:
(This example demonstrates the prioritization rule when more than 3
   candidates exist in the source text)

### Input:
{
  "source_text": "You must submit an appeal request to the Office of
     the Provost within 10 business days. Failure to comply will
     result in a final grade of F and a permanent notation on your
     transcript."
}

### Your Correct JSON Output:
(Reasoning: There are 4 candidates: "Office of the Provost" (P1), "
   permanent notation" (P1), "grade of F" (P2), and "10 business
   days" (P3). According to the rules, we must pick the top 3 by
   priority. We take the two P1 items, then the one P2 item. The P3
    item, "10 business days," must be discarded to meet the max-3
   limit.)
{
  "tainted_terms": [
    "Office of the Provost",
    "permanent notation",
    "grade of F"
  ]
}
---
**Now, generate the output for the following input:**
```

```
{input_json}
```

**Tutor**

```
# Role: Expert University Lecturer at Lifelong Agent University

## Task:
Your role is to simulate a lecture at Lifelong Agent University.
    You will be given a specific rule (`clue_a`) from the Student
    Handbook, along with its location (`chapter_title`, etc.). You
    will also be given a new, supplemental rule (`clue_b`). Your
    core task is to act as a lecturer explaining this supplemental
    rule to students by synthesizing all provided information into a
    single, cohesive teaching paragraph (`instruct`), following the
    strict constraints below.

## Instructions:

### 1. The "Tainted Terms" Blacklist (CRITICAL & NON-NEGOTIABLE)
- You are provided a JSON list named `tainted_terms`. You are
    strictly forbidden from using any term from this list.
- To refer to the concept of a tainted term, you MUST use a generic,
    abstract equivalent.

### 2. Rigid Three-Part Paragraph Structure
Your `instruct` text MUST be a single paragraph composed of the
    following three parts, executed in sequence without deviation.
- **Part A: Recall & Anchor (Concise Hierarchical Citation)**:
  1. Begin with a single, concise sentence that establishes the
      hierarchical path to the existing rule in the handbook by
      citing the `chapter_title`, `section_title`, and `
      article_title`.
  2. Follow this with an abstract reference to the conceptual area
      of the existing handbook rule (`clue_a`).

- **Part B: Teach & Detail**:
  1. Transition from Part A to the new, supplemental material.
  2. Explain the new, supplemental rule from `clue_b`, conveying
      all substantive information without omission.

- **Part C: Apply & Question (Natural Transition & High-Fidelity
    Reproduction)**:
  1. Create a brief and natural transition from the explanation in
      Part B into the application scenario. Good examples include
      "Now, let's apply this to a specific case:", "To see how this
      works in practice, consider the following:", or "To put this
      into perspective, imagine this situation:".
  2. Accurately reproduce the scenario and the interrogative
      question(s) from the `question` input.
  3. Your reproduction MUST maintain the same substantive content
      and the same number of logical questions as the original. You
      MUST NOT add, omit, or change the core substance of what is
      being asked. Minimal, natural-sounding rephrasing for
      narrative flow is permitted as long as the logical integrity
      of the question is perfectly preserved.

### 3. Formatting Constraints
- The entire `instruct` text MUST be a single, continuous block of
    plain text.
```

```
- You MUST NOT use any markdown or formatting symbols. This
   includes but is not limited to:
   - Bolding (`**text**`)
   - Italics (`*text*`)
   - Bullet points (`*`, `-`)
   - Line breaks (`\n`)

### 4. Final Consistency Checks
Before generating the final JSON, you must mentally verify these
   seven points:
1. **Formatting**: Is the output a single block of plain text with
   absolutely no formatting symbols?
2. **Tainted Terms**: Is the `instruct` text free of any tainted
   terms?
3. **Completeness (clue_b)**: Has every detail from `clue_b` been
   included?
4. **Fidelity (Question)**: Does the reproduced question have the
   same substantive content and number of logical questions as the
   input?
5. **Unified Question**: Does the `instruct` text end with a single,
    non-compound question?
6. **Structure**: Does the paragraph strictly follow the A-B-C
   structure?
7. **Concise Hierarchy**: Does the introductory sentence concisely
   establish the hierarchical path?

## Output Format:
Your final output must be a single JSON object with one key: `
   instruct`.

---
**Example**

### Input:
{
  "chapter_title": "Chapter 4: Curriculum and Academic Performance",
  "section_title": "4.2 Examination Systems",
  "article_title": "Dean's List Qualifications",
  "clue_a": "To be eligible for the Dean's List, an undergraduate
      student must achieve a semester GPA of at least 3.7, complete
      a minimum of 12 graded credit hours, and must not have any
      unresolved disciplinary actions.",
  "clue_b": "The new 'Dean's List Second Chance' protocol allows
      students to petition for eligibility. If a student's semester
      GPA is between 3.60 and 3.69, they can have one grade from a
      non-major course (up to 4 credits) excluded from the GPA
      calculation for the Dean's List eligibility check, provided
      they have no other grade below a B in that semester.",
  "question": "A student, Sarah, completed 15 credit hours this
      semester with no disciplinary issues. Her grades are: A (4
      credits, major), A (4 credits, major), B+ (3 credits, major),
      B (3 credits, non-major), and C (1 credit, non-major). Her
      calculated semester GPA is 3.53. How should Sarah's final
      eligibility for the Dean's List be determined under the full
      scope of university policy, including all supplemental
      protocols?",
  "tainted_terms": ["GPA of at least 3.7", "12 graded credit hours",
      "unresolved disciplinary actions"]
}

### Your Correct JSON Output:
{
```

```
  "instruct": "Good morning. In the student handbook, Chapter 4,
      Section 4.2, the article on 'Dean's List Qualifications'
      establishes the primary requirements for this academic honor,
      which include specific thresholds for academic performance,
      credit load, and student conduct. Now, I want to detail a new
      supplemental policy called the 'Dean's List Second Chance'
      protocol, which creates a narrow path for students to petition
       for eligibility. This protocol specifies that if a student's
      semester GPA is between 3.60 and 3.69, they can request to
      have one grade from a single non-major course, worth up to 4
      credits, excluded from their GPA calculation just for this
      eligibility check. However, this option is only available if
      the student has no other grades below a B in that semester. To
       see how these rules interact, let's analyze a specific case:
      A student, Sarah, completed 15 credit hours this semester with
       no disciplinary issues. Her grades are: A (4 credits, major),
       A (4 credits, major), B+ (3 credits, major), B (3 credits,
      non-major), and C (1 credit, non-major). Her calculated
      semester GPA is 3.53. How should Sarah's final eligibility for
       the Dean's List be determined under the full scope of
      university policy, including all supplemental protocols?"
}
---

**Now, generate the output for the following input:**

{input_json}
```

## Automated LLM-Based Audit

```
# CONTEXT
You are a meticulous and logical adjudicator AI. Your function is
    to analyze a scenario and determine the correct outcome based on
     a set of rules.

# TASK
Your primary task is to generate a single, valid JSON object as
    your output. This object must contain your reasoning and the
    final correct option letter. To do this, you must analyze the `
    question` and apply the provided rules from `clue_a` and `clue_b
    `.

# INSTRUCTIONS
1. **Synthesize Knowledge**: Your reasoning should be based on a
    synthesis of the information found in `clue_a` (the base rule)
    and `clue_b` (the special protocol).
2. **Rule Priority**: The protocol in `clue_b` is a higher
    authority. If the conditions described in `clue_b` are met in
    the `question`, its rules override any conflicting information
    in `clue_a` or your general knowledge.
3. **Conditional Application**: The special protocol in `clue_b`
    may not always be relevant. You must first assess if the
    scenario in the `question` triggers its application. If it is
    not triggered, your decision should rely solely on `clue_a`.
4. **Determine the Best Answer**: After analyzing the rules, you
    must choose the single best option from the list that correctly
    reflects the outcome.
5. **Show Your Work**: Your reasoning must be detailed and explicit.
     Do not omit any steps in your logical deduction or calculations
    . Every step of your thought process must be written out.
```

```
# FINAL REMINDER
CRITICAL: Ensure your `reasoning` string includes every single step
    of your analysis. Do not skip any part of your logical or
    computational process. Your thought process must be transparent
    and fully documented.

# INPUT
You will be provided with a JSON object containing the context for
    a single problem:
{
  "question": "A specific scenario to be evaluated.",
  "options": { "A": "...", "B": "...", "C": "...", "D": "..." },
  "clue_a": "The base rule or set of standard regulations.",
  "clue_b": "A special protocol with specific trigger conditions
      that modifies the base rule."
}

# OUTPUT ARCHITECTURE
Your output MUST be a single JSON object with two keys:
1. `reasoning`: A string containing your detailed analysis of how
    you applied the rules to the question to reach a conclusion.
2. `correct_option_letter`: A string containing only the capital
    letter of the correct option (e.g., "A", "B", "C", or "D").
```

## H.8 EXAMS

The agent's performance on this task is evaluated through two comprehensive assessments: a midterm and a final exam. These assessments are based on a comprehensive data pool of items generated by the Core Course Task, covering 8 distinct subjects. Both assessments are constructed from this data pool using a randomized algorithm to measure the agent's capacity for long-term memory and knowledge application. To succeed, the agent is required to synthesize and recall learned rules to solve complex problems derived from the course material.

**Data Partitioning and Sampling** The data pool is partitioned based on the course's progression. Material corresponding to the first half of the curriculum is allocated to the Midterm Exam, while the remaining material is reserved for the Final Exam. For a given subject's exam section, a working set of items is first randomly sampled from its designated pool. This set then serves as the exclusive source material for constructing that subject's questions. This method of data partitioning ensures that all information required to solve the problems is present within the textbook content and course instruction.

**Composite Question Formulation** The construction of each composite question begins by organizing the sampled items for a subject into groups. Each group provides the foundation for a single question, with each of its items being used to formulate one of the multiple-choice options. Within each group, one item is designated to generate the correct option, while the others are used to create plausible distractors. The text for each option is systematically constructed by combining the source item's question (context) and its value (conclusion) with a standard connector phrase. This process yields a coherent statement presenting a complete scenario and its outcome, ensuring all options are structurally parallel.

**Post-Generation Quality Assurance** Each exam undergoes a two-stage verification protocol to ensure that every question has a single, unambiguous correct answer.

- **Automated LLM-Based Audit:** The first stage is an automated audit by an independent LLM instance. Without foreknowledge of the designated answer, the LLM is tasked with deducing the correct option from the four choices based on the provided source rules. A question is considered validated if the LLM's selection aligns with the correct answer.

- **Final Manual Review:** The second stage involves a manual review by human reviewers. They verify that each question possesses a single, unambiguous correct answer and assess the linguistic clarity of all options to eliminate potential ambiguities. This step is essential for guaranteeing the fairness and validity of each question.

### H.8.1 VERBATIM PROMPTS FOR EXAMS

**Automated LLM-Based Audit**

```
# CONTEXT
You are a highly precise and logical AI Exam Proctor. Your role is
    to solve a multiple-choice exam question by synthesizing all
    available information.

# TASK
Your goal is to analyze the `exam_question`, its `options`, and a
    comprehensive set of `context_clues_for_all_options` to
    determine the single correct answer. Your output must be a
    single, valid JSON object containing your reasoning and the
    letter of the correct option.

# INSTRUCTIONS
1. **Holistic Analysis**: You will be given a collection of context
     clues, with each option letter mapping to its own set of clues
     (`clue_a` and `clue_b`). You must consider all of this
     information to understand the full context of the question and
     evaluate each option.
2. **Rule Priority**: For any given option's context, its special
     protocol (`clue_b`) is a higher authority. If the conditions
     described in `clue_b` are met by the scenario in that option,
     its rules override the corresponding `clue_a`.
3. **Synthesize and Select**: Analyze each option against its
     relevant clues and the overarching question. After evaluating
     all options, determine which one is the single, most accurate
     answer.
4. **Provide a Single Answer**: You must choose only one option as
     the correct answer.
5. **Show Your Work**: Your reasoning must be exhaustive. Explain
     your analysis for each option and how you came to your final
     conclusion. If you perform any calculations, you must show all
     the steps. Do not omit any details.

# FINAL REMINDER
CRITICAL: Ensure your `reasoning` string includes every single step
     of your analysis. Do not skip any part of your logical or
     computational process. Your thought process must be transparent
     and fully documented.

# INPUT
You will receive a JSON object containing the entire context for
    one exam question:
{
  "exam_question": "The overarching question text.",
  "options": {
   "A": "Text for option A.",
   "B": "Text for option B.",
   "C": "...",
   "D": "..."
  },
  "context_clues_for_all_options": {
   "A": {
     "clue_a": "Base rule relevant to option A.",
```

```
        "clue_b": "Special protocol relevant to option A."
      },
      "B": {
        "clue_a": "Base rule relevant to option B.",
        "clue_b": "Special protocol relevant to option B."
      },
      "...": "..."
    }
  }

  # OUTPUT ARCHITECTURE
  Your output MUST be a single JSON object with two keys:
  1. `reasoning`: A string containing your detailed analysis of how
     you evaluated all the options and their clues to arrive at your
     final answer.
  2. `correct_option_letter`: A string containing only the capital
     letter of the single best option (e.g., "A", "B", "C", or "D").
```

## I  ANALYSIS OF TASK SOLVABILITY AND POTENTIAL BIAS

To address concerns regarding potential generation bias or data leakage in our benchmark tasks, we conducted a "Perfect Context" analysis. This experiment is designed to isolate the tasks' inherent solvability from the core lifelong learning challenges, specifically long-term memory and proactiveness.

In this analysis, we evaluated several models and a human baseline on the core In-Class and Examination tasks. For each task, the agent was provided with the "perfect context"—the specific ground-truth information (e.g., excerpts from lecture notes or the student handbook) required to derive the correct answer. This setup effectively removes the bottlenecks of memory retrieval and proactive scheduling, testing only the agent's reasoning and comprehension capabilities given ideal information.

Table 7: Success Rate (%) on tasks with "Perfect Context" provided. This analysis isolates task solvability from the lifelong memory challenge. Models achieving near-perfect performance are highlighted in bold.

| Model | In-Class | Exam | Total |
|---|---|---|---|
| DeepSeek-V3 | 97.01 | 88.12 | 94.13 |
| Qwen3-8B | 89.82 | 75.00 | 85.02 |
| Qwen3-235B-A22B | 96.41 | 86.88 | 93.31 |
| **Gemini-2.5-pro** | 97.60 | 96.88 | 97.37 |
| **GPT-5** | **98.50** | **97.50** | **98.18** |
| Human | 88.62 | 82.50 | 86.64 |

The results, presented in Table 7, demonstrate two key findings:

- **High Solvability:** When provided with perfect context, state-of-the-art models, exemplified by GPT-5 (98.18%) and Gemini-2.5-pro (97.37%), achieve near-perfect success rates. This performance significantly exceeds the human baseline (86.64%), indicating that the tasks are inherently solvable, well-defined, and free from significant generation bias or data leakage.
- **Identifying the Bottleneck:** This near-perfect performance on QA tasks (e.g., 98.50% In-Class and 97.50% Exam for GPT-5) is starkly higher than the models' scores on the tasks in the full benchmark (e.g., GPT-5's 7.78% In-Class and 16.88% Exam success rates, respectively, as reported in the main paper).

This analysis strongly suggests that the tasks are solvable. This finding, combined with our use of reverse generation and rigorous human verification, provides further evidence against significant inherent bias from the generation process. This suggests that the core challenge measured by *StuLife*

lies not primarily in the comprehension of the tasks themselves, but rather in the agent's ability to autonomously find, manage, and retrieve the correct context from a persistent, long-term memory over a simulated period. This is the primary bottleneck for current models and the central focus of our benchmark.

