# OpenReview forum: "A Benchmark for Self-Evolving Agents via Experience-Driven Lifelong Learning"
_ICLR.cc/2026/Conference — Submitted to ICLR 2026_

### Official Review · Reviewer_aa6n · 2025-10-31

**Soundness:** 3
**Presentation:** 2
**Contribution:** 2
**Rating:** 4
**Confidence:** 5

**Summary:**

This paper introduces StuLife, a benchmark for evaluating self-evolving agents through assessing the capabilities of experience-driven lifelong learning methods.

**Strengths:**

StuLife offers a comprehensive and realistic evaluation framework that includes four key paradigm shifts: From Simulation to Reality, From Passive to Proactive, From Context to Memory, and From Imitation to Learning.

**Weaknesses:**

1. Why are there only large models as agents? Many existing agents in the field of embodied AI can also achieve the ability to self-evolve, but this paper does not consider work related to self-evolving agents. These self-evolving agents also belong to experience-driven lifelong learning methods.
2. There are doubts about the fairness of the comparison or the conclusions of this benchmark.

**Questions:**

1. Why are there only large models as agents? Many existing agents in the field of embodied AI can also achieve the ability to self-evolve, but this paper does not consider work related to self-evolving agents. These self-evolving agents also belong to experience-driven lifelong learning methods.
2. How does the benchmark ensure that StuGPA and other metrics isolate lifelong learning from task-specific memorization?
3. Can the authors provide a deeper analysis of why models fail in proactive tasks, such as the underlying causes for low PIS scores?
4. How was bias controlled in the data generation process, particularly when using LLMs for creating tasks and solutions?
5. What measures were taken to ensure fair comparison between models with varying architectures and training data?
6. How does the benchmark account for scalability and adaptability to unseen tasks, ensuring it evaluates general self-evolution rather than overfitting to the scenario?
7. Why are the evaluation metrics fair? Why can they be applied to all tasks?

---

> ### Author Response · Authors · 2025-11-20
> **Response to Reviewer aa6n (Part 1 of 3)**
>
> We thank the **reviewer aa6n** for their time and their "absolutely certain" assessment, which brings an important perspective from the Embodied AI field. We are glad they recognized our framework as **"comprehensive and realistic"** and **appreciated our four key paradigm shifts.** We have revised the paper content based on your suggestions. For your convenience, we have highlighted our modifications in blue font in the Supplementary Material.
> We address all questions below.
>
> **On the inclusion of Embodied AI and self-evolving agents (W1, Q1)**
>
> - **We agree this is an excellent point and have added new experiments to address it.** Our initial experiments focused on SOTA LLM-based agents, as they represent the most common current approach to general autonomous agents.
> - **We added new experiments evaluating self-evolving agents using in Embodied and self-evolving domains** to demonstrate our benchmark's flexibility. We added a new experiment to demonstrate this capability. We evaluated two self-evolving methods (**AWM** [1] and **Reflexion** [2]) on our benchmark, using Qwen3-235B-A22B as the base model.
> - The results are shown below. We observe modest improvements (e.g., AWM increases StuGPA from 16.03 to 17.81 and Total Success from 8.52 to 10.12). This confirms our benchmark is fully capable of evaluating such experience-driven methods. However, the limited gains also highlight that these existing methods, while effective in their original domains (e.g., coding, embodied tasks), do not yet solve the complex, long-term cognitive and planning challenges in our benchmark, validating its difficulty.
>
> Table: The performance of various Self-Evolving methods based on Qwen3-235B-A22B.
>
> | Method | StuGPA | LTRR | PIS | In-Class (Success) | In-Class (AvgTurn) | Daily Campus (Success) | Daily Campus (AvgTurn) | Exam (Success) | Exam (AvgTurn) | Total (Success) | Total (AvgTurn) |
> | :--- | :--- | :--- | :--- | :--- | :--- | :--- | :--- | :--- | :--- | :--- | :--- |
> | **Vanilla** | 16.03 | 5.42 | 1.80 | 2.10 | 18.71 | 10.34 | 17.17 | 16.88 | 10.75 | 8.52 | 16.95 |
> | **Self-Evolving** | | | | | | | | | | | |
> | AWM | **17.81** | **6.87** | 1.80 | 1.50 | 15.66 | **14.38** | 14.23 | 16.25 | **7.68** | **10.12** | **13.96** |
> | Reflexion | 16.18 | 5.60 | 1.44 | 0.90 | 15.66 | 12.13 | 15.05 | 16.88 | 9.08 | 8.95 | 14.54 |
>
>
> **On isolating lifelong learning from memorization (Q2)**
>
> This is a crucial point. We ensure our metrics (StuGPA, PIS, LTRR) are fair and measure true lifelong learning (not memorization) in two primary ways:
> - **Strict Benchmark Construction:** As detailed in Section 2.2 and Appendix H, our rigorous 3-step design creates explicit, complex task dependencies. A task's success often depends on learned skills from non-adjacent prior tasks, not just simple memorization.
> - **Evidence from RAG:** As shown in Section 3.2, applying a "vanilla RAG" (a form of simple memorization) worsens performance. This key finding demonstrates that simple memorization is insufficient and agents must intelligently manage knowledge, confirming our metrics are a faithful reflection of advanced learning abilities.
>
> **On the failure analysis for proactive tasks (low PIS scores) (Q3)**
>
> We have expanded **Appendix D** with additional case studies and human baselines to provide a deeper analysis.
> - **Our analysis suggests the primary reason is that current LLM agents are fundamentally stateless and reactive.** They lack the internal mechanisms for: (1) **Self-Initiation:** They wait for instructions rather than proactively identifying and initiating goals. (2) **Future Planning:** They struggle to formulate long-term plans and execute them over time. (3) **Proactive Memory Management:** They fail to proactively retrieve, rehearse, or plan their memory usage in anticipation of future needs.
> - **We substantiate this deficiency in proactivity through specific case studies in Appendix D. Case 2** illustrates a fundamental failure in associative self-initiation: even when explicitly notified of the time (e.g., 16:00), agents often fail to connect this external trigger to their internal schedule, defaulting to random hallucinations. Crucially, our newly added **Case 5** reveals a deeper failure in **proactive preparation** compared to human baselines. Humans exhibit "proactive foresight" by anticipating future memory bottlenecks and explicitly recording complex details (e.g., waypoints) into tools. In contrast, LLMs remain passive, storing only vague headers. Thus, the low PIS scores are driven by a dual failure in proactive agency: the **inability to proactively initiate actions** (Case 2) and the **inability to strategicially plan and manage memory** for long-term execution (Case 5)

---

> ### Author Response · Authors · 2025-11-20
> **Response to Reviewer aa6n (Part 2 of 3)**
>
> **On controlling data generation bias (Q4)**
>
> We used a multi-step process to control bias:
> - **Reverse Generation:** As detailed in the paper (Section 2.2 and Appendix H), we first generated "answers" and "task states" and then prompted the LLM to generate a corresponding instruction. This prevents the LLM's own knowledge from biasing the solution.
> - **Human Verification:** All tasks were rigorously checked by human annotators for correctness, clarity, and bias.
> - **"Perfect Context" Analysis:** To further validate this, we ran a new analysis. We evaluated models (and humans) on core exam tasks, giving them "perfect context." In this analysis, "Perfect Context" refers to providing the agent with all ground-truth information necessary to complete a task, thereby bypassing the challenges of long-term memory retrieval and proactiveness (**detailed in Section 2.2, Step 3**). The results (table below) show that: (1) **When provided with perfect context, top models like GPT-5 and Gemini-2.5-pro achieve near-perfect accuracy** (97-98%) on the benchmark's QA tasks. (2) **This performance is far higher than their scores in the full benchmark**, proving that the tasks themselves are **solvable and not inherently biased.** (3) **The core difficulty, therefore, lies in the lifelong aspect**: the agent's ability to autonomously find, manage, and retrieve this context over time, which is exactly what our benchmark is designed to measure. We have added this analysis to Appendix I.
>
> Table: Success Rate on QA tasks with "Perfect Context".
>
> | Model | In-Class | Exam | Total |
> | :--- | :--- | :--- | :--- |
> | DeepSeek-V3 | 97.01 | 88.12 | 94.13 |
> | Qwen3-8B | 89.82 | 75.00 | 85.02 |
> | Qwen3-235B-A22B | 96.41 | 86.88 | 93.31 |
> | Gemini-2.5-pro | 97.60 | 96.88 | 97.37 |
> | GPT-5 | **98.50** | **97.50** | **98.18** |
> | Human | 88.62 | 82.50 | 86.64 |
>
> - **This demonstrates the tasks are not biased**; the difficulty lies in the lifelong learning and memory aspects, which is precisely what we aim to measure.
>
>
> **On ensuring fair comparison (Q5)**
>
> We ensure fairness through two main strategies: benchmark design and a standardized evaluation protocol.
> - **We enforce fairness through a strict and standardized evaluation protocol that is applied identically to all models.** Every agent is interfaced through a unified API, receives a consistent system prompt, and uses identical request parameters (like temperature). Furthermore, each evaluation run begins from a consistently initialized and fully isolated environment. This technical consistency ensures that all results are directly comparable and that performance differences are attributable to the agent's capabilities, not variations in the experimental setup.
> - **This fairness extends to different learning paradigms.** The main models in our paper are evaluated in a context-based setting and do not require training on our benchmark. For the new self-evolving methods we added (like AWM and Reflexion), which do learn from experience, we ensure they are trained or updated using the identical data and interaction history generated from the environment. This allows for a fair and direct comparison between both non-training and training-based approaches.

---

> ### Author Response · Authors · 2025-11-20
> **Response to Reviewer aa6n (Part 3 of 3)**
>
> **On generalizability and preventing overfitting (Q6)**
>
> The benchmark is explicitly designed to evaluate general self-evolution .
> - **This is achieved through its high diversity:** the 10 different task types require a broad range of skills, meaning an agent must demonstrate general competence, not just memorization of one scenario.
> - **Furthermore, the tasks are highly interconnected and stateful.** An agent's actions have cascading, long-term consequences, requiring true adaptation to an evolving state rather than overfitting to static, isolated problems.
> - **Our experimental results provide strong empirical evidence that the benchmark is not easily overfitted.** As our results show (Table 3 and 4), even the most advanced SOTA models like GPT-5 (17.90 StuGPA) and the enhanced self-evolving agents (AWM score of 17.81) perform poorly, with scores below 20. This significant performance gap clearly demonstrates that the benchmark presents a deep and unsolved challenge that cannot be "gamed" or easily overfitted by current methods.
>
> **On the fairness of evaluation metrics (Q7)**
>
> - **Our evaluation metrics comprehensively capture the core dimensions of Experience-Driven Lifelong Learning.** Beyond simple task accuracy, StuLife measures StuGPA (a holistic performance score), LTRR (memory durability), PIS (intrinsic motivation), and efficiency (average turns usage). This multi-dimensional design ensures that agents are assessed not just on “what” they accomplish, but “how” they learn, remember, and act autonomously over time—reflecting the full spectrum of lifelong learning.
> - **Every task is explicitly annotated as requiring (or not requiring) long-term memory or self-motivation during benchmark construction (Section 2.2).** Consequently, metrics like Long-Term Retention Rate (LTRR) and Proactive Initiative Score (PIS) are only computed on the subset of tasks that truly demand those capabilities, ensuring no misalignment or unfair penalization.
> - **All tasks share a unified and objective success criterion that enables consistent and automatable evaluation.** Regardless of task type, whether enrolling in courses, attending exams, or booking library seats, success is defined by whether the agent achieves a verifiable, environment-grounded goal (e.g., correct calendar entries, accurate answers, proper resource booking). This binary success/failure judgment is deterministic and computable, allowing fair comparison across heterogeneous tasks.
>
> We thank the reviewer again for their expert feedback, which has helped us validate the generality of our benchmark with new, relevant experiments.
>
> [1] Wang Z Z, Mao J, Fried D, et al. Agent workflow memory[J]. arXiv preprint arXiv:2409.07429, 2024.
>
> [2] Shinn N, Cassano F, Gopinath A, et al. Reflexion: Language agents with verbal reinforcement learning[J]. Advances in Neural Information Processing Systems, 2023, 36: 8634-8652.

---

> ### Author Response · Authors · 2025-11-26
>
> Dear Reviewer aa6n,
>
> I hope this message finds you well. As the discussion period is nearing its end with **less than seven days remaining**, I wanted to ensure we have addressed all your concerns satisfactorily. If there are any additional points or feedback you'd like us to consider, please let us know. Your insights are invaluable to us, and we're eager to address any remaining issues to improve our work.
>
> Thank you for your time and effort in reviewing our paper.
>
> Best regards,
>
> The Authors

---

> ### Author Response · Authors · 2025-11-29
> **Summary of Response to Reviewer aa6n (For AC)**
>
> Dear Area Chair,
>
> To facilitate your assessment, we provide a concise summary of how we have addressed Reviewer aa6n’s key concerns regarding the inclusion of self-evolving agents, bias control, and evaluation fairness. We have **resolved all major questions** through new experiments and expanded analyses:
>
> **On the inclusion of self-evolving agents:**
> - We conducted new experiments with **AWM and Reflexion**, demonstrating that our benchmark effectively evaluates experience-driven methods while highlighting the difficulty of long-term planning (Response Part 1).
>
> **On the failure analysis for proactive tasks:**
> - We expanded **Appendix D** with detailed **case studies** and **human baselines**, identifying the lack of internal foresight and proactive preparation as the primary causes for low PIS scores (Response Part 1).
>
> **On controlling data generation bias:**
> - We added a **"Perfect Context" analysis (Appendix I)** and **human baseline** showing that SOTA models achieve ~98% accuracy when memory constraints are removed, confirming that task difficulty stems from lifelong learning challenges rather than bias (Response Part 2).
>
> **On ensuring fairness, validity, and generalizability:**
> - We rigorously validated that our metrics isolate true lifelong learning from naive memorization via RAG baselines (Response Part 1), enforced a standardized protocol for fair comparison (Response Part 2), and demonstrated that the diverse, interconnected task design prevents overfitting and ensures generalizability (Response Part 3).

---

### Official Review · Reviewer_8ybd · 2025-10-31

**Soundness:** 2
**Presentation:** 2
**Contribution:** 2
**Rating:** 2
**Confidence:** 4

**Summary:**

This paper proposes a benchmark for evaluating "self-evolving agents" that learn from experience. The benchmark -- StuLife -- simulates a college education, and involves several tasks presented to a language model in sequence. Experiments with a variety of language models show that such models can systematically struggle in certain tasks.

**Strengths:**

- The experiments provide results with various models, and provides a thorough breakdown of different measurements of model quality with respect to the benchmark.

**Weaknesses:**

- The writing in this paper is oftentimes unclear and not well supported
- Unclear value in the benchmark. While it is motivated from a good place, it is almost entirely heuristic and does not clearly demonstrate that it is a useful benchmark to drive real-world progress.
- Several of the properties and motivating principles for the benchmark are contradictory.

**Questions:**

- Table 2: What does it mean for a task type to be self evolving? The authors should define self-evolving, as it is used throughout the paper and the title but never defined. What does it mean for a benchmark to evaluate self-evolving?
- Section 3.2: In what way do the agents in this task learn from experience? It seems like all the models are frozen, and only used for inference?
- "From simulation to reality": This is not an accurate statement as the proposed benchmark is still a simulation, not reality.
- "From imitation to learning": As stated, this is a false dichotomy. Imitation is a form of learning. While the authors go on to clarify that the agent learn generalizable skills, the nuance of this is not at all reflected, nor does it disqualify the learning of such skills from imitation.
- "From context to memory": Again this feels like an orthogonal dichotomy. Given that you are evaluating language models, their context is their memory and can, in principle, be made arbitrarily large.
- "From passive to proactive": Several reinforcement learning environments already involve such distinctions.
- Table 1: What does it mean for a task to require "long term memory" or "self-motivation"?
- Table 4: Why is it that several modifications (such as vanilla RAG) actually worsen the model according to these benchmarks?
- StuLife is often referred to as a dataset, but you discuss it like an environment. It is never exactly specified what it is, or how the agent is interfaced with it. For example, how is time simulated?
- "but still fall short in supporting lifelong learning, skill abstraction, and longitudinal growth."
  While statements like this are found in the results, they are not fully justified, explained or investigated in detail How are these different axes actually compared?
- Other statements are made without explanation: "rather than tracking continuous growth or self-evolution"
  What does continuous growth mean in this context? How about "self-evolution"?
- There is little intuition for a competitive reference level of performance. Is it possible to evaluate a human on this benchmark as a hypothetical ideal for each metric, like "GPA"?

---

> ### Author Response · Authors · 2025-11-20
> **Response to Reviewer 8ybd (Part 1 of 2)**
>
> We thank the **reviewer 8ybd** for their time and for providing a breakdown of their concerns. We are encouraged that the reviewer acknowledged our **"thorough breakdown of different measurements"** and **"results with various models."** We have revised the paper content based on your suggestions. For your convenience, we have highlighted our modifications in blue font in the Supplementary Material.
>
> **However, we must respectfully but firmly disagree with the reviewer's core assessment.** We believe the reviewer has several significant factual misunderstandings about our work, which we will correct below. Many of the concerns raised (e.g., undefined terms, agent interfacing, simulation details) are, in fact, explicitly and extensively detailed in the paper and its appendices.
>
> We will address the reviewer's questions and "weaknesses" in order.
>
> **On the definitions of "Self-Evolving Agents" and "Continuous Growth" (W2, Q1, Q11)**
>
> - **The reviewer's claim that 'self-evolving' is 'never defined' is factually incorrect.** "Self-Evolving Agent" is a rapidly growing research area with a well-established consensus [1, 2]. More importantly, we explicitly define this term and our use of it in **Appendix A.2**.
> - **'Continuous growth' is used in this context** to refer to the agent's ability to acquire and refine skills over its "lifelong" evaluation period, as opposed to static, one-shot evaluations.
>
> **On how agents "learn" with frozen weights (W2, Q2)**
>
> - **The reviewer incorrectly assumes that "learning" must equate to "weight updates."** This misunderstands the modern LLM-based agent paradigm.
> - **A large body of work demonstrates that "self-evolution" can be achieved without weight updates**, by evolving the agent's context, memory, and skill libraries [3, 4, 5]. Our paper primarily analyzes this dominant "contextual engineering" approach.
> - **This is an Agent Benchmark, not a static NLP task.** The "learning" is measured by the agent's ability to update its internal state (memory, skills) to perform better on future tasks.
> - **Our benchmark does permit weight updates.** It is fully compatible with on-policy or adaptive weight update methods. We demonstrated this by applying Rejection Sampling Fine-Tuning (RFT) to Qwen3-8B, which improved its StuGPA from 13.31 to 15.43. **This confirms the benchmark supports parameter-based skill internalization.**
>
> | Model | Method | StuGPA | LTRR | PIS | Total Success |
> | :--- | :--- | :--- | :--- | :--- | :--- |
> | **Qwen3-8B** | Vanilla | 13.31 | 4.33 | 0.54 | 6.71 |
> | | + RFT (Weight Update) | **15.43** | **4.51** | **0.90** | **8.63** |
>
> **On the guiding principles of the benchmark (W3, Q3-Q6)**
>
> - **We find the reviewer's claims of 'contradiction' are based on overly literal or confused interpretations.**
> - **On "From simulation to reality" (Q3)**: This principle signifies our work's significant step closer to real-world complexity (like AlfWorld [6]) compared to traditional, static QA datasets. The reviewer's semantic argument that "it is still a simulation" would invalidate almost all benchmarks.
> - **On "From imitation to learning" (Q4):** We use "imitation" to refer to overfitting task patterns, and "learning" to mean acquiring reusable, generalizable skills. We believe this is a meaningful and non-false distinction for a benchmark.
> - **On "From context to memory" (Q5):** The reviewer's statement that "their context is their memory" is an oversimplification. Context is working memory. A robust memory system also involves retrieval, synthesis, and internalization of knowledge outside the limited context window [7, 8, 9]. Our benchmark is designed to test these more complex memory systems.
> - **On "From passive to proactive" (Q6):** The reviewer fundamentally confuses training with evaluation. Proactiveness in RL training involves designing a reward function. Our benchmark evaluates the innate, test-time proactiveness of an agent (e.g., self-initiating planning) without an explicit reward, which is a novel aspect not measured by existing environments.
>
> **On definitions for Table 1 (Q7)**
>
> - **The reviewer's claim that these terms are undefined is incorrect. They are explicitly defined in the main text and appendix.** We define these terms and their instantiation clearly in **Lines 83-100 and Lines 316-322, and Appendix B**. In brief: "Long-term memory" tasks require information from non-adjacent, preceding tasks. "Self-motivation" refers to self-initiated tasks (planning, exploration) rather than being a direct response to a new instruction.
>
> **On the performance of vanilla RAG (Q8)**
>
> - **This is not a flaw; it is a key finding of our paper.** Our tasks are long, complex, and have strong causal relationships. A "vanilla" RAG system introduces noisy, out-of-context information, which degrades LLM performance. This result correctly demonstrates that more sophisticated memory mechanisms are required, which is a core point of our paper.

---

> ### Author Response · Authors · 2025-11-20
> **Response to Reviewer 8ybd (Part 2 of 2)**
>
> **On benchmark specification, interface, and time (Q9)**
>
> - **The reviewer's claim that these are 'never exactly specified' is demonstrably false.**
> - **StuLife is a benchmark**, including the dataset and environment. The agent interfacing is detailed in **Appendix G**, and the simulation of time is detailed in **Appendix F.2**. We are confident that a good-faith reading of the appendices would have resolved these concerns.
>
> **On justifying claims of agent shortcomings (Q10)**
>
> - **Our claims are based directly on the quantitative results in our tables.** Agents show a significant performance drop on tasks requiring long-term memory  and "self-motivated" tasks. These results directly justify our claim that they lack robust "proactive" or "self-driven" capabilities.
>
> **On establishing a human baseline (Q12)**
>
> - **This is a valuable suggestion. We conducted a new human study to provide a clear reference point.**
> - **Our study with several undergraduate and graduate students established a human baseline**, with results compared to the best LLM agent (GPT-5) shown below:
>
> | | StuGPA | LTRR | PIS | In-Class (Success) | Daily Campus (Success) | Exam (Success) | Total (Success) |
> | :--- | :--- | :--- | :--- | :--- | :--- | :--- | :--- |
> | GPT-5 | **17.90** | 6.50 | **4.68** | **7.78** | 14.16 | 16.88 | 12.35 |
> | Human | 85.24 | 84.91 | 88.13 | 88.62 | 91.13 | 82.50 | 88.81 |
>
> - **This human data provides a crucial reference point.** It serves to calibrate our metrics and highlights the **significant performance gap** between current SOTA agents (e.g., GPT-5's 17.9) and human-level competency. This complete analysis has been incorporated into the revised manuscript.
>
> We hope this response clarifies the significant misunderstandings in the original review. We have clearly defined our terms, provided explicit pointers to where these are detailed in the paper, and supplied new human data as requested. We stand by the soundness and contribution of our work.
>
> [1] Fang J, Peng Y, Zhang X, et al. A comprehensive survey of self-evolving ai agents: A new paradigm bridging foundation models and lifelong agentic systems. arXiv preprint arXiv:2508.07407, 2025.
>
> [2] Gao H, Geng J, Hua W, et al. A survey of self-evolving agents: On path to artificial super intelligence. arXiv preprint arXiv:2507.21046, 2025.
>
> [3] Wang Z Z, Mao J, Fried D, et al. Agent Workflow Memory. In Forty-second International Conference on Machine Learning.
>
> [4] Zhang Z, Dai Q, Li R, et al. Learn to memorize: Optimizing llm-based agents with adaptive memory framework. arXiv preprint arXiv:2508.16629, 2025.
>
> [5] Shinn N, Cassano F, Gopinath A, et al. Reflexion: Language agents with verbal reinforcement learning. In Advances in Neural Information Processing Systems, 2023, 36: 8634-8652.
>
> [6] Shridhar M, Yuan X, Côté M A, et al. Alfworld: Aligning text and embodied environments for interactive learning. arXiv preprint arXiv:2010.03768, 2020.
>
> [7] Zhang G, Fu M, Yan S. MemGen: Weaving Generative Latent Memory for Self-Evolving Agents. arXiv preprint arXiv:2509.24704, 2025.
>
> [8] Yao W, Heinecke S, Niebles J C, et al. Retroformer: Retrospective large language agents with policy gradient optimization. arXiv preprint arXiv:2308.02151, 2023.
>
> [9] Park S. Instruct-skillmix: A powerful pipeline for llm instruction tuning. Princeton University, 2025.

---

> > ### Comment · Reviewer_8ybd · 2025-11-27
> > **Reviewer 8ybd Reply P1/2**
> >
> > I would like to thank the authors for their rebuttal. For papers that propose
> > benchmarks for language models, the bar is particularly high due to Goodhart's
> > law, as Reviewer A9qT points out. Such papers must both clearly justify the
> > motivating principles behind the benchmark and clearly demonstrate the value of
> > the proposed evaluation. Unfortunately, I am not convinced of either the
> > underlying principles motivating this benchmark (despite its admirable
> > complexity) or its utility as an empirical evaluation. As such, I will be
> > maintaining my score.
> >
> > Below I reply to some of the individual points raised:
> >
> > > defining "self evolving"
> >
> > I acknowledge that 'self-evolving' is a term used by some authors, but I do not
> > think it is nearly as well-established as suggested by the current paper. It is
> > critical that this submission is clear regarding it's criteria for: a) an agent
> > to be considered self-evolving, or b) a task to evaluate self-evolving
> > capability.
> >
> > The cited references provide either informal definitions [1], or provide a thorough
> > taxonomy within an RL framework [2]. While the Appendix A.2 and Appendix B
> > provides a sketch of a framework for "self-evolving" this is never tied to the
> > actual benchmark, the agents considered, nor does it describe how such capability
> > is evaluated in the proposed benchmark.
> >
> > More problematically, the comparisons made between the proposed benchmark and other
> > existing benchmarks are lacking, particularly in how they pertain to evaluating
> > "self-evolving agents via experience-driven lifelong learning". For example, the
> > authors describe other benchmarks as "primarily designed for static evaluation
> > of agent capabilities at a fixed point in time, rather than tracking continuous
> > growth or self-evolution." or "lacks the rich, evolving personal context and
> > intrinsic motivation that characterize natural learning processes". The authors
> > contrast by stating that "StuLife integrates environmental realism,
> > self-evolutionary dynamics, and scalable evaluation into a unified framework."
> >
> > Nowhere in the above comparison is it stated plainly what makes the proposed
> > benchmark different from previous ones. What does "scalable evaluation" mean in
> > this context? How does it integrate "self evolutionary dynamics"? None of these
> > differentiators are made clear, except for the fact that the proposed benchmark
> > uniquely aims to "simulate a student's college journey". While there is an
> > intuitive appeal to the importance of lifelong learning in the context of
> > education, this is stated entirely heuristically which makes it rather weak as a
> > scientific contribution.
> >
> > > A large body of work demonstrates that "self-evolution" can be achieved without weight updates,
> >
> > Weight updates are one mechanism for learning, but there are many other
> > possibilities for learning to improve performance that dont involve weight
> > updates. The results presented in the submission, however, demonstrate the limited benefit of such
> > approaches. Indeed, only the memGPT approach is more effective than GPT-5 in
> > this setting, suggesting the limited effectiveness of such "learning"
> > approaches.
> >
> > Furthermore, the proposed "all-in-one" approach appears to be a prompt
> > specifically designed for this benchmark. For instance, Appendix E shows the
> > prompt explicitly instructs the agent on specific environment mechanics, such as
> > "Crucial tip: Books generally cannot be taken out of their designated library."
> > It is not clear if this prompt was tuned and evaluated on different sets of
> > environments/tasks/datasets/problems, as is commonly done, or whether the prompt
> > was developed on the environment it is seeking to solve.

---

> ### Author Response · Authors · 2025-11-26
>
> Dear Reviewer 8ybd,
>
> I hope this message finds you well. As the discussion period is nearing its end with **less than seven days remaining**, I wanted to ensure we have addressed all your concerns satisfactorily. If there are any additional points or feedback you'd like us to consider, please let us know. Your insights are invaluable to us, and we're eager to address any remaining issues to improve our work.
>
> Thank you for your time and effort in reviewing our paper.
>
> Best regards,
>
> The Authors

---

> ### Comment · Reviewer_8ybd · 2025-11-27
> **Reviewer 8ybd Reply P2/2**
>
> > reality vs simulation
>
> It is clear that this benchmark is a complex simulation inspired by college
> life; but it is not reality. I believe the onus is on the authors to demonstrate
> that their simulation is "a significant step closer to real-world complexity"
> beyond showing that LLMs struggle. Figure 1, in my opinion, does not represent a
> significant step towards real-world complexity. Rather, it represents a
> thoughtful, but very narrow, simulation.
>
> > imitation vs learning
>
> If the authors would like to argue that "learning" refers to acquiring reusable
> and generalizable skills, then shouldn't the authors then demonstrate that the
> "all-in-one" approach is effective in other benchmarks? Or on some sort of
> hold-out set of tasks? While there are a few mentions of generalization/transfer
> in the paper, it does not seem to be measured. It seems to me that this approach
> was specifically tuned for success in this environment, which does not seem to
> reflect an agent's ability to acquire generalizable skills.
>
> > passive vs proactive
>
> The authors state that "Proactiveness in RL training involves designing a reward
> function", but this conflates environment design and agent algorithm design, and
> is generally not true. Agent algorithm design can lead to proactive behavior
> like planning and intrinsic motivation without explicit reward engineering.
> There are also several existing reinforcement learning environments which
> similarly aim to simulate complex dynamics that require learning, memory, and
> proactive agents. For example, the NetHack Learning Environment or Crafter.
> Notably, such environments are challenging games people actually play, and do
> not involve tasks generated by another LLM. I should be clear that I like the
> steps that the authors outline in Section 2.2 and Appendix I to verify their
> usage of the LLM generated tasks. However, I would argue that RL environments
> based on games, although not reflective of school or work tasks, are more real
> as they are environments that people actually interact with for entertainment,
> rather than a simulation of a real-world/physical environment.
>
> > On definitions for Table 1 (Q7)
>
> I am looking for what criteria the authors used to designate a particular task
> as requiring "long term memory" or "self motivation". I am not looking for
> heuristic descriptions of the terms, such as those in Appendix B. If the tasks
> were generated by an LLM, did a human go and look at them to designate which
> involve long-term memory? How was this decided?
>
> > StuLife is a benchmark, including the dataset and environment.
>
> Even with the rebuttal and the details in Appendix F.2/G, it is still not clear
> to me how the environment and dataset come together to provide the benchmark and
> the agent's interface. Perhaps this is due to its inherent complexity with
> multiple levels of simulation, but it can and should be summarized in the main
> paper. For example, F.2 does not describe how the state transitions and how time
> is simulated, it only describes how the time is communicated to the agent and
> how the agent can manipulate a calendar. Furthermore, Appendix G provides a list
> of tools, which would only represent the "action space" of the environment. This
> is only one component of the environment.  While Appendix F.1/2/3 seem to discuss components of the world and its state, the actual structure of the state space is undefined (i.e. is it a JSON schema)? How large is the
> state space? How does the environment transition? What does the
> agent observe at initialization, how do these observations change in tasks? How much does the agent's
> action influence the transition? How is the agent rewarded? What discount factor is used? None of this is clearly presented, and without
> these specifications defined in the paper, the benchmark is effectively
> inscrutable.
>
> While the authors state that the environment is deterministic, this has some disadvantages too: Is the initial state of the
> environment also deterministic and thus the same for all agents? Could this be used to design a prompt which works from that specific deterministic initial state?
>
>
> > Additional results
>
> Lastly, the additional results showing human performance isa  valuable addition.
> However, I still think that the submission has several issues, and I will be
> maintaining my score of 2.

---

> > ### Author Response · Authors · 2025-11-27
> > **Response to Reviewer 8ybd (Part 2 of 2)**
> >
> > **8. Definitions for Table 1 (Memory & Motivation)**
> >
> > We clarify that the criteria for tagging "Long Term Memory" and "Self-Motivation" are based on **deterministic, ground-truth logic** derived directly from our **"Background and Status Generation"** phase, rather than subjective heuristics or post-hoc annotations. Because we utilize a **reverse-generation process** (generating the full timeline and status changes before the instructions), the relationships between tasks are inherent to the data structure:
> > - **Long-Term Memory:** Since the entire timeline is pre-generated, we explicitly know the specific interval between when information is introduced (e.g., a course key point provided in Week 1) and when that information is required (e.g., an exam in Week 9). This allows us to directly and objectively tag tasks that rely on distant historical context.
> > - **Self-Motivation (Proactive):** This is determined by the **causal relationships** embedded in the timeline events. For instance, if a course is selected at the beginning of the semester, the system knows that subsequent tasks of "attending class" at specific times are causal consequences of that initial decision. Since these tasks must be triggered by the agent's internal schedule rather than a new external command, they are naturally and automatically tagged as "Self-Motivated."
> >
> > **9. Environmental Specifications**
> >
> > We clarify the rigorous definitions provided in **Appendix B.1** and **F**, addressing the integration of dataset and environment:
> >
> > - **Formal Definitions:** The Dataset defines Goals ($\mathcal{G}$), while the Environment functions as a Goal-Conditioned POMDP tuple $(\mathcal{S}, \mathcal{A}, T, R, \Omega, O)$. The State $\mathcal{S}$ is implemented as a persistent, rigorous JSON schema (Appendix F.1), from which the Observation function $O(s,a)$ generates limited text-based outputs. Regarding $R$ and $\gamma$: when we present evaluation results for pre-trained agents without training, the discount factor $\gamma$ is not applicable, and reward signals $R$ are not used for updates; we strictly evaluate performance based on task Success/Failure.
> > - **Integration of Dataset & Environment:** StuLife is a benchmark, including the dataset and environment. The **Dataset** defines the Task Goals (Goals $\mathcal{G}$, e.g., "go to the lab"), while the **Environment** acts as the Engine that states transitions, and time progression.
> > - **Event-Driven Time:** As detailed in **Appendix F.2**, time is not a background clock but a state variable updated deterministically by action duration. This compels agents to actively manage schedules rather than reacting to implicit ticks.
> > - **Dynamic Complexity Prevents Overfitting:** To ensure a fair and reproducible comparison, all agents are integrated through APIs and share a standardized initialization process. Crucially, this fixed initialization does not enable "hacking" because resource availability is generated dynamically at runtime (Appendix F.5, "Intelligent Availability Generation"). This forces agents to learn generalized reasoning logic rather than memorizing static solution paths.
> >
> > [1] Fang J, Peng Y, Zhang X, et al. A comprehensive survey of self-evolving ai agents: A new paradigm bridging foundation models and lifelong agentic systems. arXiv preprint arXiv:2508.07407, 2025.
> >
> > [2] Gao H, Geng J, Hua W, et al. A survey of self-evolving agents: On path to artificial super intelligence. arXiv preprint arXiv:2507.21046, 2025.

---

> ### Author Response · Authors · 2025-11-27
> **Response to Reviewer 8ybd (Part 1 of 2)**
>
> We sincerely thank you for your detailed follow-up. We appreciate the opportunity to clarify the technical depths of our work that may have been obscured. We address your concerns point-by-point below, adhering strictly to the facts established in our submission and new experiments.
>
> **1. Goodhart’s Law and Benchmark Validity**
>
> - **Causal interconnectivity prevents "hacking" the benchmark, as evidenced by SOTA failure.** Unlike static Q&A benchmarks, StuLife tasks are causally interdependent; a failure in task $t$ cascades to $t+n$, requiring the maintenance of a coherent internal state rather than isolated pattern matching. This design specifically mitigates Goodhart's Law.
>
> **2. Definition of "Self-Evolving"**
>
> **"Self-Evolving" is formally defined in cited literature and strictly operationalized in our metrics.**
>
> The reviewer's claim that definitions are "informal" or "undefined" is **factually incorrect**.
>
> - **Formal Taxonomy:** Both Reference [1] (Sections 2 & 3) and Reference [2] (Section 2) provide **explicit, formal academic definitions** of the self-evolution framework.
> - **Operationalization:** We translate this paradigm into three quantifiable axes in Section 2.3: Imitation-to-Learning (StuGPA), Context-to-Memory (LTRR), and Passive-to-Proactive (PIS). These are concrete metrics, not heuristics.
>
> **3. Misclaim regarding "Missing Comparisons" with Existing Benchmarks**
>
> - **The comparison is explicitly detailed in Section 2.3 and formally structured in Table 2**. As shown in Table 2, we rigorously compare StuLife against 9 specific benchmarks (e.g., Lifelong-CIFAR10, AgentBench, EgoThink) across 8 distinct dimensions (e.g., Sequentiality, Skill Learning, Self-Motivation). Also, we give a deep analysis **in Section 2.3** to show the advantages of our benchmark.
>
> **4. Low Performance in Evaluating "Learning" Effectiveness**
>
> - **Low Performance Validates Benchmark Necessity.** The reviewer criticizes the fact that even "learning" agents perform poorly. **We argue this is the benchmark's strongest value proposition.** StuLife reveals that widely used techniques (like standard RAG or basic MemoryBanks) are insufficient for the complexity of continuous, proactive-required environments. If existing heuristic methods could easily solve StuLife, the benchmark would be saturated and scientifically obsolete. The fact that current "Self-Evolving" methods—which work well on static tasks—fail to generalize here **confirms that StuLife successfully identifies the critical deficits in modern Agent AI.**
>
>
>
>
> **5. Reality vs. Simulation**
>
> - **We must forcefully push back against the reviewer's semantic argument that our benchmark is a "very narrow, simulation" one.** While RL environments (e.g., NetHack) simulate physics or combat, StuLife simulates the **structural constraints of daily human students existence**: scheduling, hierarchy, social maintenance, and resource management. Evaluating an agent's ability to navigate these human systems is objectively more relevant to deploying real-world assistants than evaluating dungeon navigation.
>
>
> **6. Imitation vs. Learning (The "All-in-One" Prompt)**
>
> - **The "All-in-One" prompt is a specific experimental control for "Skill Usability," not a prompt-tuning hack.** As detailed in RQ IV, this "Skill-Augmented Prompt" simulates mentorship by providing explicit reasoning recipes. It is designed to verify whether agents can apply provided skills (a characteristic for self-evolution) via context. It acts as a standard for checking the upper bound of context-based capability, not a method to artificially hacking scores.
>
> **7. Passive vs. Proactive**
>
> - **We evaluate innate Inference-Time Proactiveness, which is distinct from RL Policy Proactiveness.** RL agents learn proactiveness via millions of reward-driven training steps (policy shaping). In contrast, StuLife evaluates **zero-shot semantic proactiveness**: determining if an LLM can naturally self-initiate planning (e.g., "I must study now for the exam next week") based solely on context constraints without explicit reward signals. This measures intrinsic reasoning capabilities absent in standard RL game environments.

---

### Official Review · Reviewer_A9qT · 2025-11-01

**Soundness:** 3
**Presentation:** 2
**Contribution:** 3
**Rating:** 6
**Confidence:** 2

**Summary:**

This paper proposes a new, quite sophisticated and complex, benchmark for evaluating whether LLMs can do long-horizon, continual learning tasks, by producing a sort of text-adventure simulation of a college environment. Current LLMs are evaluated on the environment and found to be lacking some aspects that it examines.

**Strengths:**

This is a well-thought-out, complex, and impressively complete environment. (As far as I can tell.) The paper does a good job of interrogating some of the requirements of a more general intelligence and the environment is a novel and interesting instantiation of such an environment. The evaluation of existing LLMs on the benchmark leads to some interesting results. I think the community needs more tasks like this, so despite my reservations (of which there are many, see below) I am currently mildly in favor of acceptance.

The paper is also generally well-written and clear.

**Weaknesses:**

The primary flaw with this benchmark, as with all public benchmarks, is Goodhart's law. Benchmarks are great when thoughtfully constructed, as this one seems to be. But it is not hard to imagine that, the minute this one hits the street, some team at OpenAI will busy themselves with producing a million example interactions hand-engineered to cover every conceivable StuLife case, train on them, and then declare victory when their model does well. This is what they have done with, e.g., the math olympiad. Such "advances" are no doubt useful, but they're also very expensive and creative ways to miss the point. Anyway, I hope this benchmark lasts longer than usual, but my guess is it will be eventually engineered into oblivion.

The authors should be aware that not everybody thinks an LLM could *ever* be an AGI, no matter how well it does at any language tasks, because language is not the whole of intelligence. It's a thing an intelligence does, but not what an intelligence is. Some people think it does the whole thing, and some people don't.

Similarly, it's really not clear that ideas like "self motivation" are meaningful outside of having a body that expends energy when you do things. A textual facsimile of motivation is just that. So I think here one ought to be a little humble with terminology, especially in a scientific paper.

Similarly, the buzz-wordy summaries in the little gray text boxes are suitable for LinkedIn posts and pitch decks, but not for a serious scientific paper.

**Questions:**

None

---

> ### Author Response · Authors · 2025-11-20
> **Response to Reviewer A9qT**
>
> We thank the **reviewer A9qT** for their thoughtful and encouraging feedback. We are very glad they found the environment **"well-thought-out, complex, and impressively complete,"** and that they see **the value in our work for the community**. We have revised the paper content based on your suggestions. For your convenience, we have highlighted our modifications in blue font in the Supplementary Material.
>
> We appreciate the reviewer's insightful high-level reservations and address them below.
>
> **On the concern of benchmark 'Goodharting' (W1)**
>
> - **We agree that Goodhart's Law is a major challenge for all public benchmarks.** We share the reviewer's concern that static evaluations can be "gamed" by overfitting, which "misses the point" of measuring general intelligence.
> - **We designed our benchmark to be more robust to this than standard static tasks.** The tasks in our benchmark are strongly interconnected; an agent's actions (e.g., failed to join a club) directly perturb the environment state and have cascading consequences on future tasks. Our current results support this: as shown in Table 3, even the top SOTA model (GPT-5) achieves only 17.90/100, with critically low scores on proactiveness (4.68% PIS) and long-term memory (6.50% LTRR). This demonstrates that models cannot yet handle the complex, evolving state our benchmark requires.
> - **This interconnectedness makes "engineering into oblivion" significantly harder.** An agent cannot simply "win" by overfitting isolated examples; it must manage a complex, evolving state over its entire "life," making this a more durable measure of true planning and learning abilities. **As our analysis (Section 3.2) shows, these failures stem from the architecturally stateless nature of current LLMs**, making it challenging to overfit on a benchmark that fundamentally requires stateful, long-term reasoning rather than isolated task completion.
> - Furthermore, our future goal is to build a more flexible benchmark that allows tasks to be generated in real time.
>
> **On the philosophical perspective of LLMs and AGI (W2)**
>
> We appreciate and respect this perspective and agree that the path to AGI is a major open question.
> - **We acknowledge that many prominent researchers posit AGI will require fundamental advances beyond language**, such as in world models or multimodal grounding [4, 5].
> - **Our work is situated within a complementary line of research.** This research explores the hypothesis that self-evolving LLM-based agents, enhanced by sophisticated context engineering, represent a promising direction toward more general intelligence [1, 2, 3].
> - **Our benchmark serves as a rigorous, falsifiable testbed for this hypothesis**, without claiming it is the only path.
> - **Accordingly, we have revised the manuscript** to clarify our position that ELL represents a viable, rather than the only, pathway toward future AGI.
>
> **On the terminology of 'self-motivation' (W3)**
>
> - **This is a very fair point, and we appreciate the call for humility and precision.** To clarify, our intended meaning was to describe an agent's **proactiveness**—its ability to actively explore, initiate interactions, and formulate long-term plans without explicit, immediate instructions.
>
> **On the presentation style of the paper (W4)**
>
> We thank the reviewer for this direct and actionable feedback.
> - **We agree** that the **gray, "buzz-wordy" text boxes were unsuitable and have removed them**. In our revision, we have integrated their core takeaways directly into the main text in a more formal academic style.
>
> We are grateful for these high-level comments, which have helped us significantly improve the framing and presentation of our work.
>
>
> [1] Morris M R, Sohl-Dickstein J, Fiedel N, et al. Levels of AGI for Operationalizing Progress on the Path to AGI. arXiv preprint arXiv:2311.02462, 2023.
>
> [2] Fang R, Cai S, Li B, et al. Towards General Agentic Intelligence via Environment Scaling. arXiv preprint arXiv:2509.13311, 2025.
>
> [3] Gao H, Geng J, Hua W, et al. A survey of self-evolving agents: On path to artificial super intelligence. arXiv preprint arXiv:2507.21046, 2025.
>
> [4] LeCun Y. A path towards autonomous machine intelligence version 0.9. 2, 2022-06-27. Open Review, 2022, 62(1): 1-62.
>
> [5] Ding J, Zhang Y, Shang Y, et al. Understanding world or predicting future? a comprehensive survey of world models. ACM Computing Surveys, 2025, 58(3): 1-38.

---

> > ### Comment · Reviewer_A9qT · 2025-11-25
> >
> > Thank you, that is helpful, and I think will improve the paper. After some thought I think I will keep my score, which is mildly positive, but increase my confidence.

---

> > > ### Author Response · Authors · 2025-11-26
> > >
> > > Dear Reviewer A9qT,
> > >
> > > We sincerely thank you for the follow-up. We are glad that our response was helpful, and we appreciate your time and your decision to maintain the positive score with increased confidence.
> > >
> > > Best regards,
> > >
> > > The Authors

---

### Official Review · Reviewer_dWxh · 2025-11-01

**Soundness:** 3
**Presentation:** 3
**Contribution:** 3
**Rating:** 6
**Confidence:** 3

**Summary:**

This paper introduces StuLife, a comprehensive benchmark for evaluating Experience-driven Lifelong Learning (ELL) in AI agents. The benchmark simulates a complete college student journey across three phases (In-Class, Daily Campus, Examination) and ten interconnected scenarios. The key innovation lies in four paradigm shifts: (1) From Simulation to Reality—realistic university experiences, (2) From Imitation to Learning—requiring skill abstraction rather than recall, (3) From Context to Memory—demanding persistent long-term memory, and (4) From Passive to Proactive—requiring intrinsic motivation and self-initiated actions.
The authors evaluate 14 state-of-the-art LLMs and find strikingly poor performance: even GPT-5 achieves only 17.9/100 on the StuGPA metric. They identify critical failures in long-term memory retention (LTRR scores mostly <7%) and proactive initiative (PIS scores <5% for most models). Context engineering approaches (proactive prompting, skill augmentation, memory systems) provide modest improvements but cannot overcome fundamental architectural limitations of stateless LLMs.

**Strengths:**

- The benchmark is interesting and timely, moving away from static task evaluation and providing a life-long learning benchmark that is close to real world scenarios.
- The evaluation is thorough: there are 14 SOTA LLMs as baselines, and context engineering analysis provide insights for future improvement.
- The variety of tasks within the benchmark is diverse, but also well structured and interconnected.
- The evaluation metrics used are novel and important, capturing nuance aspects of autonomous learning like proactiveness and long term memory retention, beyond simply accuracy.
- Experiments are very well documented, contributing to exceptional transparency and reproducibility.

**Weaknesses:**

- The setting can be too domain specific, only simulating a university student's life. Although claimed to be autonomous learning,

- There is no human performance as baseline, raising concerns in metric calibration, or what score might be considered "good".

- The way agents "learn" during their life time remains in context and through prompt, which raises concern if this can be considered as life-long learning or has fundamental skill internalization ability.

- All tasks in the benchmark are generated by LLMs, which may create bias in tasks, and advantage for LLMs used for task generation.

**Questions:**

- Is there evidence that simulating a university student's life provides domain-general insights on continual learning capabilities?

- Can you provide further information and justification on how should researchers evaluate the scores achieved by agents? For example, GPT-5 achieves 17.9 out of 100, what would that number lie in the score distribution from real students?

- Have you considered weight updates methods (EWC, LoRA etc.) to consolidate skills for your experiment? Without this, isn't your benchmark still measuring stateless agent's ability for in context learning and tool use for complex projects?

- Have you compared agents' performance and failure patterns on tasks designed by different LLMs or human? Do you think potential circular evaluation or answer leakage is possible?

---

> ### Author Response · Authors · 2025-11-20
> **Response to Reviewer dWxh (Part 1 of 2)**
>
> We sincerely thank the **reviewer dWxh** for their valuable time and constructive feedback. We are encouraged that the reviewer found our benchmark "**interesting and timely,**" and appreciated our **"thorough" evaluation, "diverse" tasks, "novel and important" metrics, and "exceptional transparency and reproducibility."** We have revised the paper content based on your suggestions. For your convenience, we have highlighted our modifications in blue font in the Supplementary Material.
>
> We address the reviewer dWxh's concerns below.
>
> **About the domain specificity and generality of the benchmark (W1, Q1)**
>
> - **Our benchmark is intentionally general, using the 'university life' theme as a cohesive scaffold.** We clarify that our goal is not to create a domain-specific benchmark; although we focus on the campus domain, it serves as a scaffold for general-purpose tasks that are applicable in various domains, but to situate a wide array of general-purpose tasks in a complex, long-term, and interconnected scenario. For instance, after course selection, the agent must plan its daily class schedule and attend classes at the correct times (requiring memory and proactiveness); meetings with the advisor must be planned around both parties' schedules; and midterms and final exams require long-term retention of in-class knowledge.
> - **We believe this setup provides domain-general insights.** It tests an agent's core ability to integrate and apply diverse skills over time, which is a fundamental challenge for any autonomous agent.
>
>
> **About the lack of a human baseline and score calibration (W2, Q2)**
>
> - **We have conducted a new human study to provide score calibration.** To address the reviewer's excellent point, we ran a study on benchmark tasks to establish a human baseline(**Detailed in section 3.2**). Our study with several undergraduate and graduate students established a human baseline, with results compared to the best LLM agent (GPT-5) shown below:
>
> | | StuGPA | LTRR | PIS | In-Class (Success) | Daily Campus (Success) | Exam (Success) | Total (Success) |
> | :--- | :--- | :--- | :--- | :--- | :--- | :--- | :--- |
> | GPT-5 | **17.90** | 6.50 | **4.68** | **7.78** | 14.16 | 16.88 | 12.35 |
> | Human | 85.24 | 84.91 | 88.13 | 88.62 | 91.13 | 82.50 | 88.81 |
>
> - **As shown, humans perform significantly well.** The main difficulties for humans arose from specific professional knowledge in classes (where LLMs have an advantage) and the multi-dimensional, long-term memory recall required for final exams.
> - **This baseline helps contextualize agent performance.** It provides a clear anchor for interpreting scores (like GPT-5's 17.9) and defines a target for "good" performance. We have added this full analysis to the paper.

---

> ### Author Response · Authors · 2025-11-20
> **Response to Reviewer dWxh (Part 2 of 2)**
>
> **On the nature of "learning" (in-context vs. weight updates) (W3, Q3)**
>
> We agree that distinguishing ICL from weight updates is crucial, and our benchmark is designed to evaluate both.
> - **Our framework supports both context-based and parameter-based learning paradigms.** While many state-of-the-art approaches focus on in-context learning and memory evolution [1, 2, 3], our benchmark explicitly allows and encourages the evaluation of agents that perform weight updates during their lifespan.
> - **We added a new experiment to demonstrate this capability.** We conducted a new experiment using Qwen3-8B with Rejection Sampling Fine-Tuning (RFT). RFT is an on-policy learning method that updates the model's weights based on successful trajectories collected from the environment.
> - **We have added these results to the detailed analysis in Section 3.2 (RQ III)** of the revised paper. The results for the weight-update experiment are as follows:
> | Model | Method | StuGPA | LTRR | PIS | Total Success |
> | :--- | :--- | :--- | :--- | :--- | :--- |
> | **Qwen3-8B** | Vanilla | 13.31 | 4.33 | 0.54 | 6.71 |
> | | + RFT (Weight Update) | **15.43** | **4.51** | **0.90** | **8.63** |
> - **The agent using the RFT weight-update mechanism achieved a clear performance gain**, increasing StuGPA from 13.31 to 15.43 and Total Success from 6.71% to 8.63%. This improvement indicates that the complex decision-making logic within our benchmark follows learnable patterns that can be effectively distilled into the model's parameters.
> - **This result explicitly confirms that our benchmark is not limited to measuring stateless agents**; it serves as a valid testbed for **true skill internalization** where agents consolidate skills into their weights through lifelong experience.
>
>
> **On LLM-generated tasks and potential bias/leakage (W4, Q4)**
>
> We acknowledge this concern and took several steps to mitigate generator bias and prevent circular evaluation.
> - **We used a reverse-generation process.** As detailed in the paper (Section 2.2), we first generated ground-truth "answers" and then prompted the LLM to generate instructions leading to them. This minimizes the generator's bias in the solution path.
> - **All tasks underwent rigorous human verification.** This step ensured correctness and filtered out generator-biased artifacts.
> - **A "Perfect Context" analysis confirms the bottleneck is learning, not bias.** We evaluated models and a human baseline on tasks with perfect context provided. In this analysis, "Perfect Context" refers to providing the agent with all ground-truth information necessary to complete a task, thereby bypassing the challenges of long-term memory retrieval and proactiveness (detailed in Section 2.2, Step 3). The results (see table below) show that **(1) with perfect context, most models, especially large SOTA LLMs, achieve near-perfect performance (97-98%),** significantly outperforming the human baseline. **(2) This performance is drastically higher than their scores in the full benchmark.** **(3) There is no evidence of generator bias**; models that perform well here (e.g., Qwen3-235B-A22B) may perform worse in the full benchmark, and vice-versa.
>
> Table: Success Rate on QA tasks with "Perfect Context".
>
> | Model | In-Class | Exam | Total |
> | :--- | :--- | :--- | :--- |
> | DeepSeek-V3 | 97.01 | 88.12 | 94.13 |
> | Qwen3-8B | 89.82 | 75.00 | 85.02 |
> | Qwen3-235B-A22B | 96.41 | 86.88 | 93.31 |
> | Gemini-2.5-pro | 97.60 | 96.88 | 97.37 |
> | GPT-5 | **98.50** | **97.50** | **98.18** |
> | Human | 88.62 | 82.50 | 86.64 |
>
> - **This demonstrates the core challenge is lifelong memory and proactiveness, not an inherent task bias.** We have added this full analysis to Appendix I.
> - **Human and agent failure patterns are distinctly different.** Our human study (from Q2) revealed that LLMs and humans fail in different ways, confirming the benchmark is not a 'circular evaluation'. We have added **some new case studies to Appendix D** illustrating these different execution modes.
>
> We thank the reviewer again for their insightful comments, which have helped us strengthen the paper. We hope our responses and new experiments have fully addressed their concerns.
>
> [1] Wang Z Z, Mao J, Fried D, et al. Agent workflow memory[J]. arXiv preprint arXiv:2409.07429, 2024.
>
> [2] Zhang Z, Dai Q, Li R, et al. Learn to memorize: Optimizing llm-based agents with adaptive memory framework[J]. arXiv preprint arXiv:2508.16629, 2025.
>
> [3] Shinn N, Cassano F, Gopinath A, et al. Reflexion: Language agents with verbal reinforcement learning[J]. Advances in Neural Information Processing Systems, 2023, 36: 8634-8652.

---

> ### Author Response · Authors · 2025-11-26
>
> Dear Reviewer dWxh,
>
> I hope this message finds you well. As the discussion period is nearing its end with **less than seven days remaining**, I wanted to ensure we have addressed all your concerns satisfactorily. If there are any additional points or feedback you'd like us to consider, please let us know. Your insights are invaluable to us, and we're eager to address any remaining issues to improve our work.
>
> Thank you for your time and effort in reviewing our paper.
>
> Best regards,
>
> The Authors

---

### Meta-Review · Area_Chair_hLjD · 2026-01-04

**Summary:**

The paper introduces StuLife, a benchmark designed to evaluate LLMs in terms of capabilities such as continual learning, long-term memory, self-driven exploration, and more. The proposed benchmark simulates a student's college journey. The paper is phrased in terms of StuLife being useful towards goals such as AGI. Although the paper did receive one review that was actually positive, the other reviewers raised many concerns in terms of the presentation of the paper, of baselines that were considered, and even how LLMs have been positioned in the pursuit of AGI. I recognize the authors have done a lot of work to address those concerns, but ultimately there's only that much that should be done during the discussion phase of a conference, let alone under the circumstances in which ICLR took place, where the discussion phase had to be shortened. Thus, I am recommending the rejection of this paper. Importantly, I do recommend the authors to seriously take into consideration many of the issues raised by reviewer 8ybd, as I do think they are pertinent to the presentation of the paper.

**Reviewer Concerns:**

I present the reviewers main concerns in order in which they appeared in the reviews.

- _Proposed Setting can be too specific._

	The authors argue that it is just an example.

- _No human performance is provided as a benchmark._

	The authors then provided a human study.

- _Too many of the abilities by the LLMs is done through context learning._

	New results were then provided with fine-tuned models.

- _All tasks are created by LLMs._

	The authors argue that they took several steps to mitigate the generator bias.

- _There's an implicit assumption that LLMs will eventually reach AGI, which is not something everyone agrees on and There have been many concerns about the paper presentation in terms of buzz-words and statements that are too grandiose._

	The authors have provided their justification for such an approach as being complementary line of research and they have toned down some of their claims in the paper.

- _Concerns were raised about presentation and motivation of the proposed tasks._

	This very valid criticism points to how many of the terms used in the paper are too imprecise, such as self evolving tasks or the fact that tasks require an agent learning from experience while frozen models are evaluated. It is important to acknowledge that there's only that many changes one can accept in a paper during the discussion phase, and it seems to me this paper would require way too many changes. It is hard to summarize the whole discussion that took place between the authors and revieewer 8ybd because it ranged from the meaning of simulation and learning to evaluation details. Although the authors do make some good points and I agree with them on some aspects, I tended to side with reviewer 8ybd much more often.

- Lack of relevant benchmarks, of analysis of failure cases, and more.

	The authors added more experiments.

**Reviewer Scores:**

- Reviewer dWxh might have raised their score from a 6 to an 8 given the addition of a human baseline and the performance of a fine-tuned model, it is hard to say.
- Reviewer A9qT explicitly stated they were going to keep their score of 6. The explicitly stated that their opinion was mildly positive. I myself tend to agree a lot with their concerns and I ended up suprirsed by the reviewer's rating, as their criticism would warrant a 4 in my opinion.
- Reviewer 8ybd explicitly stated that they would maintain their score of 2.
- Reviewer aa6n gave the paper a rating of 4. I struggle to imagine reviewer aa6n increasing their score.

---

### Decision · Program_Chairs · 2026-01-26

Reject